# Evoking User Memory: Personalizing LLM via Recollection-Familiarity Adaptive Retrieval

**Yingyi Zhang**[1,2,*], **Junyi Li**[2,*], **Wenlin Zhang**[2], **Penyue Jia**[2], **Xianneng Li**[1,†], **Yichao Wang**[3,†],
**Derong Xu**[2,4], **Yi Wen**[2], **Huifeng Guo**[3], **Yong Liu**[3], **Xiangyu Zhao**[2,†]
[1]Dalian University of Technology, [2]City University of Hong Kong,
[3]Huawei Technologies Ltd., [4]University of Science and Technology of China
xianneng@dlut.edu.cn,wangyichao5@huawei.com,xianzhao@cityu.edu.hk

## Abstract

Personalized large language models (LLMs) rely on memory retrieval to incorporate user-specific histories, preferences, and contexts. Existing approaches either overload the LLM by feeding all the user's past memory into the prompt, which is costly and unscalable, or simplify retrieval into a one-shot similarity search, which captures only surface matches. Cognitive science, however, shows that human memory operates through a dual process: *Familiarity*, offering fast but coarse recognition, and *Recollection*, enabling deliberate, chain-like reconstruction for deeply recovering episodic content. Current systems lack both the ability to perform recollection retrieval and mechanisms to adaptively switch between the dual retrieval paths, leading to either insufficient recall or the inclusion of noise. To address this, we propose **RF-Mem** (**R**ecollection–**F**amiliarity **Mem**ory Retrieval), a familiarity uncertainty-guided dual-path memory retriever. RF-Mem measures the familiarity signal through the mean score and entropy. High familiarity leads to the direct top-$K$ *Familiarity* retrieval path, while low familiarity activates the *Recollection* path. In the *Recollection* path, the system clusters candidate memories and applies $\alpha$-mix with the query to iteratively expand evidence in embedding space, simulating deliberate contextual reconstruction. This design embeds human-like dual-process recognition into the retriever, avoiding full-context overhead and enabling scalable, adaptive personalization. Experiments across three benchmarks and corpus scales demonstrate that RF-Mem consistently outperforms both one-shot retrieval and full-context reasoning under fixed budget and latency constraints. Our code can be found in the Reproducibility Statement.

## 1 Introduction

Large Language Models (LLMs) have demonstrated remarkable performance when augmented with retrieval mechanisms. Traditional retrieval-augmented generation (RAG) primarily targets open-domain corpora, seeking to retrieve and integrate *objective facts* (Lewis et al., 2020; Li et al., 2025d; Chen et al., 2024; Han et al., 2024). In contrast, *memory retrieval* focuses on user-specific histories, preferences, and contextualized interactions, aiming to surface evidence tailored to *a particular user at a specific moment* (Zhong et al., 2024; Jiang et al., 2025; Tan et al., 2025a; Wu et al., 2025a; Xu et al., 2025c), as illustrated in the top of Figure 1. The design of memory fundamentally shapes the boundary of personalized LLMs (Zhang et al., 2025b): it can remain a static external index passively queried, or evolve into a dynamic process of recollection, thereby endowing the system with a more human-like flow of memory (Wu et al., 2025b; Hatalis et al., 2023). Just as humans sometimes recognize a familiar face instantly (i.e., an intuitive "feeling of knowing" without deliberate reasoning) and at other times reconstruct past experiences through slow chains of recollection, so too should memory retrieval move beyond static lookup. A personalized LLM needs to be flexible and alternate between fast recognition and gradual reconstruction, adapting to the demands of the interaction.

---

[*]Equal contribution.
[†]Corresponding author.

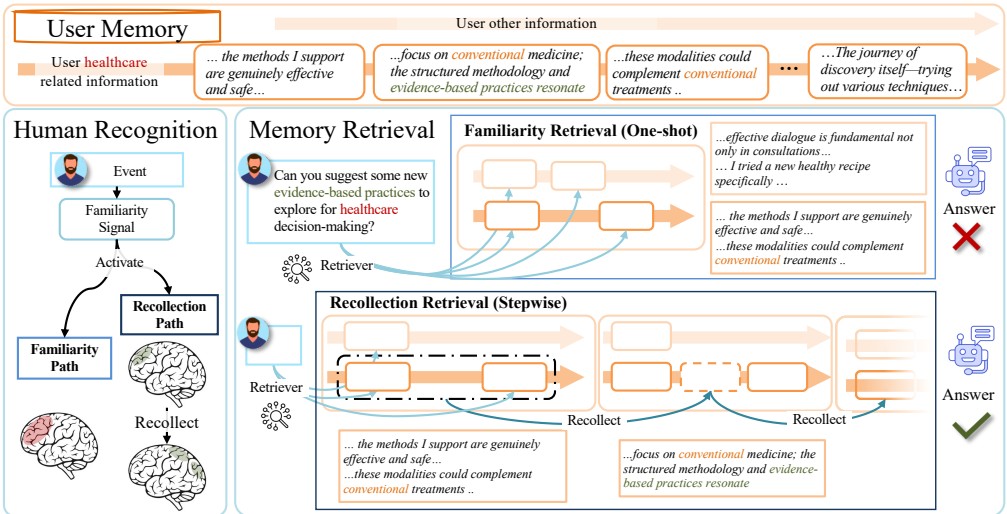

Figure 1: Comparison between standard familiarity-based retrieval and recollection-based retrieval in user health narratives. And the brain figure motivated by (Rugg & Curran, 2007; Yonelinas, 2024).

Insights from cognitive science highlight the *Recollection-Familiarity Dual-Process Theory*, as shown in Figure 1 left, which posits that human recognition and memory are driven by two complementary mechanisms (Henson et al., 1999; Yonelinas et al., 2002; Merkow et al., 2015; Bastin et al., 2019; Yonelinas, 2024). *Familiarity* provides a rapid but coarse sense of "knowing", enabling efficient yet shallow judgments. *Recollection*, in contrast, is triggered when familiarity proves insufficient, initiating a slower contextual reconstruction process that retrieves time, place, and source-specific details. Importantly, humans regulate these two processes via the familiarity signal: high confidence sustains reliance on familiarity, whereas decreasing familiarity and rising uncertainty prompt the shift to recollection (Rugg & Curran, 2007; Yonelinas, 2024). Applied to user memory retrieval in personalized LLMs, this theory suggests that retrieval should not be conceived as a one-shot operation. Instead, it should function as a dual-process controller that adaptively alternates between fast recognition and slow recollection, guided by the system's sense of familiarity.

Current retrieval systems often reduce memory to a static set of vectors, relying on direct similarity search without mechanisms to evoke richer user recollections. Retrieval in memory-augmented LLMs can be understood along three dimensions: query reformulation (Li et al., 2025a; Zhao et al., 2025; Chen et al., 2025; Shen et al., 2024; Salama et al., 2025), index construction (Zhong et al., 2024; Pan et al., 2025; Xu et al., 2025c; Tan et al., 2025b; Xu et al., 2025d; Ong et al., 2024; Chhikara et al., 2025), and retrieval strategy (Xu et al., 2021). While existing work has advanced the first two, retrieval strategies remain dominated by embedding-based one-shot top-$K$ search (Karpukhin et al., 2020; Lei et al., 2023; Wang et al., 2020; Song et al., 2020; Luo et al., 2024), corresponding to the *Familiarity* channel: fast yet shallow recall. Two key limitations remain in current memory retrieval systems: **1) they overlook the *Recollection* path**, failing to retrieve evidence chains for ambiguous queries, long-tail knowledge, or personalized reasoning; **2) they lack mechanisms for path switching between familiarity and recollection**, leading to either under-retrieval that misses deeper contextual cues or over-retrieval that introduces more retrieval latency. As shown in Figure 1, the *Familiarity* path may retrieve only partial fragments (e.g., "I support are genuinely effective and safe"), missing broader context and even introducing irrelevant noise (e.g., "new healthy recipe"). By contrast, the *Recollection* path expands iteratively and can introduce more comprehensive evidence (e.g., "focus on conventional medicine; and evidence-based practices resonate"). This gap underscores the need for a recollection retrieval path and adaptive switch mechanism to balance efficiency with reliable coverage.

To address these limitations, we propose **RF-Mem** (**R**ecollection–**F**amiliarity **Mem**ory Retrieval), an uncertainty-guided dual-path retrieval framework. RF-Mem begins with a probe retrieval that produces an initial retrieval list and estimates its familiarity by computing the mean score and entropy in the list. When familiar, the system stays on the *Familiarity* path, returning the top-$K$ candidates in a *one-shot* manner with minimal overhead. Otherwise, RF-Mem activates the *Recollection*

path: the probe results are clustered by KMeans, each cluster centroid is combined with the original query through an $\alpha$-mixing strategy, and the resulting recollect-queries are expanded iteratively. At each round, new candidates are retrieved, clustered, and mixed to form updated queries, allowing the system to *stepwise* reconstruct evidence chains. The process is explicitly bounded by beam width, fanout, and maximum rounds, ensuring controllable computation. In this way, RF-Mem preserves the efficiency of single-pass retrieval when familiarity is high, while adaptively engaging structured recollection under unfamiliar conditions, embedding chain-like reasoning directly into the retriever.

Our contributions are fourfold: (1) We ground the design of personalized memory retrieval in the Recollection–Familiarity dual-process theory, formulating retrieval as a coordination of Familiarity and Recollection paths. (2) We introduce familiarity uncertainty-driven selection for adaptive switching between Familiarity and Recollection. (3) We develop a recollection retrieval based on clustering and query–centroid mixing, achieving chain-like evidence reconstruction only in embedding space. (4) RF-Mem is lightweight, relying solely on vector search and small-scale clustering, achieving high accuracy and recall at near one-shot retrieval latency. Extensive experiments on three personalized memory datasets show that RF-Mem consistently surpasses both one-shot top-$K$ retrieval and the full-context method with low latency. An adaptive study shows that RF-Mem complements and generalizes to index-building methods like MemoryBank (Zhong et al., 2024).

## 2 Method: Recollection–Familiarity Memory Retrieval

We propose **RF-Mem**, a dual-process memory retrieval framework that adapts the retrieval strategy according to uncertainty. As illustrated in Figure 2, RF-Mem consists of five stages: ① user query input, ② user memory retrieval, *i.e.* the RF-Mem module, is a familiarity uncertainty-guided selection introduced in Section 2.1 that adaptively switches between one-shot *Familiarity* introduced in Section 2.2 and stepwise *Recollection* retrieval introduced in Section 2.2, ③ extracting the memory text, and ④ answer generation by LLM using the memory text.

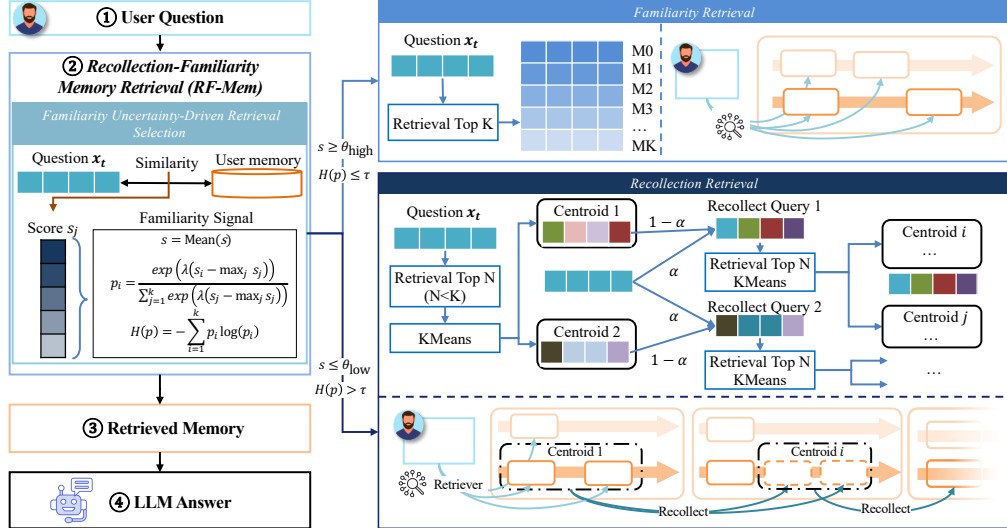

Figure 2: The overall architecture of **RF-Mem**. A dual-process memory retrieval system dynamically switches between the Familiarity and the Recollection paths.

### 2.1 Familiarity Uncertainty-Driven Retrieval Selection

Guided by dual-process theory, we argue that memory retrieval should not be reduced to a single static top-$K$ similarity search. Instead, retrieval should dynamically select between a fast Familiarity path and a deeper Recollection path depending on the familiarity signal.

We first perform a probe retrieval for calculating the familiarity signal between this question and the user's memory. Let $\mathcal{M} = \{m_1, \ldots, m_M\}$ denote the set of user memory fragments, where $m_i$ is encoded as an embedding vector $\mathbf{z}_i = \phi(m_i)$. Given a query $q$, its embedding $\mathbf{x}_t = \phi(q)$ is used

to compute cosine similarity with each fragment $m_i$ as $s_i = \langle \mathbf{x}_t, \mathbf{z}_i \rangle$, and the retriever returns the candidate set $\mathcal{C} = \text{Top-}K\big(\{(m_i, s_i)\}_{i=1}^M\big)$.

The familiarity signal is governed jointly by similarity and uncertainty, thus integrating efficiency with robustness. Formally, given probe scores $\{s_i\}_{i=1}^k$, we normalize them as follows:

$$p_i = \frac{\exp(\lambda(s_i - \max_j s_j))}{\sum_{j=1}^k \exp(\lambda(s_j - \max_j s_j))}, \qquad i = 1, \ldots, K, \tag{1}$$

where $\lambda$ controls sharpness. The uncertainty is then measured by the entropy:

$$H(p) = -\sum_{i=1}^K p_i \log p_i. \tag{2}$$

The final selection combines the familiarity of the mean score with the uncertainty of the entropy. If the mean similarity score $\bar{s} = \frac{1}{K} \sum_{i=1}^K s_i$ exceeds the upper threshold $\theta_{\text{high}}$, the memory is highly relevant and retrieval proceeds along the **Familiarity** path. Conversely, when $\bar{s}$ falls below the lower threshold $\theta_{\text{low}}$, the memory is considered weakly relevant and the system switches to the **Recollection** path. For intermediate cases where $\bar{s}$ lies between $\theta_{\text{low}}$ and $\theta_{\text{high}}$, entropy $H(p)$ serves as the disambiguator: low entropy, indicating concentrated evidence, selects **Familiarity**, whereas high entropy, reflecting uncertainty, triggers **Recollection**. Formally, the policy is defined as:

$$\text{Strategy}(q) = \begin{cases} \text{Familiarity}, & \bar{s} \geq \theta_{\text{high}}, \\ \text{Recollection}, & \bar{s} \leq \theta_{\text{low}}, \\ \begin{cases} \text{Familiarity}, & H(p) \leq \tau, \\ \text{Recollection}, & H(p) > \tau, \end{cases} & \theta_{\text{low}} < \bar{s} < \theta_{\text{high}}. \end{cases} \tag{3}$$

This gating mechanism follows our cognitive motivation: when the familiarity signal is strong, the system relies on the fast Familiarity path for confident recognition; when the signal is weak, it engages the Recollection path to deliberately reconstruct evidence; and in the intermediate regime, the entropy serves as an additional cue to regulate switching. In this way, RF-Mem mirrors the dual-process nature of human memory, dynamically balancing efficiency with retrieval depth.

## 2.2 Familiarity Retrieval

When the probe retrieval yields a familiarity signal, that is, a sufficiently high average similarity score or low uncertainty $H(p) \leq \tau$, or the system interprets the evidence as confident. In this case, it adopts the **Familiarity** path. Consistent with dual-process theory, this path corresponds to rapid, low-effort recognition, i.e., the retriever directly selects the top-$K$ memory evidence based on raw similarity, ensuring efficiency without invoking further reasoning or expansion.

**Scoring and Retrieval.** Given a query $q_t$, we encode it as $\mathbf{x}_t = \phi(q_t)$ and each memory entry $m_i$ as $\mathbf{z}_i = \phi(m_i)$, where all embeddings are unit-normalized. The similarity score is computed as $s(\mathbf{x}_t, \mathbf{z}_i) = \langle \mathbf{x}_t, \mathbf{z}_i \rangle$. We then obtain top-$K$ candidate memory fragments according to their similarity scores:

$$\mathcal{C}_t = \text{Top-}K\Big(\{(m_i, s(\mathbf{x}_t, \mathbf{z}_i))\}_{i=1}^M\Big). \tag{4}$$

This retrieval mode reflects the *Familiarity* process in dual-process theory: when the question is familiar, it signals that sufficient evidence has been retrieved, so recognition can be completed in a single step with minimal latency.

## 2.3 Recollection Retrieval

When the probe yields an unfamiliar signal, indicated by insufficient mean scores or a high entropy $H(p) > \tau$, the system interprets the evidence as uncertain. It then transitions into the **Recollection** path, corresponding to the deliberate and effortful retrieval mode in dual-process theory. Instead of stopping at surface matches, this path initiates multi-round evidence expansion, progressively reconstructing context and recovering more diagnostic evidence. A case can be found at D.7.

**Candidate Memory Retrieval.** Let $\mathbf{x}^{(0)} = \mathbf{x}_t = \phi(q_t)$ denote the query embedding. At each round $r$, we obtain the candidate memory set as $\mathcal{C}^{(r)} = \text{Top-}N\Big(\{(m_i, \langle \mathbf{x}^{(r)}, \mathbf{z}_i \rangle)\}_{i=1}^M\Big)$, where $N = (B + r) \times F < K$ is determined by beam width $B$, fanout size $F$, and round number $r \in \{0, \ldots, R\}$. We enlarge $N$ with $r$ to prevent the query in round $r$, which is formulated from previous rounds, from repeatedly retrieving the same memories. Duplicated memories in $\mathcal{C}^{(r)}$ are excluded if they appeared in earlier rounds.

**Relevant Memory Clustering.** Given the top-$N$ memory $\mathcal{C}^{(r)}$, we group the candidate memory embeddings $\{\mathbf{z}_i : m_i \in \mathcal{C}^{(r)}\}$ into $B$ clusters using KMeans. Each cluster $G_b^{(r)}$ corresponds to a branch in the retrieval tree, and its **centroid** vector $\mathbf{g}_b^{(r)}$ serves as the base for further expansion:

$$\mathbf{g}_b^{(r)} = \frac{1}{|G_b^{(r)}|} \sum_{m_i \in G_b^{(r)}} \mathbf{z}_i, \qquad b = 1, \ldots, B. \tag{5}$$

These centroids serve as branching points from which new retrieval paths unfold, simulating a tree-like process of recollection. By clustering memories into semantically coherent sets, they reduce redundancy while highlighting anchors that capture essential cues. In line with dual-process theory, these anchors initiate progressive recollection, supporting chain-like reconstruction of evidence that expands outward yet remains grounded in the user's memory context.

**Recollect Queries Generation via $\alpha$-mix.** Each centroid $\mathbf{g}_b^{(r)}$ is blended with the current query to form a *recollect query*, with weights matching the annotations $\alpha$ (current query) and $1-\alpha$ (centroid), and uses residual to maintain original query information:

$$\mathbf{x}_b^{(r+1)} = \text{norm}\big(\alpha \mathbf{x}^{(r)} + (1 - \alpha) \mathbf{g}_b^{(r)} + \mathbf{x}_t\big), \qquad \alpha \in [0, 1]. \tag{6}$$

**Retrieve-Cluster-Mix Loop.** The recollect query $\mathbf{x}_b^{(r+1)}$ will be used to perform the next round of retrieval to obtain the candidate set:

$$\mathcal{C}^{(r+1)} = \text{Top-}N\Big(\{(m_j, \langle \mathbf{x}_b^{(r+1)}, \mathbf{z}_j \rangle)\}_{j=1}^M\Big). \tag{7}$$

Afterwards, the memory cluster and recollect query mix are conducted. This *retrieve-cluster-mix* routine is repeated across rounds, progressively expanding evidence chains. To keep the search tractable, we maintain at most $B$ active branches per round and caps the recursion depth at $R$.

**Stop and Generation.** The process stops when a round limit $R$ is reached or a target number of items is gathered. The recollection evidence is a truncated union $\mathcal{C}_t = \text{Top-}K\Big(\bigcup_{r=0}^R \mathcal{C}^{(r)}\Big)$.

In analogy to human memory, this recollection triggers deliberate, cue-driven reconstruction, where related fragments are progressively retrieved, clustered, and mixed to surface latent context. Through structured retrieve-cluster-mix iterations under beam and depth budgets, which trade additional latency for more diagnostic evidence, enabling the evocation of question-specific memories. The pseudocode is provided in Appendix E. And theoretical analysis can be found at Appendix F.

## 3 EXPERIMENTS

### 3.1 EXPERIMENTAL SETUP

**Datasets** We use **PersonaMem** (Jiang et al., 2025), which includes multiple simulated user-LLM interaction histories over 7 real-world tasks, with memory lengths of 32K, 128K, and 1M tokens. Each history comprises up to 60 multi-turn sessions with evolving user personas and preferences. We also evaluate on **PersonaBench** (Tan et al., 2025a), a synthetic benchmark composed of private user documents and queries probing personal information (e.g., preferences, background), designed to assess the relevant personal memory retrieval ability before generation. And we include **Long-MemEval** (Wu et al., 2025a), a benchmark targeting long-term personalized retrieval, where factual questions require retrieving task-relevant information under both small and medium context settings. More details can be found at Appendix A. And **implementation details** can be found in Appendix B

**Metrics** On PersonaMem, performance is measured by *Accuracy* of generated responses, i.e., the proportion of responses that correctly align with the user's current persona and conversation context. On PersonaBench and LongMemEval, since the focus is on the retrieval of personal memory pieces, we use *Recall@K* to assess how well relevant personal memories are retrieved.

**Baselines** Unlike prior baselines that often rely on LLM-generated queries or external indexing strategies, our comparisons are restricted to retrieval-only methods to ensure fairness: all systems operate on the same memory vectors. Since our focus is on the retrieval component itself, we compare RF-Mem against four direct baselines: **(1) Zero Memory**: Following (Pan et al., 2025; Jiang et al., 2025), the model answers without using user memory. **(2) Full Context**: Following (Pan et al., 2025; Jiang et al., 2025), the entire user history memory is input to the model without retrieval. **(3) Dense Retrieval**: Following (Wu et al., 2025a; Pan et al., 2025; Zhong et al., 2024), a standard retriever that returns the top-$K$ memories based on similarity scores. This corresponds to the *Familiarity* retrieval in dual-process theory. **(4) Recollection** (ours): The recollection mode we proposed in this paper. This represents the system that only enters the *Recollection* path.

## 3.2 Overall performance in Personalized Generation

Table 1: Performance comparison over the PersonaMem across different memory corpus sizes. Columns are grouped by question type, rows by retrieval strategy. "NA" indicates the method does not need retrieval. "OOC" means out-of-context of the LLM input window. The best results are in **bold**, and the second-best results are underlined. "**\***" indicates the statistically significant improvements (i.e., two-sided t-test with $p < 0.05$) over the best baseline.

| Method | Retri Time | Avg. Tokens | Revisit Reasons | Track Evolution | Latest Prefs | Aligned Recs | New Scenarios | Shared Facts | New Ideas | Overall |
|---|---|---|---|---|---|---|---|---|---|---|
| **32K memory corpus data** | | | | | | | | | | |
| Zero Memory | NA | 464.6 | 0.7273 | 0.6259 | 0.1765 | 0.2182 | 0.2105 | 0.2326 | 0.1183 | 0.3854 |
| Full Context | NA | 24657.8 | 0.9394 | **0.7194** | **0.7647** | 0.7455 | 0.5614 | 0.5039 | 0.1828 | 0.6129 |
| Dense Retrieval | 3.14ms | 3515.9 | 0.9091 | 0.6475 | 0.6471 | 0.6364 | 0.5614 | 0.5426 | 0.2151 | 0.5908 |
| Recol. (ours) | 7.09ms | 3711.1 | **0.9495** | 0.6547 | 0.7059 | **0.7818** | 0.5965 | 0.5194 | **0.2688** | 0.6214 |
| RF-Mem (ours) | 5.09ms | 3566.6 | **0.9495** | 0.6619 | 0.7059 | **0.7818** | **0.6140** | 0.5659 | **0.2688** | **0.6350**\* |
| **128K memory corpus data** | | | | | | | | | | |
| Zero Memory | NA | 416.3 | 0.6766 | 0.6422 | 0.2136 | 0.2751 | 0.1925 | 0.2281 | 0.1737 | 0.3124 |
| Full Context | NA | 115601.4 | 0.5613 | 0.3930 | 0.2783 | 0.3868 | 0.2770 | 0.3977 | 0.1795 | 0.3231 |
| Dense Retrieval | 3.24ms | 3540.1 | 0.7881 | 0.6804 | 0.5346 | 0.5330 | 0.3662 | 0.6082 | 0.3069 | 0.5259 |
| Recol. (ours) | 7.86ms | 3680.3 | **0.8141** | 0.6716 | 0.5254 | 0.5301 | 0.3765 | 0.6140 | **0.3263** | 0.5288 |
| RF-Mem (ours) | 4.27ms | 3565.5 | 0.8030 | **0.6862** | **0.5427** | **0.5358** | **0.4131** | **0.6257** | **0.3263** | **0.5394**\* |
| **1M memory corpus data** | | | | | | | | | | |
| Zero Memory | NA | 415.1 | 0.6000 | 0.6178 | 0.1797 | 0.3179 | 0.1831 | 0.2569 | 0.1816 | 0.2730 |
| Full Context | NA | 912148.5 | OOC | OOC | OOC | OOC | OOC | OOC | OOC | OOC |
| Dense Retrieval | 4.42ms | 3816.1 | 0.7702 | **0.6933** | **0.4544** | 0.4464 | 0.3085 | 0.5903 | 0.3040 | 0.4518 |
| Recol. (ours) | 8.12ms | 3847.4 | 0.7532 | 0.6800 | 0.4440 | 0.4500 | **0.3593** | 0.5833 | 0.3136 | 0.4544 |
| RF-Mem (ours) | 6.28ms | 3827.8 | **0.7787** | 0.6889 | 0.4492 | **0.4536** | 0.3390 | **0.6111** | 0.3150 | **0.4589**\* |

To verify the effectiveness of RF-Mem, we evaluate it on **PersonaMem** across memory corpora of 32K, 128K, and 1M tokens per query. This setup stresses retrieval under different memory scales and question types, allowing us to examine how retrieval methods adapt as corpora grow larger and tasks become more complex. Table 1 compares zero-memory baselines, full-context input, and three retrieval strategies (Dense, Recollection, RF-Mem). We also report per-category accuracy under three corpus sizes in Appendix D.1 and different $K$ settings in Appendix D.5.

**First, RF-Mem delivers the best *overall* accuracy at every corpus scale while keeping inputs compact.** In Table 1, RF-Mem attains the top overall score at 32K (0.6350), 128K (0.5394), and 1M (0.4589). At 32K it surpasses Full Context by +0.0221 with only 3.6k average tokens versus 24.7k for Full Context, and with a modest 5.09ms retrieval time. As the memory grows, Full Context deteriorates sharply, reaching 0.3231 at 128K and becoming out of context at 1M, whereas RF-Mem remains stable and leads Dense Retrieval (Familiarity) by +0.0135 at 128K and +0.0071 at 1M. These trends validate that when a question is familiar, RF-Mem saves budget; when unfamiliar, it upgrades to structured recollection without committing to the cost of running it unconditionally.

**Second, RF-Mem leads on hybrid and transfer-style tasks with lower overhead.** At 32K it achieves top scores on *Aligned Recommendations* (0.7818), *New Scenarios* (0.6140), and *Shared*

*Facts* (0.5659) , while remaining close on *Track Evolution*. At 128K it remains ahead on *Track Evolution*, *Latest Prefs*, *Aligned Recs*, *New Scenarios*, and *Shared Facts*, tying on *New Ideas*. At 1M, where Full Context is infeasible, RF-Mem is strongest on *Revisit Reasons*, *Aligned Recs*, *Shared Facts*, and *New Ideas*, with only small gaps on *Track Evolution* and *Latest Prefs*, indicating that adaptive depth better balances precise anchoring and selective expansion than single-mode retrieval.

**Third, RF-Mem achieves a favorable accuracy–efficiency trade-off by regulating retrieval depth via entropy.** Compared to always-on recollection, RF-Mem improves overall accuracy while reducing latency: 5.09ms vs 7.09ms at 32K, 4.27ms vs 7.86ms at 128K, and 6.28ms vs 7.12ms at 1M, with similar token budgets to Dense Retrieval. This efficiency arises from treating familiarity as the default and route to the recollection path only when the question is unfamiliar. The result is a scalable retrieval controller that avoids the out-of-context cliff of Full Context, outperforms dense retrieval baselines as memory scales, and preserves recollection's advantages precisely.

## 3.3 OVERALL PERFORMANCE IN PERSONALIZED RETRIEVAL

Table 2: Performance comparison over the PersonaBench dataset across multiple question types. The best results are in **bold**, and the second-best results are underlined.

| Metrics | Recall@5 | | | | | | Recall@10 | | | | | |
|---|---|---|---|---|---|---|---|---|---|---|---|---|
| Method | Time | Basic Info | Social Info | Pref Easy | Pref Hard | Overall | Time | Basic Info | Social Info | Pref Easy | Pref Hard | Overall |
| *multi-qa-MiniLM-L6-cos-v1* | | | | | | | | | | | | |
| Famili. | 8.40ms | 0.4515 | 0.4852 | 0.4904 | 0.3659 | 0.4484 | 13.68ms | 0.5879 | 0.6220 | **0.6442** | 0.5561 | 0.5964 |
| Recol. | 9.65ms | 0.4379 | 0.4903 | **0.5128** | 0.3854 | 0.4491 | 17.29ms | **0.5924** | **0.6859** | 0.5659 | **0.6267** | 0.6062 |
| RF-Mem | 9.16ms | **0.4788** | **0.5091** | 0.4872 | **0.3854** | **0.4701** | 15.22ms | **0.5924** | 0.6799 | 0.5707 | **0.6267** | **0.6071** |
| *all-mpnet-base-v2* | | | | | | | | | | | | |
| Famili. | 7.64ms | 0.4242 | 0.2730 | 0.4487 | **0.4049** | 0.3887 | 10.55ms | 0.5409 | **0.4434** | **0.6795** | 0.5366 | 0.5333 |
| Recol. | 10.94ms | 0.4333 | **0.2918** | **0.4583** | 0.4000 | 0.3976 | 13.23ms | **0.6000** | 0.4365 | 0.6378 | 0.5220 | 0.5527 |
| RF-Mem | 8.33ms | **0.4515** | 0.2730 | 0.4487 | 0.4000 | **0.4009** | 10.55ms | 0.5955 | 0.4384 | 0.6378 | **0.5463** | **0.5553** |
| *BAAI/bge-base-en-v1.5* | | | | | | | | | | | | |
| Famili. | 8.92ms | 0.3970 | 0.3204 | **0.4583** | **0.3268** | 0.3738 | 10.19ms | 0.5121 | **0.4748** | **0.5673** | **0.4585** | 0.5002 |
| Recol. | 12.14ms | 0.3833 | **0.3619** | 0.4327 | 0.3171 | 0.3722 | 20.71ms | 0.5212 | **0.4748** | **0.5673** | **0.4585** | 0.5046 |
| RF-Mem | 10.14ms | **0.4015** | **0.3619** | 0.4487 | 0.3220 | **0.3836** | 18.13ms | **0.5303** | **0.4748** | **0.5673** | **0.4585** | **0.5089** |

Table 3: Performance comparison over the LongMemEval under small (S) and medium (M) memory versions. The best results are in **bold**, and the second-best results are underlined.

| Method | LongMemEval-S | | | | LongMemEval-M | | | |
|---|---|---|---|---|---|---|---|---|
| | Recall@5 | Recall@10 | Recall@50 | Time | Recall@5 | Recall@10 | Recall@50 | Time |
| *multi-qa-MiniLM-L6-cos-v1* | | | | | | | | |
| Fami. | 0.7136 | 0.8282 | 0.9761 | 24.91ms | 0.4177 | 0.5465 | 0.7518 | 27.72ms |
| Recol. | 0.7351 | 0.8425 | **1.0000** | 50.62ms | 0.4368 | 0.5585 | 0.7590 | 57.93ms |
| RF-Mem | **0.7375** | **0.8473** | **1.0000** | 39.58ms | **0.4391** | **0.5609** | **0.7613** | 41.22ms |
| *all-mpnet-base-v2* | | | | | | | | |
| Fami. | 0.7303 | 0.8353 | 0.9832 | 27.25ms | 0.4176 | 0.5489 | 0.7637 | 33.18ms |
| Recol. | **0.7398** | 0.8305 | **0.9952** | 51.79ms | 0.4386 | 0.5871 | 0.7422 | 62.11ms |
| RF-Mem | **0.7398** | **0.8377** | **0.9952** | 42.39ms | **0.4391** | **0.5894** | **0.7684** | 50.80ms |
| *BAAI/bge-base-en-v1.5* | | | | | | | | |
| Fami. | 0.7924 | 0.8926 | **1.0000** | 29.65ms | 0.4964 | 0.6611 | 0.8305 | 30.77ms |
| Recol. | 0.8162 | 0.9165 | **1.0000** | 43.65ms | 0.5131 | **0.6635** | 0.8234 | 58.05ms |
| RF-Mem | **0.8186** | **0.9189** | **1.0000** | 37.34ms | **0.5155** | **0.6635** | **0.8329** | 44.74ms |

To verify the retrieval performance of RF-Mem, we evaluate it against one-shot *Familiarity* and stepwise *Recollection* across both **PersonaBench** and **LongMemEval**. PersonaBench covers multi-domain user interactions, while LongMemEval stresses retrieval over extended memory corpora under small (S) and medium (M) settings. We report Recall@5, Recall@10, and Recall@50, together with average retrieval latency, under three retriever backbones (MiniLM, MPNet, BGE). This setup allows us to examine how retrieval strategies behave across tasks with varying difficulty and under different embedding models. We also conduct parameter sensitivity studies at D.2 to D.4.

**First, RF-Mem achieves the most balanced and robust performance across retrievers.** As shown in Table 2, RF-Mem either matches or surpasses the best baseline in overall Recall@5 and Recall@10, while avoiding the pitfalls of single-mode strategies. For example, under MiniLM it achieves an overall Recall@10 of 0.6071, slightly higher than Familiarity (0.5964) and Recollection (0.6062). On LongMemEval, in Table 3, RF-Mem further demonstrates stability. Its familiarity uncertainty-driven selection allows the retriever to exploit confident familiarity matches when possible, and to activate deeper recollection only when necessary, yielding consistently strong results.

**Second, the comparison between Familiarity and Recollection reveals complementary strengths.** Familiarity excels on fact-centric queries such as Basic Information and Preference Easy, where direct surface similarity suffices. *Recollection*, however, proves highly effective on context-heavy tasks such as Preference Hard and Social queries. On **PersonaBench**, in Table 2, Recollection reaches a Recall@10 of 0.6267 on Preference Hard under MiniLM, outperforming Familiarity at 0.5561. Similarly, on **LongMemEval**, it consistently lifts Recall@5 by more than 0.02 across multiple retrievers (e.g., 0.7351 vs. 0.7136 under MiniLM) in Table 3. Its iterative expansion uncovers deeper, temporally dispersed cues, making it a powerful strategy despite higher cost. These results highlight that Recollection is not merely slower, but offers indispensable diagnostic evidence, and that neither mode alone can achieve robustness across all task types.

**Third, RF-Mem delivers superior efficiency–effectiveness trade-offs.** On **PersonaBench** (Table 2), Familiarity is fastest (8–10ms) but shallow, while Recollection is stronger but nearly twice as slow (15–20ms). RF-Mem closes this gap, sustaining latency near Familiarity (9–15ms) with higher accuracy (e.g., Recall@10 of 0.6071 vs. 0.5964/0.6062). On **LongMemEval** (Table 3), Familiarity is low-latency (25–31ms) but loses coverage, while Recollection reaches perfect Recall@50 at much higher cost (40–62ms). RF-Mem balances both, keeping latency lower (37–50ms) while matching or exceeding accuracy.

## 3.4 ADAPTIVE EXPERIMENT

### 3.4.1 ADAPTIVE TO INDEX BUILDING METHOD

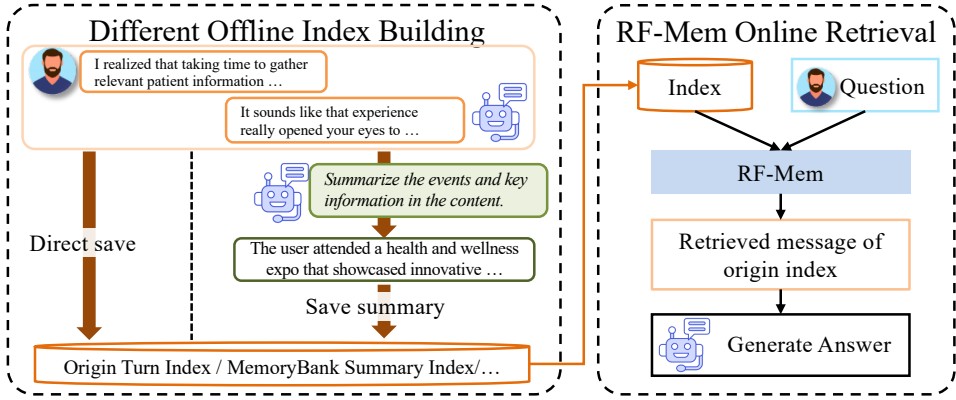

Figure 3: Illustration of adaptive study setup. Offline indexes (e.g., MemoryBank or origin memory) provide different storage, while RF-Mem serves as an online retrieval layer that adapts to them.

Table 4: Results by using MemoryBank summary index on PersonaMem (32K corpus).

| Method | Avg. Tokens | Revisit Reasons | Track Evolution | Latest Prefs | Aligned Recs | New Scenarios | Shared Facts | New Ideas | Overall |
|---|---|---|---|---|---|---|---|---|---|
| **MemoryBank summary index** | | | | | | | | | |
| Familiarity | 1267.6 | 0.7475 | 0.6187 | 0.5882 | 0.5818 | 0.3333 | 0.4341 | 0.1505 | 0.4941 |
| Recollection | 1441.8 | 0.8182 | 0.6259 | 0.5294 | 0.6909 | 0.4737 | 0.4031 | 0.1398 | 0.5212 |
| RF-Mem | 1421.8 | 0.8384 | 0.6259 | 0.5294 | 0.6545 | 0.4211 | 0.4419 | 0.1828 | **0.5314** |

To further examine the modularity of RF-Mem, we integrate it with external indexing schemes beyond raw turn-level memory. As shown in Figure 3, offline methods such as *MemoryBank* (Zhong et al., 2024) first summarize user dialog into indexes (e.g., turn-level summaries), while RF-Mem

operates as an online module during retrieval. This separation of offline indexing and online retrieval highlights that RF-Mem does not compete with summarization- or graph-based memory banks, but instead complements them by enabling human-like remembering. Table 4 highlights that RF-Mem demonstrates robustness across both settings: it achieves the highest overall accuracy under the turn-level index, and crucially, it narrows the performance drop under the summary index compared to single-path baselines. This adaptivity confirms that RF-Mem is modular and can be layered on top of heterogeneous memory indices, providing an uncertainty-aware dual-process retrieval mechanism that complements, rather than replaces, external indexing methods.

### 3.4.2 ADAPTIVE TO QUERY EXPANSION METHOD

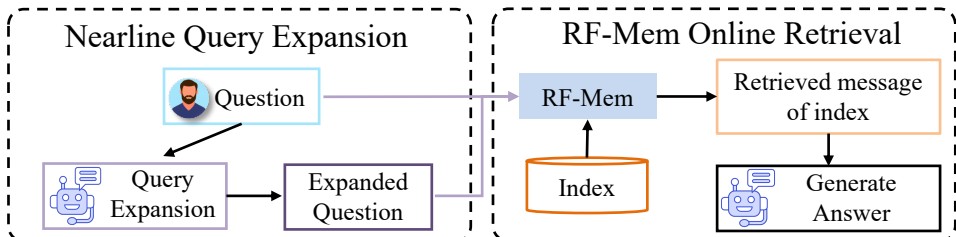

Figure 4: Illustration of the adaptive study setup. Nearline query expansion (e.g., HyDE) enriches the query representation, and RF-Mem operates as the online retrieval layer.

Table 5: Results by using HyDE query expansion method on PersonaBench.

| Metrics | Recall@5 | | | | | Recall@10 | | | | |
|---|---|---|---|---|---|---|---|---|---|---|
| HyDE | Basic Info | Social Info | Pref Easy | Pref Hard | Overall | Basic Info | Social Info | Pref Easy | Pref Hard | Overall |
| *multi-qa-MiniLM-L6-cos-v1* | | | | | | | | | | |
| Famili. | 0.3106 | 0.3909 | 0.4615 | 0.3122 | 0.3464 | 0.5000 | 0.4991 | 0.5737 | 0.5220 | 0.5120 |
| Recol. | 0.3015 | 0.4135 | 0.4615 | 0.3171 | 0.3482 | 0.4909 | 0.5028 | 0.5929 | 0.4878 | 0.5046 |
| RF-Mem | 0.3061 | 0.4135 | 0.4615 | 0.3171 | **0.3504** | 0.5091 | 0.5028 | 0.5929 | 0.5220 | **0.5194** |

To further assess the adaptability of RF-Mem, we combine nearline query expansion with online retrieval and examine their interaction on PersonaBench. As illustrated in Figure 4, the nearline expansion methods we adopt, HyDE (Gao et al., 2023), generate pseudo-relevance feedback that enriches the original query. RF-Mem then operates as an online retrieval layer. Table 5 reports the results when applying HyDE-based expansion. Across all categories, RF-Mem consistently matches or surpasses Familiarity baselines, demonstrating that its dual-process mechanism remains effective even when the upstream query representation shifts. These findings confirm that RF-Mem is modular and can be seamlessly integrated into nearline expansion pipelines.

### 3.4.3 ADAPTIVE TO ITERATIVE RAG METHOD

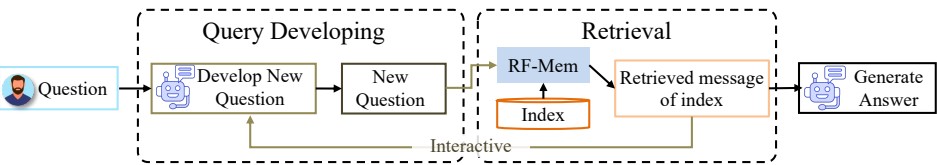

Figure 5: Illustration of adaptive study setup. Iterative RAG (e.g., Search-o1) provides a multi-turn retrieval for answer generation, while RF-Mem serves as the retrieval layer that adapts to it.

To further evaluate the adaptability of RF-Mem under iterative retrieval settings, we pair it with a multi-turn reasoning pipeline based on Search-o1 (Li et al., 2025d). As illustrated in Figure 5, Search-o1 develops refined follow-up queries through iterative question generation, while RF-Mem functions as the retrieval layer that reacts to these evolving queries in real time. Table 6 summarizes results using the Search-o1 interactive retrieval on the PersonaMem. RF-Mem still achieves the highest overall score. These findings demonstrate that RF-Mem retains its effectiveness in iterative RAG settings.

Table 6: Results by using Search-o1 iterative retrieval on PersonaMem (32K corpus).

| Search-o1 | Avg. Tokens | Revisit Reasons | Track Evolution | Latest Prefs | Aligned Recs | New Scenarios | Shared Facts | New Ideas | Overall |
|---|---|---|---|---|---|---|---|---|---|
| | | | | Search-o1 | | | | | |
| Familiarity | 4948.7 | 0.8687 | 0.6259 | 0.5882 | 0.6545 | 0.5614 | 0.5271 | 0.2581 | 0.5823 |
| Recollection | 5103.9 | 0.8990 | 0.6619 | 0.6471 | 0.7091 | 0.5789 | 0.4961 | 0.2796 | 0.6010 |
| RF-MEM | 5158.2 | 0.9293 | 0.6978 | 0.6471 | 0.7455 | 0.5789 | 0.5349 | 0.2043 | **0.6146** |

## 4 RELATED WORKS

Personalized memory retrieval for LLMs has emerged to complement the fixed context window and enable user-specific, context-aware responses. Unlike standard knowledge-based RAG that targets factual data, personal memory retrieval draws on a user's own history and preferences (Pan et al., 2025; Wu et al., 2025a; Xu et al., 2025c) (e.g., retrieving long-term user-AI dialogue context (Jiang et al., 2025; Maharana et al., 2024; Wu et al., 2025a) and user-user dialogue (Tan et al., 2025a)) to fill in missing details and tailor responses. Memory-augmented personalized LLM systems combine an LLM with a non-parametric memory to provide relevant background information. We review related works in three areas: (1) query reformulation, (2) index construction, and (3) retrieval frameworks.

**Query reformulation.** These methods expand or refine queries to improve memory retrieval. LD-Agent extracts keyphrases for retrieval (Li et al., 2025a), MemoCue proposes memory-inspired cue query (Zhao et al., 2025), and LQ-TOD generates task-oriented queries (Chen et al., 2025). Other approaches leverage LLM-based query expansion, such as LameR (Shen et al., 2024) and MemInsight (Salama et al., 2025), to enrich the query with contextual attributes.

**Index construction.** Prior work has also emphasized structuring user memory into searchable indices. Text-based methods summarize or cluster memories into compact representations, as in MemoryBank (Zhong et al., 2024), SeCom (Pan et al., 2025), and MemGas (Xu et al., 2025c), while others rely on reflective or hierarchical summaries (Tan et al., 2025b). Graph-based approaches instead capture relational structures, exemplified by A-Mem (Xu et al., 2025d), THEANINE (Ong et al., 2024), Mem0 (Chhikara et al., 2025), and Zep (Rasmussen et al., 2025). Although differing in representation, these methods share a common assumption: retrieval remains static. Most ultimately depend on standard dense retrievers to encode queries and memory keys, applying a uniform retrieval process regardless of uncertainty or task complexity.

**Retrieval frameworks.** From early keyword-based search (Robertson et al., 2009) to dense retrievers like DPR (Karpukhin et al., 2020) and Contriever (Lei et al., 2023), retrieval methods aim to rank memory items by semantic similarity. More advanced encoders such as MiniLM (Wang et al., 2020), MPNet (Song et al., 2020), and BGE (Luo et al., 2024) improve efficiency and accuracy, and are widely adopted in multi-session dialog retrieval (Xu et al., 2021). However, these frameworks largely adopt a single-process retrieval paradigm, treating all queries as homogeneous regardless of confidence or task complexity.

**However, existing methods overlook the dual-process nature of human memory retrieval.** Most prior works focus on query reformulation, index optimization, or stronger retrievers, yet they implicitly reduce retrieval to a one-shot recognition process (*Familiarity*). This not only neglects the crucial role of deliberate (*Recollection*), but also ignores the need for adaptive switching between the two paths. Our proposed RF-Mem addresses this gap by introducing a recollection retrieval path and a familiarity uncertainty-driven selection that adaptively alternates between fast Familiarity and deeper Recollection retrieval, thereby enabling more personalized memory retrieval.

## 5 CONCLUSION

In this work, we revisited personalized memory retrieval through the lens of dual-process theory. Existing methods are limited to one-shot retrieval based on similarity *Familiarity*, whereas we introduce *Recollection* as a deliberate stepwise retrieval mechanism. Building on this, we proposed **RF-Mem**, a familiarity uncertainty-driven framework that adaptively switches between them. Experiments show that RF-Mem achieves robust gains across both generation and retrieval tasks, and scales reliably to million-entry corpora, demonstrating the importance of integrating deliberate recollection into memory retrieval for personalizing LLM.

## ETHICS STATEMENT

This work focuses on improving personalized memory retrieval for large language models. Our study relies solely on simulated and publicly available benchmark datasets (PersonaMem, PersonaBench, LongMemEval), which do not contain personally identifiable information or sensitive data. We do not collect or release any private user data. While the proposed framework is designed to enhance efficiency and robustness in memory retrieval, potential misuse could arise if deployed without safeguards in sensitive applications such as healthcare or personal decision-making. We therefore emphasize that future deployment should follow strict data governance, privacy preservation, and fairness guidelines, and we encourage the research community to consider ethical implications when extending this work to real-world user data. We acknowledge that real-world deployments of preference-based systems may inadvertently over-amplify user traits or reinforce behavioral biases. Additionally, improper handling of user histories could expose sensitive information or lead to unintended profiling effects, underscoring the need for careful governance.

## REPRODUCIBILITY STATEMENT

We make every effort to ensure reproducibility. All datasets used in this study are publicly available, and we provide detailed references to their sources in the main text. Model architectures, hyperparameter choices ($B$, $F$, $\alpha$, $\tau$, and thresholds $\theta_{\text{high}}$, $\theta_{\text{low}}$) are explicitly documented in Appendix B. We also describe hyperparameter sensitivity analyses in Appendix D to illustrate robustness under different configurations. To further facilitate replication, we release code and scripts for reproducing all reported results in the supplementary materials. Our code repository is available at https://github.com/Applied-Machine-Learning-Lab/ICLR2026_RF-Mem.

## ACKNOWLEDGEMENT

This research was supported by the National Natural Science Foundation of China (NSFC) under Grants 72071029, 72231010, 62502404, and the Graduate Research Fund of the School of Economics and Management of Dalian University of Technology (No. DUTSEMDRFKO1). This research was partially supported by Hong Kong Research Grants Council (Research Impact Fund No.R1015-23, Collaborative Research Fund No.C1043-24GF, General Research Fund No.11218325), Institute of Digital Medicine of City University of Hong Kong (No.9229503), Huawei (Huawei Innovation Research Program).

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

# Technical Appendix

# A    DATASET DETAILS

**PersonaMem.**    We use **PersonaMem**[1] (Jiang et al., 2025), a large-scale personalized memory benchmark introduced at COLM 2025, as the core dataset for evaluating personalized retrieval and generation. PersonaMem is designed to evaluate memory-augmented LLMs by requiring them to retrieve user-specific histories and preferences from long-term dialogue traces. To stress-test retrieval under different scales, we follow the original paper and construct three memory corpora of increasing size: 32K, 128K, and 1M entries. Table 7 reports dataset statistics, showing that query length remains relatively stable across scales while memory size grows by orders of magnitude, creating a challenging retrieval environment. Table 8 further summarizes the task distribution, covering diverse domains such as family relations, study consultation, legal and medical consults, and recommendation tasks (movies, food, books, etc.). These heterogeneous task types capture both fact-oriented and reasoning-intensive scenarios, making PersonaMem a suitable benchmark for evaluating the balance between *Familiarity*-based fast recall and *Recollection*-driven contextual reconstruction.

Table 7: Dataset Statistics across Different Memory Corpora.

| Dataset name | 32k memory corpus | 128k memory corpus | 1M memory corpus |
|---|---|---|---|
| # of samples | 589 | 2727 | 2674 |
| Avg tokens of question | 464.6 | 416.3 | 415.1 |
| Avg tokens of memory | 24193.2 | 15185.1 | 911733.4 |
| # of Revisit Reasons | 99 | 269 | 235 |
| # of Track Evolution | 139 | 341 | 225 |
| # of Latest Prefs | 17 | 866 | 768 |
| # of Aligned Recs | 55 | 349 | 280 |
| # of New Scenarios | 57 | 213 | 295 |
| # of Shared Facts | 129 | 171 | 144 |
| # of New Ideas | 93 | 518 | 727 |

Table 8: Task Distribution across Different Memory Corpora

| Dataset name | 32k memory corpus | 128k memory corpus | 1M memory corpus |
|---|---|---|---|
| # of Home Decoration | 3 | 153 | 164 |
| # of Family Relations | 32 | 144 | 193 |
| # of Therapy | 2 | 249 | 161 |
| # of Travel Plan | 71 | 155 | 192 |
| # of Medical Consult | 19 | 195 | 163 |
| # of Legal Consult | 32 | 283 | 197 |
| # of Study Consult | 35 | 157 | 191 |
| # of Dating Consult | 94 | 197 | 208 |
| # of Financial Consult | 72 | 198 | 181 |
| # of Food Rec | 2 | 229 | 191 |
| # of Movie Rec | 104 | 158 | 212 |
| # of Music Rec | 56 | 44 | 128 |
| # of Book Rec | 67 | 172 | 139 |
| # of Sports Rec | – | 197 | 148 |
| # of Online Shopping | – | 196 | 206 |

**PersonaBench.**    We further evaluate on the **PersonaBench**[2] dataset, which benchmarks personalized retrieval in multi-source user environments introduced in ACL 2025 (Tan et al., 2025a). As shown in Table 9, PersonaBench aggregates heterogeneous user histories across six users, including *conversations with friends*, *user-AI interactions*, and *e-commerce purchase records*. Each user contributes on average 44 queries grounded in a corpus of about 88 items, yielding rich signals of both

---

[1] https://github.com/bowen-upenn/PersonaMem
[2] https://github.com/SalesforceAIResearch/personabench

social and transactional behavior. The resulting setting allows us to test retrieval robustness across diverse memory types, from casual dialogue to structured purchase history. This design contrasts with PersonaMem, which primarily focuses on long user-AI dialogues, and complements our study by introducing multi-modal user traces that better capture the breadth of personalization scenarios.

Table 9: Statistics of the PersonaBench dataset across users.

| User Id | # of Queries | # of Corpus | # of Conversations with friends | # of User-AI Conversation | # of User e-comerce purchase histories |
|---|---|---|---|---|---|
| 1 | 48 | 110 | 84 | 23 | 3 |
| 2 | 43 | 90 | 78 | 8 | 4 |
| 3 | 42 | 64 | 51 | 12 | 1 |
| 4 | 46 | 85 | 71 | 14 | 0 |
| 5 | 44 | 84 | 59 | 21 | 4 |
| 6 | 40 | 94 | 79 | 14 | 1 |
| **Sum** | 263 | 527 | 422 | 92 | 13 |
| **Avg** | **43.83** | **87.83** | **70.33** | **15.33** | **2.17** |

**LongMemEval.** We further evaluate on the **LongMemEval**[3] dataset, which benchmarks the ability to retrieve task-relevant information from extended user-specific corpora. introduced in ICLR 2025 (Wu et al., 2025a). Each instance in the dataset consists of a factual question paired with a synthetic memory corpus simulating historical user data. The dataset includes two settings: *LongMemEval-s*, where each question is associated with approximately 50 memories, and *LongMemEval-m*, where each corpus contains over 500 memories. In total, both settings consist of 500 questions, allowing evaluation of memory retrieval precision under varying context lengths. This benchmark enables controlled analysis of personalized retrieval performance under realistic long-context scenarios.

Table 10: Statistics of the LongMemEval dataset, with different sizes of associated memory corpora.

| Statistic | LongMemEval-s | LongMemEval-m |
|---|---|---|
| Total Questions | 500 | 500 |
| Total Session-level Memory | 25,112 | 250,948 |
| Min Session-level Memory per Question | 39 | 501 |
| Max Session-level Memory per Question | 66 | 506 |
| Avg Session-level Memory per Question | **50.22** | **501.90** |

## B  IMPLEMENTATION DETAILS

All experiments are conducted on a single NVIDIA A100 GPU with Ubuntu OS, where the GPU is exclusively allocated to the process when measuring runtime.

For **PersonaMem**, we follow the original setup and build the memory corpus at the dialogue-turn level, where each chunk corresponds to a user query and a single LLM response. We use `GPT-4.1-mini` as the generator and `multi-qa-MiniLM-L6-cos-v1` as the retriever. The hyperparameters are set as $\lambda = 20$, $B = 3$, $F = 2$, with thresholds $\theta_{high} = 0.6$ and $\theta_{low} = 0.3$, ensuring stable regulation across turn-level retrieval. And prompt for generation can be found at Appendix C. And we illustrate the mean score $\bar{s}$ and entropy $H(p)$ of the PersonaMem dataset in Figure 6. The cumulative distributions reveal a consistent pattern across corpus sizes: as the scale increases from 32K to 1M, the mean score distribution shifts slightly toward lower values, reflecting weaker overall familiarity in larger search spaces, whereas entropy remains concentrated within a narrow band (0.1-0.2), indicating stable uncertainty resolution. The annotated quartiles further confirm that the median values of both $\bar{s}$ ($\approx$0.50-0.55) and $H(p)$ ($\approx$0.17-0.18) remain largely invariant, which demonstrates that the familiarity signal preserves a stable operating range across different scales. Such stability provides empirical support for RF-Mem's threshold design, ensuring that

---

[3] https://github.com/xiaowu0162/LongMemEval

the switching mechanism can generalize without costly re-tuning and maintaining robustness under varying corpus sizes. For support our theorical assumption in Appendix F.3, we show the empirical distribution of mean score $\bar{s}$ in the Figure 7. All three datasets exhibit light-tailed, bounded distributions without heavy-tail behavior, and the tail shape remains stable as the corpus size grows. This empirically confirms that the similarity landscape does not display the heavy-tailed structure.

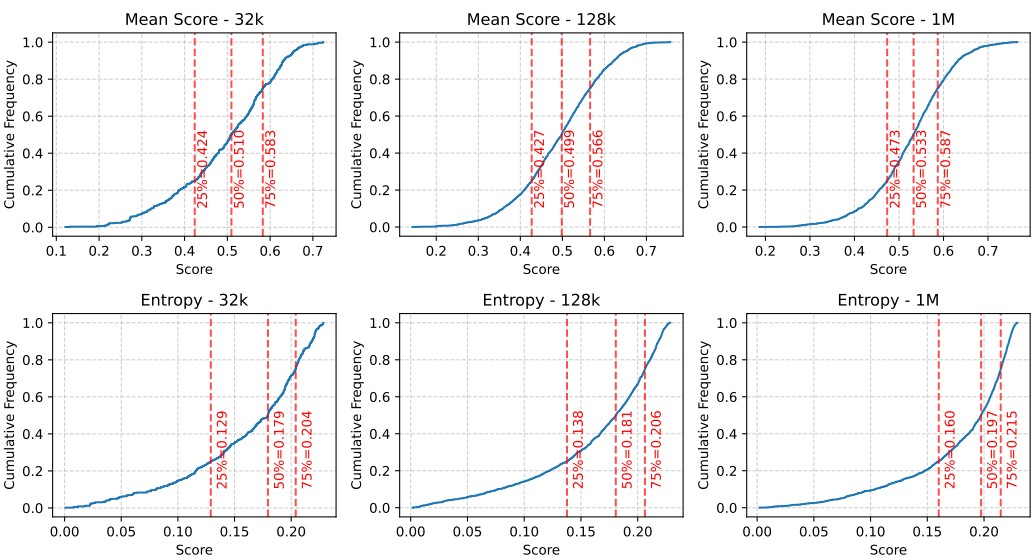

Figure 6: Mean score $\bar{s}$ and entropy $H(p)$ of the PersonaMem dataset.

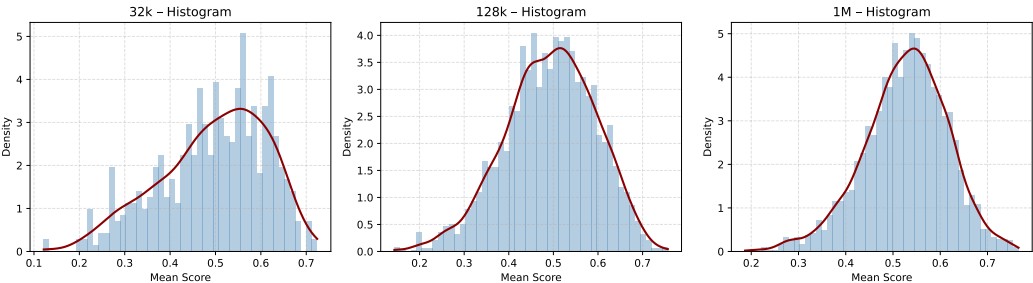

Figure 7: Empirical distributions of mean score $\bar{s}$ of the PersonaMem dataset.

For **PersonaBench**, we adopt the session-level memory construction as in the benchmark paper, which enables fair evaluation of retrieval quality using Recall. We evaluate our method under multiple retrievers, including `multi-qa-MiniLM-L6-cos-v1`[4], `all-MiniLM-L6-v2`[5], and `bge-base-en-v1.5`[6] to ensure generality. For Recall@5, we use $B = 3$, $F = 1$; for Recall@10, we use $B = 3$, $F = 2$, and $\lambda = 30$ for both. The thresholds are set to $\theta_{\text{high}} = 0.6$ and $\theta_{\text{low}} = 0.0$, reflecting the lower similarity scores in session-level indices. And we illustrate the mean score $\bar{s}$ and entropy $H(p)$ of the PersonaMem dataset in Figure 8. Also, for support our theorical assumption in Appendix F.3, we show the empirical distribution of mean score $\bar{s}$ in the Figure 9.

For **LongMemEval**, we also adopt the session-level memory construction as in the paper, with recall as a metric. We also evaluate RF-Mem under `multi-qa-MiniLM-L6-cos-v1`, `all-MiniLM-L6-v2`, and `bge-base-en-v1.5` to ensure generality. For LongMemEval-S and LongMemEval-M, we use $B = 4$, $F = 1$, and $\lambda = 20$. The thresholds are set to $\theta_{\text{high}} = 0.6$ and $\theta_{\text{low}} = 0.0$. And we illustrate the mean score $\bar{s}$ and entropy $H(p)$ of the two datasets in Figure 10 and Figure 12. The empirical distribution of mean score $\bar{s}$ shown in Figure 11 and Figure 13 also support our theorical assumption in Appendix F.3.

---

[4] https://huggingface.co/sentence-transformers/multi-qa-MiniLM-L6-cos-v1
[5] https://huggingface.co/sentence-transformers/all-MiniLM-L6-v2
[6] https://huggingface.co/BAAI/bge-base-en-v1.5

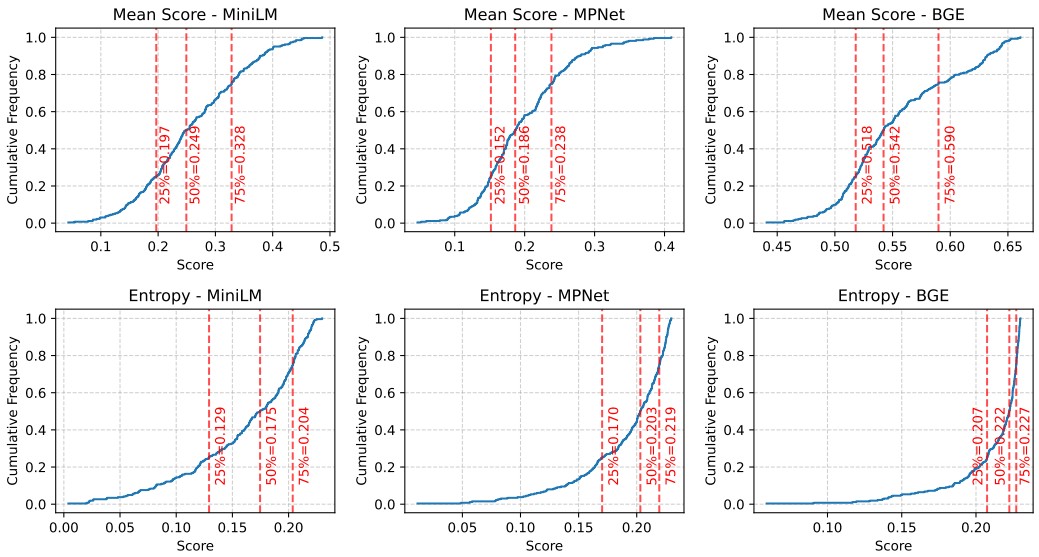

Figure 8: Mean score $\bar{s}$ and entropy $H(p)$ of the PersonaBench dataset.

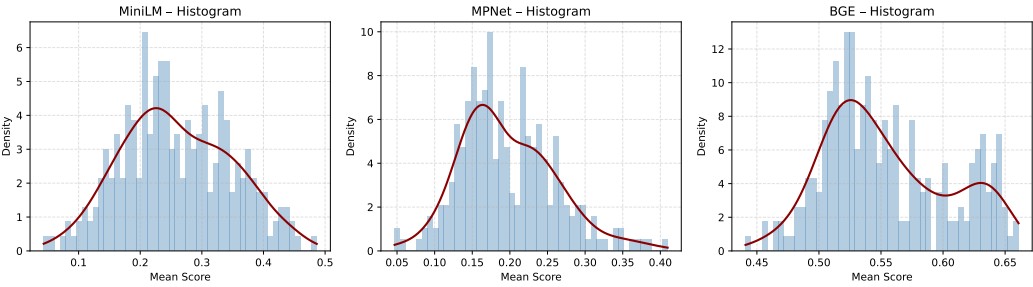

Figure 9: Empirical distributions of mean score $\bar{s}$ of the Personabench dataset.

## C  PROMPT FOR PERSONAMEM

For the multiple-choice evaluation of PerosnaMem dataset, we adopted a strict prompt template to ensure consistency and avoid ambiguous model outputs. The instruction explicitly restricts the model to return exactly one option among (a), (b), (c), or (d), without any reasoning or additional text. This design prevents uncontrolled generation and makes results directly comparable across models.

---

**Prompt Instruction**

You are a multiple-choice answer generator. You MUST respond with exactly one option in the form of (a), (b), (c), or (d). Do not include any explanation, reasoning, or extra text. Do not output anything else besides the chosen option. If the correct answer is unknown, make the best guess and still only respond in that format. If your output does not exactly match one of (a), (b), (c), or (d), your answer will be considered incorrect.

---

During evaluation, the prompt is constructed by concatenating the question, the fixed instruction above, and all candidate options. Retrieved memory context (from RF-Mem or baselines) is prepended as dialogue history to provide user-specific background. This structure guarantees deterministic outputs while isolating the effect of retrieval on answer quality.

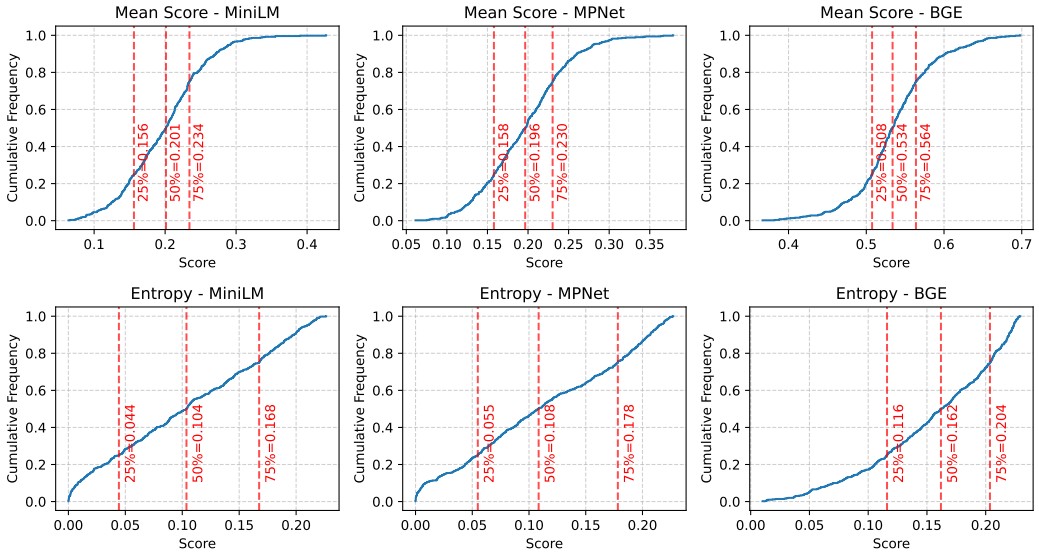

Figure 10: Mean score $\bar{s}$ and entropy $H(p)$ in the LongMemEval-S dataset.

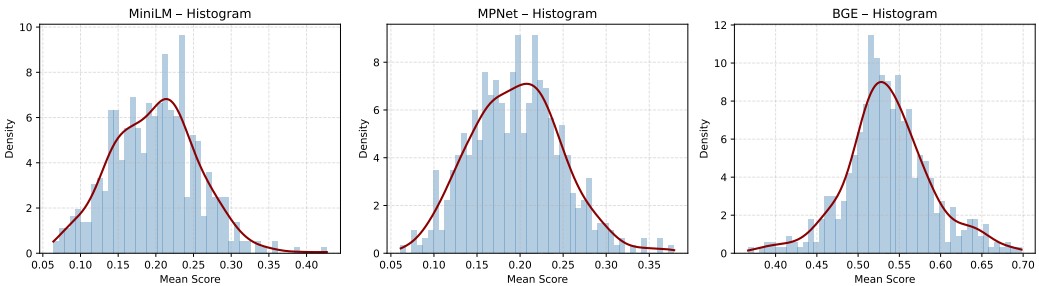

Figure 11: Empirical distributions of mean score $\bar{s}$ of the LongMemEval-S dataset.

**Example Prompt**

**Question:** Can you suggest some new evidence-based practices to explore for healthcare decision-making?

**Instruction:** You are a multiple-choice answer generator. You MUST respond with exactly one option in the form of (a), (b), (c), or (d). Do not include any explanation, reasoning, or extra text. Do not output anything else besides the chosen option. If the correct answer is unknown, make the best guess and still only respond in that format. If your output does not exactly match one of (a), (b), (c), or (d), your answer will be considered incorrect.

**Options:**

    (a) Exploring patient-centered communication strategies might be a great new evidence-based practice to consider...

    (b) One practice that could be impactful is the integration of artificial intelligence in predictive analytics for healthcare decision-making...

    (c) You might consider exploring systematic reviews and meta-analyses of clinical trials, as these often provide robust evidence for healthcare practices...

    (d) Implementing shared decision-making models in clinical practice is a promising evidence-based strategy to explore...

**Correct Answer:** (c) # This is the correct answer and not in the prompt.

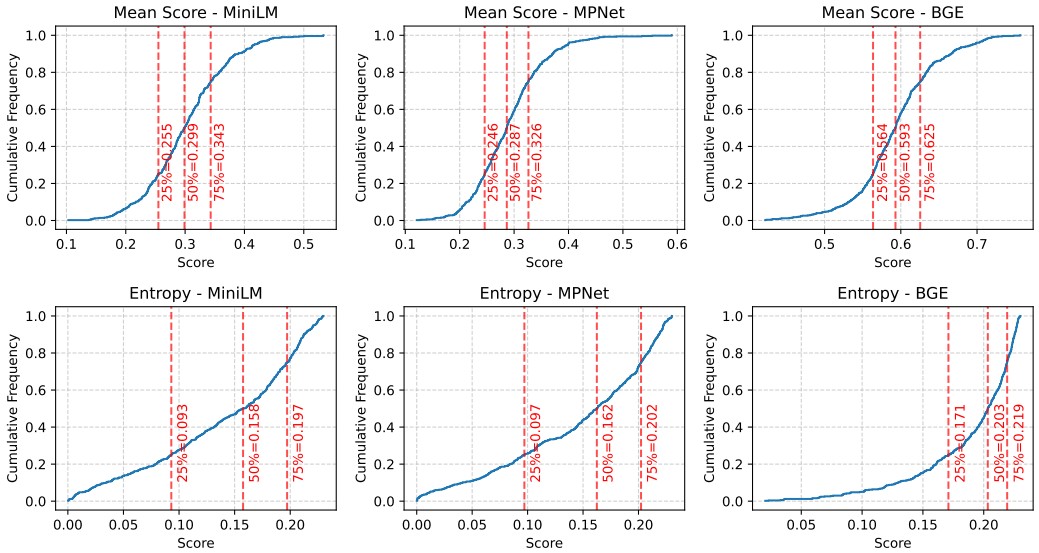

Figure 12: Mean score $\bar{s}$ and entropy $H(p)$ in the LongMemEval-M dataset.

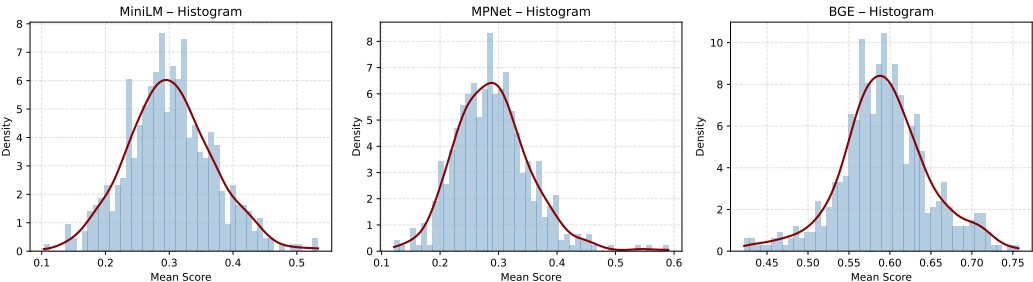

Figure 13: Empirical distributions of mean score $\bar{s}$ of the LongMemEval-M dataset.

# D  ADDITIONAL EXPERIMENT

## D.1  RESULT ACROSS CATEGORY IN PERSONAMEM

To further examine retrieval behaviors across different task types, we conduct a category-level analysis on **PersonaMem** (Table 11). This evaluation spans factual queries (e.g., *food recommendation*, *medical consult*), reasoning-intensive domains (e.g., *family relations*, *therapy*), and hybrid scenarios (e.g., *aligned recommendations*, *new scenarios*), allowing us to disentangle how each retrieval strategy responds to varying demands of factual precision, contextual reasoning, and personalization.

**First, Familiarity excels in direct recall but struggles as complexity grows.** As shown in Table 11, dense retrieval achieves peak scores on fact-centric categories where surface similarity is sufficient. For instance, it reaches perfect accuracy on *food recommendation* (1.0000 at 32k corpus) and strong results in *movie recommendation* (0.5759 at 128k). These cases confirm its cognitive role as a fast, coarse recognition process. However, as tasks demand contextual integration—such as *track evolution* or *aligned recommendations*—Familiarity shows clear degradation, particularly when scaling to 1M memories.

**Second, Recollection proves valuable for reasoning-intensive tasks.** In categories like *family relations*, *therapy*, and *dating consultation*, Recollection consistently surpasses Familiarity by reconstructing dispersed cues across sessions (e.g., 0.5938 vs. 0.5625 in *family relations* at 32k, and 0.5280 vs. 0.4534 in *therapy* at 1M). Yet, on factual categories such as *food recommendation* or *study consultation*, where surface matches are already diagnostic, its advantage diminishes or even reverses. This confirms its role as a slower but more diagnostic mode.

**Third, RF-Mem achieves robust gains through adaptive switching.** As shown in Table 11, RF-Mem consistently outperforms single-mode baselines by combining the efficiency of Familiarity with the diagnostic depth of Recollection. For instance, it improves *legal consultation* at 32k (0.94 vs. 0.78/0.88) and sustains advantages in *family relations* at 1M (0.60 vs. 0.57/0.58). These examples illustrate its ability to maintain high accuracy across both factual and reasoning categories. Importantly, RF-Mem also delivers the best overall results at all corpus scales, confirming that uncertainty-aware routing enables robust retrieval without committing to a single mode.

Table 11: Performance comparison on different memory corpus sizes.

| Question Category | 32k memory corpus | | | 128k memory corpus | | | 1M memory corpus | | |
|---|---|---|---|---|---|---|---|---|---|
| | Famili. | Recol. | RF-Mem | Famili. | Recol. | RF-Mem | Famili. | Recol. | RF-Mem |
| **Overall** | 0.5908 | 0.6214 | **0.6350** | 0.5259 | 0.5288 | **0.5394** | 0.4518 | 0.4544 | **0.4589** |
| Home Decoration | **0.6667** | **0.6667** | **0.6667** | 0.6275 | 0.6275 | **0.6471** | 0.4207 | **0.4451** | 0.4207 |
| Family Relations | 0.5625 | **0.5938** | **0.5938** | 0.5486 | 0.5625 | **0.5764** | 0.5734 | 0.5803 | **0.6010** |
| Therapy | **0.5000** | **0.5000** | **0.5000** | 0.5060 | 0.5301 | **0.5341** | 0.4534 | **0.5280** | 0.4845 |
| Travel Plan | 0.4789 | **0.5775** | 0.5634 | 0.5806 | 0.6194 | **0.6452** | **0.4948** | 0.4792 | 0.4896 |
| Medical Consult | **0.8421** | 0.7895 | **0.8421** | 0.4718 | **0.5128** | 0.5026 | 0.4294 | 0.4110 | **0.4356** |
| Legal Consult | 0.7812 | 0.8750 | **0.9375** | **0.5124** | 0.4735 | 0.4841 | **0.4061** | 0.3909 | 0.3909 |
| Study Consult | **0.6000** | **0.6000** | **0.6000** | **0.5096** | 0.4904 | 0.4773 | 0.4503 | 0.4293 | **0.4817** |
| Dating Consult | 0.5851 | 0.5851 | **0.6277** | 0.4772 | **0.5025** | 0.4975 | **0.5192** | 0.5000 | 0.5096 |
| Financial Consult | 0.5556 | 0.5972 | **0.6389** | **0.4899** | 0.4697 | **0.4899** | **0.4365** | 0.4309 | 0.4199 |
| Food Rec | **1.0000** | **1.0000** | **1.0000** | 0.5240 | **0.5721** | 0.5633 | 0.3717 | **0.4241** | 0.3874 |
| Movie Rec | **0.6154** | **0.6154** | 0.5865 | **0.5759** | 0.5696 | 0.5633 | 0.4434 | 0.4481 | **0.4575** |
| Music Rec | 0.6071 | **0.6250** | **0.6250** | 0.4091 | 0.4773 | **0.5096** | 0.4609 | 0.4688 | **0.5000** |
| Book Rec | 0.5373 | 0.5970 | **0.6418** | 0.5640 | 0.5407 | **0.5814** | 0.4964 | 0.4532 | **0.5036** |
| Sports Rec | - | - | - | **0.5228** | 0.4670 | 0.4975 | 0.4189 | **0.4392** | **0.4392** |
| Online Shopping | - | - | - | 0.5408 | 0.5459 | **0.5561** | 0.3738 | **0.3932** | 0.3786 |

## D.2 SENSITIVITY ANALYSIS OF $\alpha$ AND $\tau$

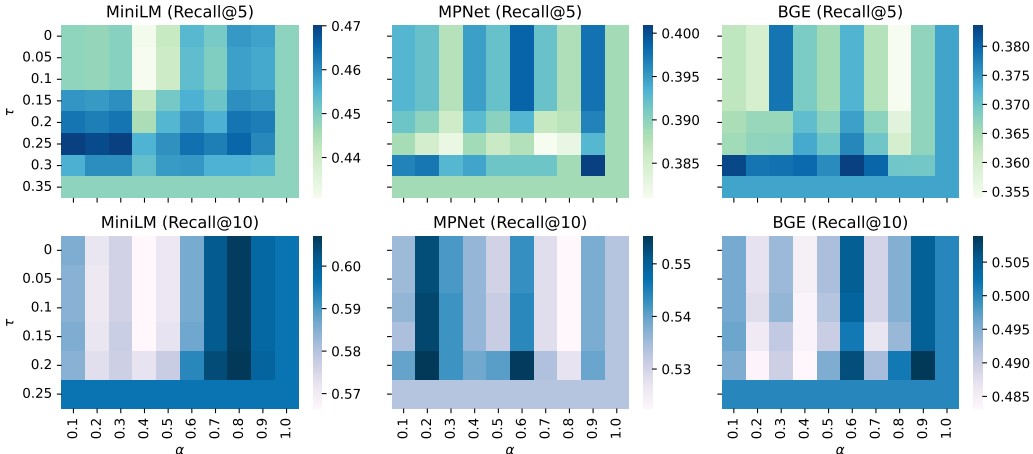

Figure 14: Hyperparameter sensitivity study on PersonaBench. Heatmaps report Recall@5 and Recall@10 across different retrievers, varying $\alpha$ (query–centroid mixing) and $\tau$ (entropy threshold).

To examine the robustness of RF-Mem, we conduct a systematic study varying two key hyperparameters, $\alpha$ and $\tau$, across different retrievers (MiniLM, MPNet, BGE) and evaluation metrics (Recall@5/10). Figure 14 visualizes the results as heatmaps, where warmer colors indicate higher retrieval performance. This setup allows us to directly assess how query mixing ($\alpha$) and entropy thresholding ($\tau$) interact to balance efficiency and coverage.

**First, the query–centroid mixing coefficient $\alpha$ controls how strongly recollection expands beyond the original probe.** In Recall@5, moderate $\alpha$ (around 0.3–0.6) yields the best results, such

as MiniLM improving overall recall to ∼0.47, while extreme values ($\alpha = 0.0$ or $\alpha = 1.0$) reduce stability. For Recall@10, performance is less sensitive, but mid-range $\alpha$ still avoids degradation observed at the boundaries. This indicates that balanced mixing is crucial: too little restricts recollection, too much overwhelms with noisy expansions.

**Second, the entropy threshold $\tau$ regulates the switch between Familiarity and Recollection.** Low $\tau$ values push the system toward frequent recollection, leading to higher gains in complex categories but also exposing variance. For instance, MiniLM Recall@10 achieves its highest values (∼0.60) when $\tau$ is set around 0.2–0.25, while overly high thresholds ($\tau \geq 0.35$) reduce adaptivity, as retrieval defaults prematurely to Familiarity. The results highlight that moderate entropy gating achieves the best trade-off between efficiency and contextual depth.

### D.3 SENSITIVITY ANALYSIS OF $B$ AND $F$

Figure 15 reports the effect of beam width $B$ and fanout $F$ on retrieval performance in LongMemEval-S and LongMemEval-M, under the retrievaler `multi-qa-MiniLM-L6-cos-v1` of recollection retrieval. We observe three consistent patterns.

**First, increasing $F$ generally reduces recall@5.** As shown in the top-left and bottom-left plots, recall@5 steadily drops when $F$ grows, since higher fanout expands the search too widely, diluting precision in top-ranked results. For example, in LongMemEval-S, recall@5 decreases from 0.72 ($F = 1$) to below 0.60 when $F = 4$.

**Second, recall@10 is more stable under moderate $F$, but declines when $B$ or $F$ are too large.** The middle plots show that recall@10 peaks around $F = 1$ or $F = 2$ (e.g., 0.83 in LongMemEval-S, 0.56 in LongMemEval-M), but drops once $F = 3$ or $F = 4$, reflecting over-expansion and redundancy. This suggests that moderate fanout provides useful diversification, while excessive expansion introduces noise.

**Third, recall@50 is robust and even improves with higher $F$.** As shown in the rightmost plots, recall@50 remains very high in LongMemEval-S (above 0.98 across all settings), and increases slightly in LongMemEval-M (up to 0.76 when $B = 3, F = 2$). This indicates that large fanout helps cover more relevant memories at longer retrieval depths, consistent with recollection's role of broad exploration.

Overall, these results demonstrate that small beam width and low fanout ($B = 2$ or 3, $F = 1$ or 2) offer the best balance between early precision and broad coverage, aligning with the intuition of controlled, stepwise recollection.

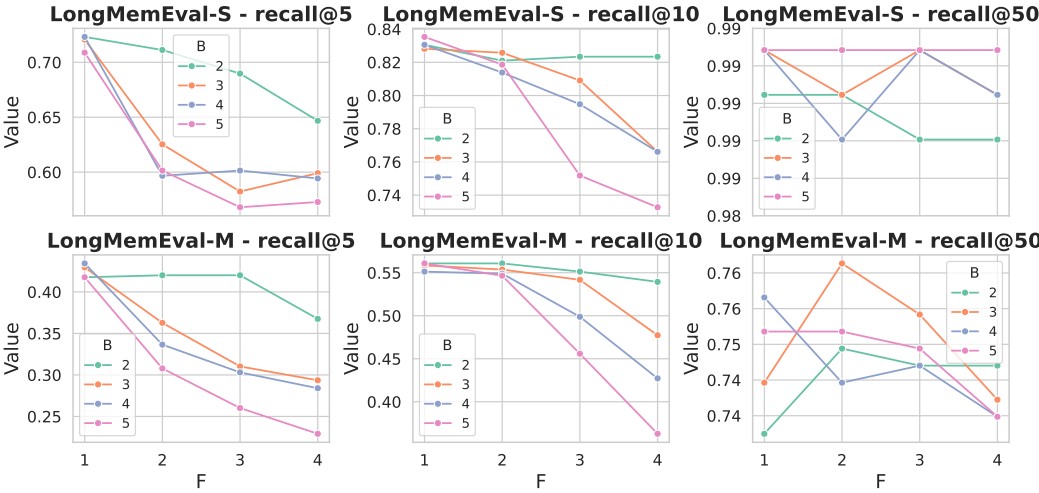

Figure 15: Hyperparameter $B$ and $F$ sensitivity of Recollection retrieval on LongMemEval-S and LongMemEval-M.

## D.4 IMPACT OF $\alpha$ UNDER VARYING RETRIEVAL SIZE $K$

**Effect of $\alpha$ on Retrieval.** To investigate the role of the mixing weight $\alpha$, we conduct a sensitivity study on the LongMemEval-M dataset under the **Recollection** retrieval, measuring Recall@5, Recall@10, and Recall@50. As shown in Figure 16, the impact of $\alpha$ exhibits a clear dependence on retrieval depth. For Recall@5, which emphasizes short-hop retrieval precision, performance decreases steadily as $\alpha$ grows, suggesting that placing excessive weight on the original query suppresses exploratory expansion and thereby reduces the system's ability to capture nearby but diverse evidence. In contrast, Recall@50 improves markedly with larger $\alpha$, indicating that stronger query–centroid mixing facilitates broader exploration and enables the retriever to recover more distant but relevant memories over long retrieval chains. Recall@10 demonstrates an intermediate pattern, peaking around $\alpha = 0.3$–$0.5$ before declining, which reflects the delicate balance between preserving the specificity of the original query and promoting contextual expansion through recollection.

These results highlight a fundamental trade-off: smaller values of $\alpha$ favor precision in short-path retrieval, while larger values enhance coverage in long-path retrieval. The presence of an intermediate optimum for Recall@10 further confirms that no single setting universally dominates across depths, underscoring the need to calibrate $\alpha$ according to application demands. More broadly, this sensitivity analysis validates the design choice in RF-Mem to treat $\alpha$ as a tunable parameter, enabling the framework to flexibly adjust the balance between efficiency and breadth when navigating different levels of the memory space.

Figure 16: Effect of $\alpha$ on short- vs. long-path retrieval performance (Recall@$K$).

## D.5 SENSITIVITY ANALYSIS OF RETRIEVAL SIZE $K$

To further examine the sensitivity of RF-Mem to the probe retrieval parameter $K$, we vary $K$ from 5 to 50 and report the resulting accuracy under different corpus sizes (32K, 128K, and 1M) in PersonaMem. As shown in Figure 17, the overall performance of RF-Mem remains consistently stable across a wide range of $K$, demonstrating the robustness of our familiarity-based regulation. In contrast, the baselines exhibit stronger fluctuations: for Familiarity, accuracy quickly rises with $K$ but plateaus once $K \geq 10$, while Recollection benefits from larger $K$ due to finer entropy resolution, yet its gains taper off and even slightly degrade after $K = 20$.

RF-Mem combines the advantages of both pathways, achieving higher accuracy than either baseline across all corpus scales, and crucially avoids the over-expansion problem of Recollection by adaptively invoking recollection only when necessary. Notably, the performance curves at 128K and 1M confirm that the improvement saturates beyond moderate probe sizes, suggesting that small values of $K$ (e.g., 10–20) are already sufficient for effective familiarity estimation and adaptive switching. These results highlight that RF-Mem does not depend on finely tuned probe parameters, making it both robust and practical for deployment in large-scale personalized memory retrieval scenarios.

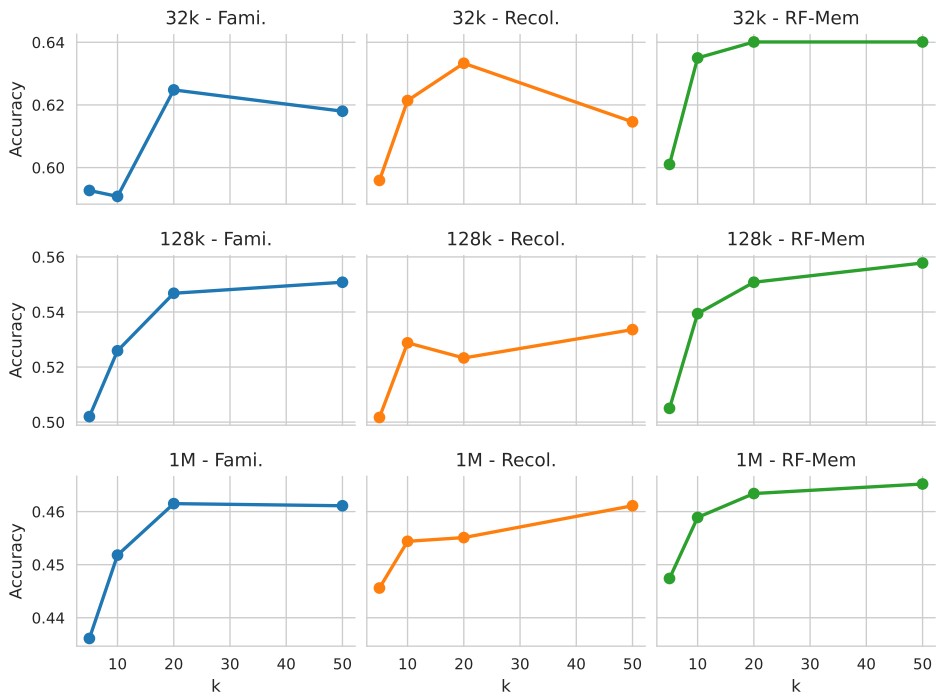

Figure 17: Effect of probe size $K$ on retrieval performance across different corpus scales.

### D.6 LEARNING THE STRATEGY SELECTION MECHANISM

Although the gating thresholds in RF-Mem are manually specified, it is important to assess whether the routing policy can also be learned from data. To this end, we construct a binary classification task in which each memory instance is represented by the two gating signals, namely the mean similarity $\bar{s}$ of the probe retrieval and the entropy $H(p)$ over the top-$K$ similarity distribution. The classifier predicts whether the familiarity path or the recollection path yields higher retrieval quality. This setting allows us to examine (i) how much supervision is needed to approximate the decision boundary and (ii) whether the two handcrafted signals contain sufficient information for reliable automatic routing.

**Training data.** We utilize two subsets of PersonaMem: the 32k corpus (589 samples) and the 128k corpus (2727 samples). For 32k, the first 400 samples are used for training and the remaining 189 for evaluation; for 128k, the first 2000 samples are allocated for training and the remaining 727 for evaluation. Only cases in which one path clearly outperforms the other are retained, forming a clean binary-labeled dataset. Table 12 summarizes the distribution of positive and negative labels.

**Training Details.** A three-layer MLP is trained as a binary classifier, with hidden dimensions of 32 and 16, and a final sigmoid output unit. ReLU activations are used in the hidden layers. Training follows an 80/20 train–validation split.

Table 12: Training data statistics for the learned gating classifier.

| Dataset | Fami. win | Recol. win | Tie | Training Samples |
|---|---|---|---|---|
| PersonaMem-32k | 34 | 41 | 325 | 75 (34+41) |
| PersonaMem-128k | 178 | 185 | 1637 | 363 (178+185) |

**Experimental findings.** Tables 13 and 14 report detailed performance across evaluation categories. We can find **while the hand-tuned thresholds remain slightly superior, the learned gate**

**shows clear improvements when trained on more data, demonstrating its potential applicability in real-world scenarios where larger-scale logs are accessible.** On the smaller 32k set, the learned gate shows high variance across runs and does not consistently outperform simple heuristics, indicating that limited data makes the decision boundary difficult to estimate reliably. When moving to the larger 128k set, the learned gate becomes considerably more stable and approaches the performance of the manually tuned mechanism. These results suggest that the two handcrafted signals already capture a meaningful structural separation between the two retrieval modes, and that automatic tuning becomes increasingly effective as more user data is available.

Table 13: Results of the learned strategy selector on PersonaMem-32k (189 test samples).

| Method | Overall | Revisit Reasons | Shared Facts | Track Evolu-tion | Aligned Recs | Latest Prefs | New Scenar-ios | New Ideas | Num. of Famili. | Num. of Recol. |
|---|---|---|---|---|---|---|---|---|---|---|
| Dense | 0.6296 | 0.9688 | 0.7297 | 0.6327 | 0.8125 | 0.2857 | 0.5385 | 0.2286 | 189 | 0 |
| Recol. | 0.6667 | 0.9688 | 0.7297 | 0.7551 | 0.8750 | 0.1429 | 0.7297 | 0.2286 | 0 | 189 |
| RF-Mem | **0.6720** | 1.0000 | 0.7838 | 0.6939 | 0.8750 | 0.1429 | 0.6154 | 0.2571 | 83 | 106 |
| RF-Mem (Learned) | 0.6482 ±0.0067 | 0.9688 ±0.0000 | 0.7514 ±0.0114 | 0.6878 ±0.0255 | 0.8750 ±0.0000 | 0.1429 ±0.0000 | 0.5385 ±0.0000 | 0.2286 ±0.0000 | 97.5 ±6.70 | 91.5 ±6.70 |

Table 14: Results of the learned strategy selector on PersonaMem-128k (727 test samples).

| Method | Overall | Revisit Reasons | Shared Facts | Track Evolu-tion | Aligned Recs | Latest Prefs | New Scenar-ios | New Ideas | Num. of Famili. | Num. of Recol. |
|---|---|---|---|---|---|---|---|---|---|---|
| Dense | 0.5131 | 0.7742 | 0.6410 | 0.6162 | 0.5176 | 0.5122 | 0.3529 | 0.3043 | 727 | 0 |
| Recol. | 0.5158 | 0.7634 | 0.5897 | 0.6162 | 0.5647 | 0.5366 | 0.3824 | 0.2609 | 0 | 727 |
| RF-Mem | **0.5199** | 0.7527 | 0.6410 | 0.6162 | 0.5059 | 0.5463 | 0.3824 | 0.2971 | 402 | 325 |
| RF-Mem (Learned) | 0.5175 ±0.0039 | 0.7591 ±0.0055 | 0.6180 ±0.0255 | 0.6091 ±0.0048 | 0.5553 ±0.0182 | 0.5400 ±0.0108 | 0.3794 ±0.0152 | 0.2718 ±0.0070 | 334.4 ±28.33 | 392.6 ±28.33 |

## D.7 CASE STUDY.

### D.7.1 RECOLLECTION-PATH WINNING CASE

To illustrate the difference between one-shot familiarity retrieval and recollection retrieval, we conduct a case study on the **PersonaMem** dataset. For clarity, we set $B = 2$ and $F = 2$ in the recollection process. Figure 18 presents the outputs under different retrieval modes for the query *"Can you suggest some new evidence-based practices to explore for healthcare decision-making?"*. The *Familiarity* retriever directly returns the top-10 highest-scoring entries, capturing salient but fragmented memories such as mentions of "conventional medicine" or "therapeutic modalities." While efficient, this strategy surfaces isolated fragments—sometimes with noisy mentions like "trying a new healthy recipe"—and fails to integrate them into a coherent chain. As a result, crucial contextual cues that span across sessions remain overlooked, highlighting the inherent limitation of one-shot retrieval.

By contrast, the *Recollection* process unfolds in multiple rounds. At each step, retrieved items are clustered into semantically coherent groups, and their centroids are blended with the query to form new probes. As shown in Figure 18, this branching expansion progressively uncovers complementary evidence: for instance, r=1 surfaces "modern science blending with age-old methods," r=2 brings in references to "gathering patient history" and "reviewing health records," and r=3 integrates higher-level anchors like "structured methodology and evidence-based practices." This stepwise enrichment not only reduces redundancy by grouping similar items but also reconstructs temporally dispersed details into a coherent memory trace. Compared with the static list from Familiarity retrieval, RF-Mem's recollection branch demonstrates a chain-like reconstruction process, aligning with the dual-process theory by simulating deliberate, effortful recall. Ultimately, this yields more diagnostic and contextually grounded evidence for answering the user's query.

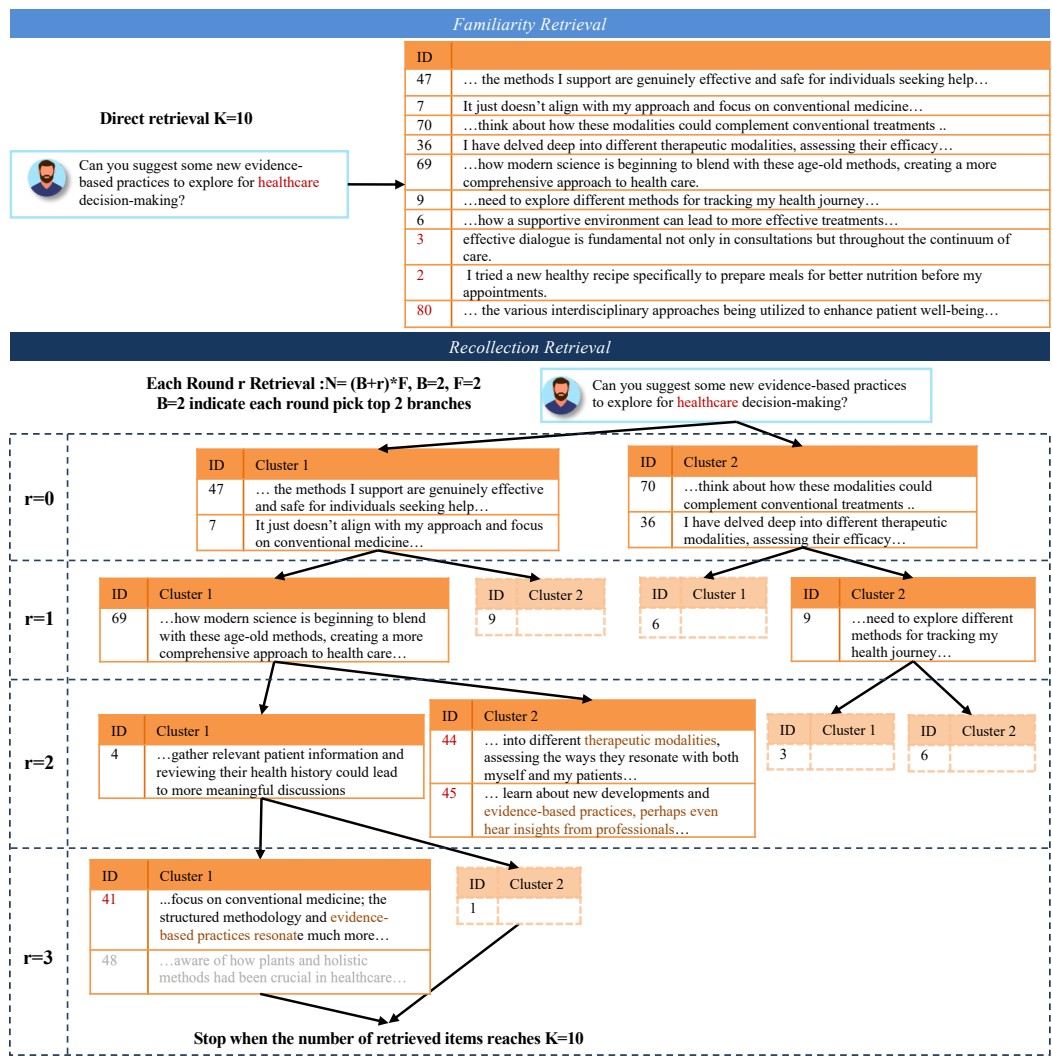

Figure 18: Case study from PersonaMem: comparison between *Familiarity* and *Recollection* retrieval where Recollection wins. Familiarity surfaces salient but fragmented evidence, while Recollection progressively reconstructs temporally distributed details via clustering and query refinement.

### D.7.2 FAMILIARITY-PATH WINNING CASE

To analysis the failure case, we again compare one-shot *Familiarity* retrieval with the multi-round *Recollection* process on the PersonaMem dataset. As shown in Figure 19, we show the retrieval result of dual paths for the query *"I'm considering developing a tool to manage finances while traveling. How could I ensure it helps me prioritize experiences like local cuisine or guided tours?"*. *Familiarity* retrieval outperforms *Recollection* in serving the user's intent. The one-shot Familiarity retriever returns a broad and query-aligned set of memories, many of which explicitly mention 'local experiences', 'finance management apps', or 'last trip'. These results directly address the user's need to "prioritize experiences while traveling," illustrating that high-scoring surface cues can sometimes be sufficient for intent coverage.

In contrast, the multi-round *Recollection* process drifts along a different semantic trajectory. Although its iterative clustering-and-refinement mechanism successfully produces deeper and more structured traces, the retrieved branches gradually converge toward 'long-term financial habits' and 'personal finance management'. This indicates that the recollection trajectory can overfit to dominant semantic clusters in the memory corpus, especially when several high-density branches are thematically coherent but misaligned with the user's immediate intent. As a result, the recollection path

becomes increasingly anchored in financial-management narratives, overlooking the travel-related aspects of the query.

Despite being a bad case for retrieval accuracy, this example is instructive. It exposes a core trade-off of deliberative recollection: the mechanism excels at reconstructing temporally dispersed, conceptually rich memory chains, yet it may over-emphasize internal coherence at the cost of task-oriented alignment. This case highlights the importance of balancing depth with intent fidelity, and underscores the need for more precise familiarity-uncertainty–guided strategy selection in future work.

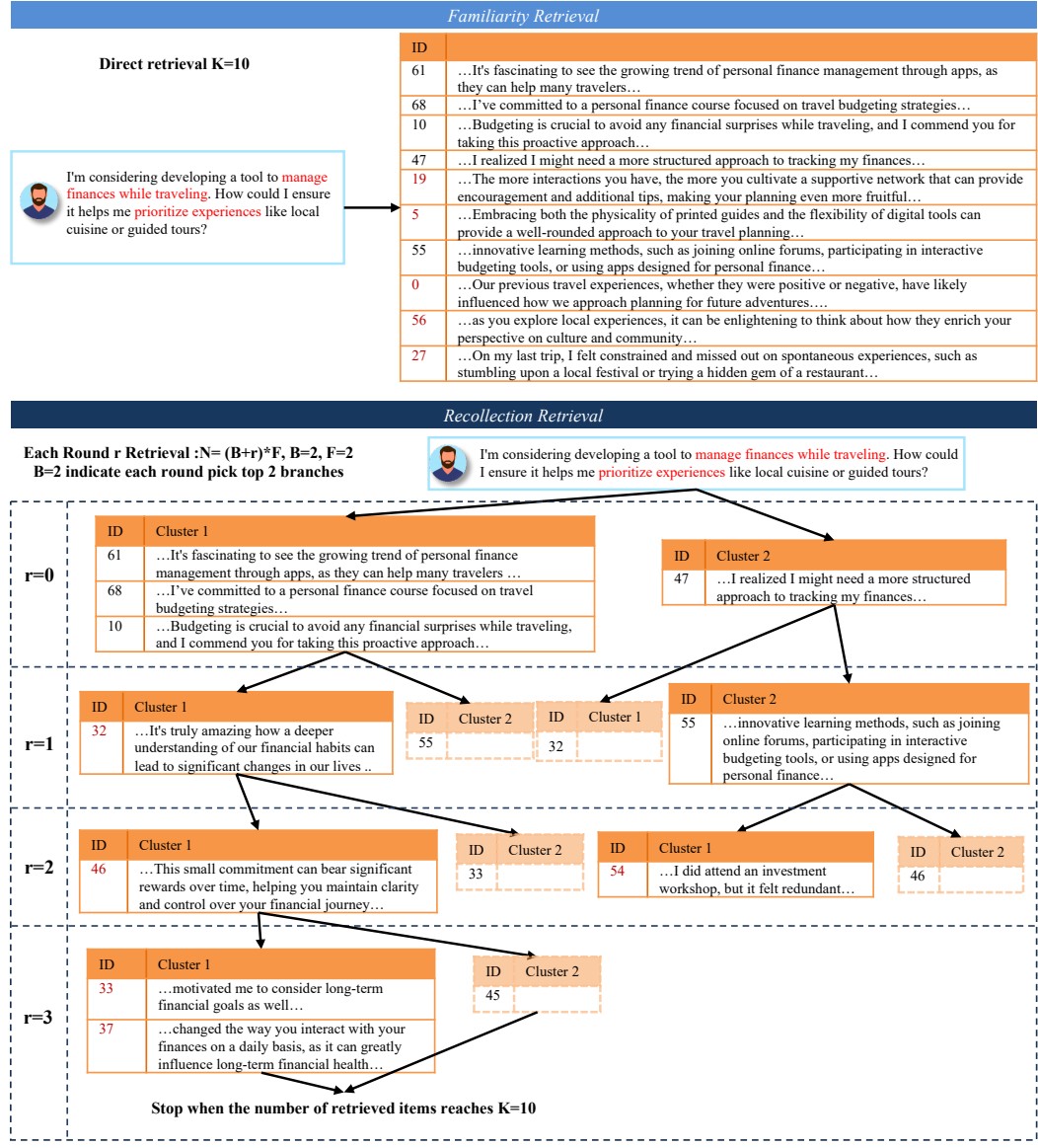

Figure 19: Case study from PersonaMem: comparison between *Familiarity* and *Recollection* retrieval where Familiarity wins. Familiarity produces broader and more relevant evidence aligned with prioritizing travel experiences, while Recollection progressively drills into long-term financial management and drifts away from the user's actual intent.

## D.8 ALTERNATIVE STUDY

In this section, we evaluate several alternative recollection mechanisms to test the robustness of RF-Mem beyond the proposed KMeans-based and $\alpha$-mix design. We replace the centroid module

with DBSCAN and Spectral Clustering, replace the $\alpha$-mix with gated mixing, and replace the entire module with graph-based expansion, while keeping all other components fixed. The comparison results in Table 15 highlight the generality of RF-Mem across diverse structural assumptions.

Table 15: Retrieval performance comparison under different recollection strategies.

| Metrics | Recall@5 | | | | | Recall@10 | | | | |
|---|---|---|---|---|---|---|---|---|---|---|
| Methods | Basic Info | Social Info | Pref Easy | Pref Hard | Overall | Basic Info | Social Info | Pref Easy | Pref Hard | Overall |
| *Basic KMeans* | | | | | | | | | | |
| Famili. | 0.4515 | 0.4852 | 0.4904 | 0.3659 | 0.4484 | 0.5879 | 0.6220 | 0.6442 | 0.5561 | 0.5964 |
| Recol. | 0.4379 | 0.4903 | 0.5128 | 0.3854 | 0.4491 | 0.5924 | 0.6859 | 0.5659 | 0.6267 | 0.6062 |
| RF-Mem | 0.4788 | 0.5091 | 0.4872 | 0.3854 | **0.4701** | 0.5924 | 0.6799 | 0.5707 | 0.6267 | **0.6071** |
| *DBSCAN cluster (KMeans alternative)* | | | | | | | | | | |
| Recol. | 0.4424 | 0.4940 | 0.4872 | 0.3512 | 0.4431 | 0.6045 | 0.6079 | 0.6891 | 0.5366 | 0.6028 |
| RF-Mem | 0.4879 | 0.5129 | 0.4744 | 0.3463 | **0.4669** | 0.5659 | 0.6079 | 0.6667 | 0.6015 | **0.6040** |
| *Spectral cluster (KMeans alternative)* | | | | | | | | | | |
| Recol. | 0.4591 | 0.4739 | 0.5000 | 0.3756 | 0.4522 | 0.6045 | 0.5978 | 0.6474 | 0.5366 | 0.5957 |
| RF-Mem | 0.4818 | 0.4928 | 0.4872 | 0.3707 | **0.4651** | 0.6015 | 0.6041 | 0.6474 | 0.5610 | **0.6001** |
| *Gate ($\alpha$-mix alternative)* | | | | | | | | | | |
| Recol. | 0.4439 | 0.4739 | 0.5128 | 0.3902 | 0.4491 | 0.5924 | 0.6016 | 0.6795 | 0.5317 | 0.5936 |
| RF-Mem | 0.4667 | 0.4928 | 0.5000 | 0.3756 | **0.4602** | 0.5924 | 0.6079 | 0.6667 | 0.5610 | **0.5988** |
| *Graph + Breadth-First Search (whole alternative)* | | | | | | | | | | |
| Recol. | 0.4167 | 0.4739 | 0.4872 | 0.3805 | 0.4314 | 0.5591 | 0.5887 | 0.5887 | 0.5366 | 0.5722 |
| RF-Mem | 0.4258 | 0.4739 | 0.4872 | 0.3756 | **0.4349** | 0.5742 | 0.6220 | 0.6442 | 0.5561 | **0.5899** |

### D.8.1 DBSCAN RECOLLECTION (KMEANS ALTERNATIVE)

**Experiment setup.** To examine whether the assumptions of KMeans (balanced and approximately spherical clusters) influence recollection, we evaluate DBSCAN (Ester et al., 1996) as a density-based alternative. DBSCAN identifies arbitrarily shaped clusters and automatically detects noise points, providing a contrasting clustering geometry. The retrieval backbone, mixing rule, and evaluation setup remain unchanged to ensure a clean comparison between recollection mechanisms.

**Findings.** Results in Table 15 show that DBSCAN-based recollection performs competitively, and RF-Mem (DBSCAN) consistently improves over its recollection baseline. This demonstrates that the effectiveness of $\alpha$-mixing is not tied to KMeans-specific assumptions and remains stable under density-driven neighborhood structures.

### D.8.2 SPECTRAL CLUSTERING RECOLLECTION (KMEANS ALTERNATIVE)

**Experiment setup.** We further evaluate Spectral (Ng et al., 2001) Clustering, which identifies clusters via graph Laplacian eigenvectors and effectively models manifold-shaped structures. This setup tests whether recollection remains stable when memory clusters deviate from centroid-based geometry. All configurations follow the evaluation results summarized in Table 15.

**Findings.** As reported in Table 15, RF-Mem (Spectral) achieves consistent gains over the Spectral baseline, improving R@5 Overall from 0.4522 to 0.4651. These results indicate that $\alpha$-mixing generalizes well across diverse cluster geometries and maintains its geometry-respecting update behavior even under manifold-structured partitions.

### D.8.3 NONLINEAR GATED MIXING ($\alpha$-MIX ALTERNATIVE)

**Experiment setup.** Motivated by nonlinear interpolation strategies, we implement a gated mixing variant in which the update coefficient is determined by a sigmoid gate applied to the similarity between the current query representation and the cluster centroid. Specifically, given the query vector $\mathbf{x}^{(r)}$ at iteration $r$ and the corresponding centroid $\mathbf{g}_b^{(r)}$, the gated interpolation is defined as:

$$g(\mathbf{x}^{(r)}, \mathbf{g}_b^{(r)}) = \sigma\left(\mathbf{x}^{(r)\top}\mathbf{g}_b^{(r)}\right), \tag{8}$$

and the updated query is computed via:

$$\mathbf{x}_b^{(r+1)} = \text{norm}\left(g\,\mathbf{x}^{(r)} + (1-g)\,\mathbf{g}_b^{(r)} + \mathbf{x}_t\right), \tag{9}$$

where $\sigma(\cdot)$ denotes the sigmoid function and $\mathbf{x}_t$ is the original query. This formulation introduces a nonlinear, content-adaptive interpolation mechanism that contrasts with the proposed linear $\alpha$-mixing. All experiments follow the unified evaluation protocol in Table 15 to ensure comparability.

**Findings.** As shown in Table 15, gated mixing improves over Familiarity retrieval but does not surpass the effectiveness of the proposed $\alpha$-mixing rule. RF-Mem (Gate) obtains 0.4602 R@5 Overall, compared with 0.4701 for our KMeans-based implementation. The performance gap is particularly noticeable in Social Info and Pref-Hard subsets, where recollection plays a critical role. These results indicate that although nonlinear interpolation is a reasonable alternative, $\alpha$-mixing provides a stable and geometry-aligned update, consistent with its interpretation as a manifold-aware update rule on the unit hypersphere.

### D.8.4 GRAPH-BASED RECOLLECTION (WHOLE ALTERNATIVE)

**Experiment setup.** Finally, we examine a structure-free alternative by constructing a KNN graph (Zhang et al., 2025a) over a user memory corpus and performing Breadth-First Search (BFS) expansion. This mechanism collects local neighborhoods via graph traversal rather than via clustering. The full evaluation follows the protocol summarized in Table 15.

**Findings.** Table 15 shows that graph-based recollection performs weaker than clustering-based recollection. Nevertheless, RF-Mem consistently improves over its graph baseline, demonstrating the robustness. The comparison suggests that clustering more effectively aggregates semantically coherent neighborhoods, whereas BFS may over-expand into dense yet irrelevant regions.

### D.9 FULL-EXPLORATION STUDY

To further examine whether the effectiveness of RF-Mem depends on the early-stop rule used in the recollection path, we conduct an additional analysis where the expansion process is allowed to continue until all reachable candidates (within a predefined maximum depth) have been explored.

Table 16: Retrieval performance under full exploration.

| Metrics | Recall@5 | | | | | Recall@10 | | | | |
|---|---|---|---|---|---|---|---|---|---|---|
| Methods | Basic Info | Social Info | Pref Easy | Pref Hard | Overall | Basic Info | Social Info | Pref Easy | Pref Hard | Overall |
| *Basic KMeans (Early-Stop)* | | | | | | | | | | |
| Famili. | 0.4515 | 0.4852 | 0.4904 | 0.3659 | 0.4484 | 0.5879 | 0.6442 | 0.5561 | 0.5964 |
| Recol. | 0.4379 | 0.4903 | 0.5128 | 0.3854 | 0.4491 | 0.5924 | 0.6859 | 0.5659 | 0.6267 | 0.6062 |
| RF-Mem | 0.4788 | 0.5091 | 0.4872 | 0.3854 | 0.4701 | 0.5924 | 0.6799 | 0.5707 | 0.6267 | 0.6071 |
| *Basic KMeans (Fully Exploration)* | | | | | | | | | | |
| Recol. | 0.4530 | 0.4827 | 0.5128 | 0.3805 | 0.4537 | 0.6015 | 0.6204 | 0.6763 | 0.5415 | 0.6036 |
| RF-Mem | 0.4848 | 0.5016 | 0.5000 | 0.3756 | **0.4709** | 0.6015 | 0.6267 | 0.6667 | 0.5659 | **0.6083** |
| *Graph + Breadth-First Search (Early-Stop)* | | | | | | | | | | |
| Recol. | 0.4167 | 0.4739 | 0.4872 | 0.3805 | 0.4314 | 0.5591 | 0.5887 | 0.5887 | 0.5366 | 0.5722 |
| RF-Mem | 0.4258 | 0.4739 | 0.4872 | 0.3756 | 0.4349 | 0.5742 | 0.6220 | 0.6442 | 0.5561 | 0.5899 |
| *Graph + Breadth-First Search (Fully Exploration)* | | | | | | | | | | |
| Recol. | 0.4530 | 0.4701 | 0.4776 | 0.3902 | 0.4486 | 0.5818 | 0.5642 | 0.6635 | 0.5659 | 0.5841 |
| RF-Mem | 0.4758 | 0.4890 | 0.4776 | 0.3854 | **0.4629** | 0.5924 | 0.6220 | 0.6442 | 0.5561 | **0.5986** |

**Experiment setup.** Unlike the default setting, in which the recollection process terminates once the top-$K$ quota is satisfied, this full-exploration variant aggregates the complete expanded set and performs a final dense re-ranking step over all collected items. This design enables us to test whether the proposed recollection mechanism remains robust when given a substantially larger search budget. Table 16 summarizes the results for both the KMeans-based and Graph-based recollection mechanisms under the early-stop and full-exploration settings. Across all categories, full exploration increases the coverage of semantically relevant nodes, providing a stricter evaluation of the recollection path.

**Findings. First**, we observe that allowing full exploration followed by a global re-ranking step consistently improves RF-Mem compared to the early-stopped variant. For instance, RF-Mem (Full) achieves an Overall Recall@5 of 0.4709 under KMeans, surpassing the early-stop score of 0.4701. This demonstrates that RF-Mem continues to benefit from deeper recollection when additional evidence is made available, reinforcing the stability and scalability of the proposed $\alpha$-mixing updates.

**Second**, under the full-exploration setting, the graph-based variant shows clear improvements over both its early-stopped counterpart and the Familiarity (dense) baseline. RF-Mem (Graph+bfs, Full) obtains an Overall Recall@10 of 0.5986, outperforming the early-stop result of 0.5899. This indicates that graph-guided recollection, despite being weaker than KMeans in the default setting, becomes more competitive when provided with additional traversal depth. These findings support our broader claim that deliberate, structure-aware exploration of the memory space offers a valuable and robust alternative formulation for retrieval.

### D.10 ROUTING ROBUSTNESS ACROSS EMBEDDING MODELS

To assess the sensitivity of RF-Mem to the underlying embedding model, we investigate how embedding quality and calibration influence routing decisions. Because both the similarity-based uncertainty measure and the recollection path depend on the geometry of the embedding space, understanding the extent to which different retrievers alter routing behavior or introduce failure risks is crucial for evaluating the robustness of the framework.

**Experiment Setup** To answer the question of whether routing accuracy varies across different retrievers, we measure routing statistics for three embedding models on PersonaBench: `multi-qa-MiniLM-L6-cos-v1`, `all-mpnet-base-v2`, and `bge-base-en-v1.5`. For each model, we record (i) routing frequencies into the familiarity and recollection branches, and (ii) retrieval accuracy. All other components of RF-Mem remain fixed to isolate the effect of the embedding backbone.

**Findings.** Table 17 summarizes the result. We can find: **First**, Stronger retrievers route more queries into the recollection path, while weaker retrievers rely more heavily on fast familiarity. MiniLM and MPNet trigger substantially more recollection transitions, indicating that their embedding spaces provide more reliable cluster structures for iterative refinement. In contrast, BGE, which exhibits less stable similarity distributions, routes a significantly larger number of queries to familiarity. **Second**, the routing behavior demonstrates that the mechanism adapts to the embedding geometry in a principled way. Stronger retrievers provide clearer cluster boundaries, which encourages recollection, whereas noisier or weakly calibrated embeddings shift routing toward familiarity as a more reliable fallback.

Table 17: Routing statistics and retrieval performance across different embedding models.

| Retriever | Time | Route→Famili. | Route→Recol. | R@5 Overall |
|---|---|---|---|---|
| multi-qa-MiniLM-L6-cos-v1 | 9.16ms | 60 | 203 | 0.4701 |
| all-mpnet-base-v2 | 8.33ms | 15 | 248 | 0.4009 |
| bge-base-en-v1.5 | 10.14ms | 140 | 123 | 0.3836 |

## E PSEUDOCODE OF RF-MEM

**RF-Mem algorithm.** Algorithm 1 begins with a short probe to estimate retrieval certainty and then switches between a fast *Familiarity* path and a deliberate *Recollection* path. Given the query embedding $\mathbf{x}_t$, the retriever returns the top-$k_p$ probe candidates and produces a temperature–scaled score distribution $p$; we compute the list entropy $H(p)$ and the mean score $\bar{s}$, then apply thresholds $(\theta_{\text{high}}, \theta_{\text{low}})$ together with an entropy gate $\tau$ as specified in Eqs. (1)–(2). If the familiarity signal is strong, that is $\bar{s} \geq \theta_{\text{high}}$ or $H(p) \leq \tau$, RF-Mem executes a one–shot *Familiarity* retrieval that returns the top-$K$ items by similarity. If the signal is weak, that is $\bar{s} \leq \theta_{\text{low}}$ or $H(p) > \tau$, or if the probe yields no hits, RF-Mem invokes Algorithm 3 to perform stepwise *Recollection*. In each round $r$ of recollection, we retrieve top-$N$ candidates with $N = (B + r)F$, cluster them by KMeans into at most $B$ groups, form a centroid for each group, and update the query by $\alpha$-mixing the current query with the centroid while retaining a residual from the original query, then continue with the resulting queries as a beam of size $B$. Unique hits are accumulated across rounds, scores are aggregated per item, and the process stops early when at least $K$ unique items have been collected or when the round limit $R$ is reached. The probe therefore preserves one–shot efficiency when certainty is high, while the recollection loop builds chain–like evidence under uncertainty, where $B$ and $F$ regulate breadth and $\alpha$ controls the balance between exploration and query stability. And theoretical analysis can be found at Appendix F.

---

**Algorithm 1** RF-Mem: Entropy-guided Switching Between Familiarity and Recollection (with in-lined entropy)

---

**Require:** Retriever $\mathcal{R}$ with indexed memories $\{m_i\}_{i=1}^M$ and embeddings $z_i = \phi(m_i)$; query $q$ with embedding $x_t = \phi(q)$; probe size $k_p$; temperature $\lambda$; entropy threshold $\tau$; mean-score thresholds $(\theta_{\text{high}}, \theta_{\text{low}})$; final budget $K$. Recollection params: beam $B$, fanout $F$, max rounds $R$, mix rate $\alpha$, per-round candidate size $N$.

**Ensure:** Ranked memory set $\mathcal{S}$ with $|\mathcal{S}| \leq K$.

1: $(\mathbf{s}, \mathbf{id}) \leftarrow \mathcal{R}.\text{PROBE}(x_t, K)$            $\triangleright$ Top-$K$ probe scores
2: $\bar{s} \leftarrow \text{mean}(\mathbf{s})$
3: **(Calculate entropy)** Let $k \leftarrow |\mathbf{s}|$ and for $i = 1..k$ set
4:     $z_i \leftarrow \lambda(s_i - \max_j s_j); \quad p_i \leftarrow \exp(z_i)/\sum_{j=1}^k \exp(z_j)$
5: $H \leftarrow -\sum_{i=1}^k p_i \log p_i$
6: **if** $\bar{s} \geq \theta_{\text{high}}$ **or** $H \leq \tau$ **then**
7:     **return** $\text{FAMILIARITYTOPK}(x_t, K, \mathcal{R})$
8: **else if** $\bar{s} \leq \theta_{\text{low}}$ **or** $H > \tau$ **then**
9:     **return** $\text{RECOLLECTION}(x_t, K, B, F, R, \alpha, N, \mathcal{R})$
10: **end if**

---

**Algorithm 2** Familiarity Retrieval

---

**Require:** Query $x_t$, budget $K$, retriever $\mathcal{R}$

1: For each memory $m_i$, compute $s_i \leftarrow \langle x_t, z_i \rangle$      $\triangleright$ cosine or inner product after normalization
2: $\mathcal{S} \leftarrow$ top-$K$ items by $s_i$ (optionally filter by a floor)
3: **return** $\mathcal{S}$

---

**Algorithm 3** Recollection Retrieval: retrieve $\rightarrow$ cluster $\rightarrow$ $\alpha$-mix $\rightarrow$ iterate

---

**Require:** $x_t, K, B, F, R, \alpha, N$, retriever $\mathcal{R}$

**Ensure:** Ranked set $\mathcal{S}$ with $|\mathcal{S}| \leq K$

1: $x^{(0)} \leftarrow \text{norm}(x_t);$   Beam $\leftarrow \{x^{(0)}\};$   Seen $\leftarrow \emptyset;$   Bag $\leftarrow \emptyset$
2: **for** $r = 0$ **to** $R - 1$ **do**
3:     Next $\leftarrow \emptyset;$   $N \leftarrow (B + r) \times F$                        $\triangleright$ $N < K$
4:     **for all** $x^{(r)} \in$ Beam **do**
5:        $C^{(r)} \leftarrow \mathcal{R}.\text{TOPN}(x^{(r)}, N)$           $\triangleright$ $C^{(r)} = \text{Top-}N\{(m_i, \langle x^{(r)}, z_i \rangle)\}$
6:        Cluster $\{z_i : m_i \in C^{(r)}\}$ into $k = \min(B, |C^{(r)}|)$ groups by KMeans
7:        **for** $b = 1$ **to** $k$ **do**
8:           $G_b^{(r)} \leftarrow$ index set of cluster $b;$   $g_b^{(r)} \leftarrow \text{norm}\big(\frac{1}{|G_b^{(r)}|} \sum_{i \in G_b^{(r)}} z_i\big)$
9:           $x_b^{(r+1)} \leftarrow \text{norm}\big(\alpha\, x^{(r)} + (1 - \alpha)\, g_b^{(r)} + x_t\big)$
10:          Append $\big(x_b^{(r+1)}, G_b^{(r)}\big)$ to Next
11:        **end for**
12:     **end for**
13:     **if** Next $= \emptyset$ **then break**
14:     **end if**
15:     Score each $(x_b^{(r+1)}, G_b^{(r)})$ by $\sum_{i \in G_b^{(r)}} \langle x_b^{(r+1)}, z_i \rangle$; keep top-$B$ as new Beam
16:     **for all** kept $\big(x_b^{(r+1)}, G_b^{(r)}\big)$ **do**
17:        **for all** $i \in G_b^{(r)}$ **do**
18:           **if** $i \notin$ Seen **then** insert $(i, \langle x_b^{(r+1)}, z_i \rangle)$ into Bag; add $i$ to Seen
19:           **end if**
20:        **end for**
21:     **end for**
22:     **if** $|\text{Bag}| \geq K$ **then break**
23:     **end if**
24: **end for**
25: For each id in Bag take top-$K$ as $\mathcal{S}$
26: **return** $\mathcal{S}$

---

# F  THEORETICAL ANALYSIS

**Preliminaries.**  Let the user memory be $\mathcal{M} = \{m_i\}_{i=1}^M$ with embeddings $\mathbf{z}_i = \phi(m_i)$ and a query $q$ encoded as $\mathbf{x}_t = \phi(q)$, with unit normalization. Define similarity scores $s_i = \langle \mathbf{x}_t, \mathbf{z}_i \rangle$ and the probe list $\mathcal{C} = \texttt{Top-}K\big(\{(m_i, s_i)\}_{i=1}^M\big)$. Following the paper, define a tempered softmax over probe scores

$$p_i = \frac{\exp\big(\lambda(s_i - \max_j s_j)\big)}{\sum_{j=1}^K \exp\big(\lambda(s_j - \max_\ell s_\ell)\big)}, \qquad i = 1, \ldots, K, \tag{10}$$

with entropy $H(p) = -\sum_{i=1}^K p_i \log p_i$ and mean similarity $\bar{s} = \frac{1}{K} \sum_{i=1}^K s_i$. The RF-Mem selection is

$$\text{Strategy}(q) = \begin{cases} \text{Familiarity}, & \bar{s} \geq \theta_{\text{high}}, \\ \text{Recollection}, & \bar{s} \leq \theta_{\text{low}}, \\ \begin{cases} \text{Familiarity}, & H(p) \leq \tau, \\ \text{Recollection}, & H(p) > \tau, \end{cases} & \theta_{\text{low}} < \bar{s} < \theta_{\text{high}}. \end{cases} \tag{11}$$

In the Familiarity path, the retriever returns $\texttt{Top-}K$ by $s_i$. In the Recollection path, the system iterates *retrieve–cluster–mix*: at round $r$ it retrieves $\texttt{Top-}N_r$ with $N_r = (B + r)F \leq K$, clusters into $B$ groups with centroids $\mathbf{g}_b^{(r)}$, and forms recollect queries

$$\mathbf{x}_b^{(r+1)} = \text{norm}\big(\alpha \mathbf{x}^{(r)} + (1 - \alpha)\mathbf{g}_b^{(r)} + \mathbf{x}_t\big), \quad b = 1, \ldots, B, \tag{12}$$

for $r = 0, \ldots, R - 1$, then returns $\texttt{Top-}K$ from the union $\bigcup_{r=0}^R \mathcal{C}^{(r)}$. This matches the method description and notation in the main content.

## F.1  RISK-MINIMIZING SELECTION

We formalize the selection in equation 11 as minimizing a retrieval risk that trades correctness and cost. Let $\mathcal{E}(\text{mode} \mid q)$ denote the probability of returning an insufficient set for $q$ under a mode $\text{mode} \in \{\text{Familiarity}, \text{Recollection}\}$, and let $C(\text{mode})$ denote the expected computation cost. For a penalty $\beta > 0$ define

$$\mathcal{L}(\text{mode} \mid q) = \mathcal{E}(\text{mode} \mid q) + \beta\, C(\text{mode}). \tag{13}$$

**Lemma 1 (Monotonicity of proxy signals)** *Assume the similarity distribution admits a monotone likelihood ratio in $s_i$ between relevant and nonrelevant items, and that the probe softmax temperature $\lambda$ is fixed. Then $\mathcal{E}(\text{Familiarity} \mid q)$ is nonincreasing in $\bar{s}$ and nondecreasing in $H(p)$, while $C(\text{Familiarity}) < C(\text{Recollection})$.*

*Proof.* Let $Y_i \in \{0, 1\}$ for the relevance label of item $i$ (1 means relevant). Assume a monotone likelihood ratio for scores so the posterior relevance $\pi(s) = \Pr(Y = 1 \mid s)$ is nondecreasing in $s$. For Familiarity, the miss probability given scores is $\prod_{i \in K}\big(1 - \pi(s_i)\big)$. Any coordinatewise increase of $(s_i)_{i \in K}$ decreases each factor, hence decreases the product, so $\mathcal{E}(\text{Familiarity} \mid q)$ is nonincreasing in $\bar{s}$. For entropy, softmax $p$ is Schur–concave: smaller $H(p)$ implies larger $p_{\max}$ and larger top margins $s_{(1)} - s_{(j)}$, which increase $\pi(s_{(1)})$ and $\sum_{i \in I_K} \pi(s_i)$, thus the product and its expectation are nonincreasing; therefore $\mathcal{E}(\text{Familiarity} \mid q)$ is nondecreasing in $H(p)$. For costs, $C(\text{Familiarity}) = \Theta(K\, c_{\text{sim}})$, while Recollection performs at least one extra retrieve–cluster–mix round, so $C(\text{Recollection}) \geq C(\text{Familiarity}) + R\, c_{\text{clust}}(B) > C(\text{Familiarity})$ for $R \geq 1$. $\square$

**Theorem 1 (Threshold optimality within monotone policies)** *Consider the class $\Pi$ of policies that are monotone in $(\bar{s}, H)$ in the sense of Lemma 1. Within $\Pi$, a two-threshold policy of the form equation 11 minimizes the pointwise risk equation 13.*

*Proof.* By Lemma 1, $\mathcal{E}(\text{Familiarity} \mid q)$ is nonincreasing in $\bar{s}$ and nondecreasing in $H$. Define

$$t := \mathcal{L}(\text{Recollection} \mid q), \qquad f(\bar{s}, H) := \mathcal{L}(\text{Familiarity} \mid q) = \mathcal{E}(\text{Familiarity} \mid q) + \beta\, C(\text{Familiarity}). \tag{14}$$

Then the Familiarity-region is the sublevel set

$$\mathcal{D} := \{(\bar{s}, H) : f(\bar{s}, H) \leq t\}, \tag{15}$$

which is a down-set in $(\bar{s} \uparrow, H \downarrow)$ ("south–east orthant"). Hence the risk-minimizing monotone policy is $\mathbf{1}_{\mathcal{D}}$. Introduce axis-aligned thresholds

$$\theta_{\text{high}} := \inf\{\bar{s} : \sup_H f(\bar{s}, H) \le t\}, \qquad \theta_{\text{low}} := \sup\{\bar{s} : \inf_H f(\bar{s}, H) > t\}, \tag{16}$$

and, on $\bar{s} \in (\theta_{\text{low}}, \theta_{\text{high}})$,

$$H_\star(\bar{s}) := \inf\{H : f(\bar{s}, H) \le t\}, \qquad \tau := \sup_{\bar{s} \in (\theta_{\text{low}}, \theta_{\text{high}})} H_\star(\bar{s}). \tag{17}$$

Therefore

$$\text{Familiarity} \iff (\bar{s} \ge \theta_{\text{high}}) \text{ or } (\theta_{\text{low}} < \bar{s} < \theta_{\text{high}} \ \& \ H \le \tau), \tag{18}$$

and Recollection otherwise, which is exactly the two-threshold gate. $\qquad\square$

### F.2 An Entropy Certificate for Correctness

Let $p_{\max} = \max_i p_i$. For fixed $p_{\max}$, the maximal entropy occurs when the residual mass is uniform, that is

$$H(p) \le h_2(p_{\max}) + (1 - p_{\max}) \log(K - 1), \tag{19}$$

where $h_2(x) = -x \log x - (1 - x) \log(1 - x)$ is the binary entropy. Define the inverse certificate

$$\phi_K(\tau) = \max\{x \in [1/K, 1] : h_2(x) + (1 - x) \log(K - 1) \le \tau\}. \tag{20}$$

**Lemma 2 (Entropy certificate)** *If $H(p) \le \tau$, then $p_{\max} \ge \phi_K(\tau)$.*

*Proof.* Rearrange equation 19 and apply the definition of $\phi_K(\tau)$. $\qquad\square$

**Proposition 1 (Bound on familiarity error under low entropy)** *Suppose that returning the $\text{Top-}K$ set suffices whenever the true relevant item has $p_i \ge \rho$ for some $\rho \in (0, 1)$. Under $H(p) \le \tau$ with $\phi_K(\tau) \ge \rho$, the familiarity mode achieves negligible miss probability with respect to this sufficient condition.*

Proposition 1 provides a certificate: a small entropy ensures a large $p_{\max}$, which guarantees that the best evidence is included by $\text{Top-}K$ under a mild sufficiency condition, hence Familiarity is safe in the certified region.

### F.3 Sub-Gaussian Mean Similarity and Gating Reliability

Assume probe scores $\{s_i\}_{i=1}^K$ are independent sub-Gaussian with proxy mean $\mu$ and variance proxy $\sigma^2$. Let $\widehat{\mu} = \bar{s}$ and fix thresholds $\theta_{\text{low}} < \theta_{\text{high}}$.

**Proposition 2 (Gating error bound via concentration)** *For any $\delta > 0$,*

$$\Pr\left(\widehat{\mu} - \mu \le -\delta\right) \le \exp\left(-\frac{K\delta^2}{2\sigma^2}\right), \qquad \Pr\left(\widehat{\mu} - \mu \ge \delta\right) \le \exp\left(-\frac{K\delta^2}{2\sigma^2}\right). \tag{21}$$

*If the true regime satisfies $\mu \ge \theta_{high} + \delta$ (familiar) or $\mu \le \theta_{low} - \delta$ (unfamiliar), the probability of mis-selection due to mean estimation is bounded by $\exp(-K\delta^2/(2\sigma^2))$.*

Proposition 2 shows that increasing $K$ tightens the reliability of the mean-based selection of the familiarity and the recollection paths.

### F.4 Complexity–Coverage Trade-off

Let the cost of a similarity evaluation be $c_{\text{sim}}$ and of a $B$-way $k$-means update be $c_{\text{clust}}(B)$ on a batch of size $N_r$.

**Proposition 3 (Complexity bounds)** *Familiarity has time $T_{Fam} = O(K c_{sim})$. Recollection with parameters $(B, F, R)$ has time*

$$T_{Rec} = O\left(\sum_{r=0}^R \left(N_r c_{sim} + c_{clust}(B)\right)\right) = O\left(R(BF) c_{sim} + R c_{clust}(B)\right), \tag{22}$$

*since $N_r \le (B+r)F \le O(BF+RF)$. Therefore $T_{Rec} = O\big((BF+RF)\,c_{sim} + R\,c_{clust}(B)\big)$, which is polynomial in $(B, F, R)$ and strictly lower than full-context processing $O(M)$ when $BF+RF \ll M$.*

This formalizes the method's bounded overhead relative to full-context, consistent with empirical latency advantages reported in the paper.

**Discussion.** Lemmas 1 and 2 justify using $(\bar{s}, H)$ as reliable control signals. Theorems 1 show that the selection is optimal within a broad monotone class. Propositions 2 and 3 bound mis-selection and computation. Together these results explain the empirical accuracy–latency improvements of RF-Mem across corpora and tasks.

## G   LLM USAGE DISCLOSURE

In accordance with ICLR 2026 policy, we disclose our use of large language models (LLMs) in preparing this manuscript. We employed GPT-5 (OpenAI) solely to aid in polishing the writing, specifically for improving clarity, grammar, and sentence structure across sections. All technical content, algorithmic contributions, experimental results, and scientific conclusions remain entirely the authors' own work, without any LLM involvement.

## H   LIMITATION AND FUTURE WORK

**Limitations.** While RF-Mem demonstrates consistent advantages across corpora and tasks, several aspects remain simplified. First, our current evaluation is confined to dialogue-style personalized memory and does not yet explore other modalities (e.g., cross-modal histories). Second, we adopt list entropy as a lightweight uncertainty proxy; although effective, it may not fully capture finer-grained task difficulty or user intent. Third, our uncertainty signal is based on similarity scores and entropy, which may not fully capture semantic ambiguity. Finally, our retrieval operates over a static embedding index, without modeling temporal updates of memory or potential conflicts across long-term sessions. Moreover, because RF-Mem improves the surfacing of long-term user history, downstream deployments should include safeguards to prevent unintended resurfacing of sensitive information, a direction we identify as important future work for quantifying and mitigating such risks.

**Future Work.** Several directions arise naturally from the above limitations. First, extending RF-Mem to multi-modal or cross-domain settings (e.g., multimodal dialogue) would allow testing its generality beyond personalized text-based memory. Second, uncertainty estimation could be enhanced by integrating richer signals—such as calibration measures or user feedback—to complement list entropy and better capture task difficulty. In addition, incorporating semantic or contextual uncertainty cues beyond similarity distributions may further improve the reliability of the strategy selection mechanism. Third, incorporating temporal dynamics into the memory index (e.g., recency weighting, conflict resolution, or session-aware updates) may further improve long-term personalization. Finally, exploring tighter integration between retrieval and generation—for example, jointly optimizing the entropy threshold with the LLM's decoding behavior—could yield a more unified and adaptive personalized reasoning framework.

