# OpenReview forum: "Evoking User Memory: Personalizing LLM via Recollection-Familiarity Adaptive Retrieval"
_ICLR.cc/2026/Conference — ICLR 2026 Poster_

### Official Review · Reviewer_QRxk · 2025-10-27

**Soundness:** 3
**Presentation:** 4
**Contribution:** 3
**Rating:** 8
**Confidence:** 4

**Summary:**

The paper proposes RF-Mem, a retrieval framework for LLM personalization. The core idea is to adaptively choose between two memory-retrieval modes, inspired by dual-process human memory:
Familiarity: fast dense top-K retrieval when the query matches past user memory with high confidence.
Recollection: iterative, multi-hop expansion when confidence is low — retrieve candidates, cluster them, mix cluster centroids with the original query embedding, and retrieve again to surface contextual / episodic details.
A simple gate based on mean similarity and similarity-score entropy decides which path to use. The goal is to surface only the most relevant personal memories for context, instead of injecting the full user history.
The method is evaluated on multi-session personalization benchmarks with up to ~1M tokens of per-user memory. RF-Mem outperforms (i) no memory, (ii) giving the whole history, (iii) plain dense retrieval, and (iv) recollection-only, while keeping inference context small and latency close to dense retrieval. The authors also show RF-Mem working on top of summarized memory stores, not only raw logs.

**Strengths:**

### 1. Strong motivations and grounding in cognitive science.

The familiarity vs recollection split is explicitly motivated by dual-process models of human memory. The authors lay out a concrete computational mapping (confidence gating → familiarity, uncertainty → recollection). This gives the method conceptual clarity and makes the design feel principled.

### 2. Clear formalism / presentation.

The framework is described cleanly, with the retrieval controller, the gating rule, and the recollection procedure all specified in enough detail to reimplement.

### 3. Scales to large memories.
RF-Mem remains effective under very large user histories (hundreds of thousands to a million tokens) where “put everything in context” is impossible and vanilla dense retrieval starts missing important long-tail details.

### 4. Method is practical with low overhead.
The approach adds only modest latency over standard dense retrieval and explicitly tracks context budget. No LLM fine-tuning is required. It is interesting to see how much latency the dual process formulation saves over pure recollection, without losing performance.

### 5. Modularity.
RF-Mem can sit on top of summarized memories rather than raw transcripts, suggesting it can plug into existing assistant memory stacks.

**Weaknesses:**

### 1. Lack of real-user evaluations.

All experiments use synthetic or benchmarked personalization data. There’s no human evaluation of helpfulness or preference alignment, and no analysis of noisy, contradictory, or sensitive histories.

### 2. Heuristic gating mechanism.
The decision to enter recollection mode is based on manually chosen similarity / entropy thresholds. There’s no learned controller and no robustness study across domains. Furthermore, the authors do not detail why certain thresholds were selected, or how one should optimise these.

### 3. Baselines.
Comparisons are mostly against: full context, dense top-K, and recollection-only. Stronger long-term memory agents (e.g. reflective / hierarchical memory systems) aren’t fully represented.

### 4. Safety/privacy.
Since RF-Mem is explicitly better at surfacing detailed personal history, it can also resurface sensitive information the user didn’t mean to bring into the current turn. The authors acknowledge this risk but do not quantify or mitigate it.

**Questions:**

1. How are the gating thresholds selected? Could the gate be learned?
2. Can you show qualitative failure cases where the gate picked the wrong mode?

---

> ### Author Response · Authors · 2025-11-19
> **Response to Reviewer QRxk [Part 1]**
>
> **Dear Reviewer QRxk**,
>
> We sincerely thank you for your thoughtful, detailed, and exceptionally constructive review. We are very grateful for your generous recognition of the paper’s conceptual motivations, clarity of presentation, scalability, and practical relevance. Your evaluation accurately captures the core intention of RF-Mem, and your comments reflect a deep understanding of both cognitive foundations and retrieval system design. We especially appreciate the strengths you highlighted—including the principled dual-process grounding, the clean and reproducible formulation, the efficiency advantages over pure recollection, and the modularity of RF-Mem within existing memory stacks. Your remarks were highly encouraging to us, and they significantly helped us improve the clarity, robustness, and completeness of the revised version. We have carefully addressed each point below and substantially updated the paper accordingly.
>
>
> ## 1. User evaluation
> >**Your Concern (W1):** Lack of real-user evaluations.All experiments use synthetic or benchmarked personalization data. There’s no human evaluation of helpfulness or preference alignment, and no analysis of noisy, contradictory, or sensitive histories.
>
> We sincerely thank the reviewer for highlighting the importance of real-user evaluation. In response, we conducted a small-scale human study to directly assess whether the proposed recollection path is beneficial and whether RF-Mem’s routing decisions align with human judgments.
>
> ### 1.1 Experiment Setup
> We recruited **five PhD-level annotators** to evaluate **25 samples** randomly selected from PersonaMem.
> For each sample, annotators were shown:
>
> 1. The **user query**, and
> 1. The **Familiarity** retrieval result, and
> 2. The **Recollection** retrieval result,
>
> Annotators independently judged which retrieval result (familiarity vs. recollection) better matched the user’s query and intent.   Majority vote was used as the ground truth.  To measure agreement between RF-Mem and human evaluators, we report the **Matthews Correlation Coefficient (MCC)**.
>
> ### 1.2 Findings
> 1. **Recollection retrieval is empirically meaningful.**
>    Among 25 samples, **9** were judged by humans to favor recollection over familiarity.
>
> 2. **RF-Mem’s routing aligns well with human preferences.**
>    RF-Mem achieved an **MCC = 0.4909**, indicating substantial agreement despite the small sample size.
>
> ### 1.3 Results
>
> |     |fami win|recol win|tie|
> |---------------|----------|------------|------|
> |**number**|6  |9    |10|
>
> **Matthews Correlation Coefficient**: $\text{MCC} = 0.4909$

---

> ### Author Response · Authors · 2025-11-19
> **Response to Reviewer QRxk [Part 2]**
>
> ## 2. The learning mehtod of the strategy selection mechanism
> > **Your Concern (W2):** Heuristic gating mechanism. The decision to enter recollection mode is based on manually chosen similarity / entropy thresholds. There’s no learned controller and no robustness study across domains. Furthermore, the authors do not detail why certain thresholds were selected, or how one should optimise these.
> >
> >**Your Concern (Q1):** How are the gating thresholds selected? Could the gate be learned?
>
> Thank you very much for this insightful comment. We fully agree that the thresholds in our current gate are manually set. This design choice is primarily motivated by `privacy considerations`: under data protection regulations such as GDPR, systems are generally prohibited from adaptively learning from a user’s personal memory corpus without explicit consent. In many real-world industry deployments, this means that a non-learned gating mechanism is not only operationally simpler but also the legally compliant option.
>
> As discussed in Appendix B, we performed a full sensitivity study over all gating-related hyperparameters (including the mean-score threshold, entropy threshold), and the results show that RF-Mem is generally stable across a broad range of settings.
>
> Nevertheless, inspired by your suggestion, we additionally designed and trained a lightweight MLP classifier that takes probe retrieval statistics (mean similarity, entropy) as input and predicts whether to route to familiarity or recollection. This learned gate exhibits robustness, and its generalization benefits grow naturally with increasing training data. **This experiment have added in the Appendix D.6 in the revised version.**
>
> ### 2.1 Experiment setup
> Motivated by your suggestion, we conducted a new experiment to evaluate whether the gate can be learned from data.
>
> **Traning Data.** We used the public PersonaMem-32k (589 samples) and PersonaMem-128k (2727 samples) subsets. For 32k, we used the first 400 samples for training and the remaining 189 for testing; for 128k, the first 2000 samples for training and the remaining 727 for testing. We formed a binary classification dataset using only cases where one path clearly wins (ties removed).
>
> **Training Details.** Each sample is represented by the two gate signals, mean similarity s and entropy H(p), and the label 0/1 indicates whether Familiarity or Recollection is better, respectively. We used an 80%–20% train–validation split and accuracy as the metric. Each experiment was repeated 10 times with different random seeds.
>
> ### 2.2 Findings
> 1. **The effectiveness of the auto-tuning mechanism grows with sample sizes.**
>    With limited supervision (32k), the learned gate is statistically unstable and may even underperform a simple “always recollect’’ baseline. As the training data increases (128k), the gate becomes noticeably more stable and its performance steadily improves, approaching—but not surpassing—the hand-tuned parameters.
>
> 2. **The mean similarity $\bar{s}$ and entropy $H(p)$ are proper signals.**
>    The auto-tuning results validate our choice of using mean similarity $\bar{s}$ and entropy $H(p)$ as the two gating signals: even a simple classifier using only these two features achieves competitive performance across data scales. This indicates that our hand-tuned parameters are not ad hoc but capture a meaningful decision boundary in the data.
>
> ### 2.3 Results
> **PersonaMem-32k**: First 400 samples for training, last 189 for testing
>
> |Training Size|Famili. win|Recol. win|Tie|
> |-|-|-|-|
> |75=(34+41)|34|41|325|
>
> |Method|Overall|Revisit Reasons|Shared Facts|Track Evolution|Aligned Recs|Latest Prefs|New Scenarios|New Ideas|Num of Famili.|Num of Recol.|
> |-|-|-|-|-|-|-|-|-|-|-|
> |Famili.|0.6296|0.9688|0.7297|0.6327|0.8125|0.2857|0.5385|0.2286|189|0|
> |Recol.|$\underline{\text{0.6667}}$|0.9688|0.7297|0.7551|0.8750|0.1429|0.7297|0.2286|0|189|
> |RF-Mem (hand gate)|**0.6720**|1.0000|0.7838|0.6939|0.8750|0.1429|0.6154|0.2571|83|106|
> |RF-Mem (learned gate)|0.6482±0.0067|0.9688±0.0000|0.7514±0.0114|0.6878±0.0255|0.8750±0.0000|0.1429±0.0000|0.5385±0.0000|0.2286±0.0000|97.5±6.7|91.5±6.7|
>
>
> **PersonaMem-128k**: First 2000 samples for training, last 727 for testing
>
> |Training Size|Famili. win|Recol. win|Tie|
> |-|-|-|-|
> |363=(178+185)|178|185|1637|
>
> |Method|overall|Revisit Reasons|Shared Facts|Track Evolution|Aligned Recs|Latest Prefs|New Scenarios|New Ideas|Num of Famili.|Num of Recol.|
> |-|-|-|-|-|-|-|-|-|-|-|
> |Famili.|0.5131|0.7742|0.6410|0.6162|0.5176|0.5122|0.3529|0.3043|727|0|
> |Recol.|0.5158|0.7634|0.5897|0.6162|0.5647|0.5366|0.3824|0.2609|0|727|
> |RF-Mem (hand gate)|**0.5199**|0.7527|0.6410|0.6162|0.5059|0.5463|0.3824|0.2971|402|325|
> |RF-Mem (learned gate)|$\underline{\text{0.5175±0.0039}}$|0.7591±0.0055|0.6180±0.0255|0.6091±0.0048|0.5553±0.0182|0.5400±0.0108|0.3794±0.0152|0.2718±0.0070|334.4±28.3|392.6±28.3|

---

> ### Author Response · Authors · 2025-11-19
> **Response to Reviewer QRxk [Part 3]**
>
> ## 3. RF-Mem under stronger LLM-based query expansion and interactive retrieval baselines
> >**Your Concern (W3):** Baselines. Comparisons are mostly against: full context, dense top-K, and recollection-only. Stronger long-term memory agents (e.g. reflective / hierarchical memory systems) aren’t fully represented.
>
> Thank you very much for raising this important point. We agree that positioning RF-Mem against more advanced LLM-based retrieval frameworks is necessary for a comprehensive evaluation. Conceptually, RF-Mem operates **solely at the retrieval stage** and is orthogonal to query expansion,or iterative retrieval. In principle, RF-Mem can be inserted into these pipelines as a *retriever-side controller*, and we have demonstrated one such adaptation on MemoryBank (Seciton 3.4) in the main paper. For further validation, we have now conducted additional experiments integrating RF-Mem with two representative LLM-based retrieval frameworks: (1) HyDE [3] and (2) Search-o1 [4]. **This experiment have added in the Section 3.4.2 and 3.4.3 in the revised version.**
>
> ### 3.1 Experiment setup
>
> 1. **LLM-based query expansion (HyDE, ACL 2023, CMU)[3]**
>    We adopt HyDE as a representative query expansion baseline.
>    Following the standard protocol for retrieval-oriented methods, we evaluate Recall on PersonaBench using multi-qa-MiniLM-L6-cos-v1.
>
> 2. **LLM-based iterative retrieval (Search-o1, EMNLP 2025, RUC)[4]**
>    We adopt Search-o1 as a representative iterative multi-step retrieval baseline, which implements a comparable multi-step retrieval process. Following standard practice for iterative methods, we evaluate generation accuracy on PersonaMem.
>
>
> ### 3.2 Findings
> 1. **RF-Mem is compatible with LLM-based query expansion.**
>    Adding RF-Mem on top of HyDE consistently improves R@5 and R@10. Notably, HyDE alone underperforms the base query, which we conjecture is due to the personalized nature of this task, where generic query expansions may dilute user-specific signals.
> 2. **RF-Mem is also compatible with iterative LLM-based retrieval.**
>    Integrated into Search-o1, RF-Mem further improves personalized generation accuracy across multiple categories. Interestingly, Search-o1 alone underperforms the base RAG setting, which we conjecture is due to the personalized nature of the task, where the iteratively generated queries may fail to stay aligned with the user’s memory or may amplify retrieval errors over multiple steps.
>
> These results confirm that RF-Mem complements query expansion and LLM-based iterative retrieval rather than competing with them, and that its benefits extend beyond standard dense retrieval.
>
> ### 3.3 Results
> **HyDE (LLM-based query expansion) on PersonaBench**
>
> |Method|R@5 Overall|Basic Info|Social Info|Pref Easy|Pref Hard|R@10 Overall|Basic Info|Social Info|Pref Easy|Pref Hard|
> |-|-|-|-|-|-|-|-|-|-|-|
> |Famili.|0.4484|0.4515|0.4852|0.4904|0.3659|0.5964|0.5879|0.6220|0.6442|0.5561|
> |Recol.|0.4491|0.4379|0.4903|0.5128|0.3854|0.6062|0.5924|0.6859|0.5659|0.6267|
> |**RF-Mem**|**0.4701**|0.4788|0.5091|0.4872|0.3854|**0.6071**|0.5924|0.6799|0.5707|0.6267|
> ||||||||||
> |HyDE + Famili.|0.3464|0.3106|0.3909|0.4615|0.3122|0.5120|0.5000|0.4991|0.5737|0.5220|
> |HyDE + Recol.|0.3482|0.3015|0.4135|0.4615|0.3171|0.5046|0.4909|0.5028|0.5929|0.4878|
> |**HyDE + RF-Mem**|**0.3504**|0.3061|0.4135|0.4615|0.3171|**0.5194**|0.5091|0.5028|0.5929|0.5220|
>
>
> **Search-o1 (LLM-based iterative retrieval) on PersonaMem**
>
> |Method|overall|Revisit Reasons|Shared Facts|Track Evolution|Aligned Recs|Latest Prefs|New Scenarios|New Ideas|
> |-|-|-|-|-|-|-|-|-|
> |RAG-Famili. (Dense)|0.5908|0.9091|0.5426|0.6475|0.6364|0.6471|0.5614|0.2151|
> |RAG-Recol.|0.6214|0.9495|0.5194|0.6547|0.7818|0.7059|0.5965|0.2688|
> |**RAG-RF-Mem**|**0.6350**|0.9495|0.5659|0.6619|0.7818|0.7059|0.6140|0.2688|
> |Search-o1 + Famili. (Dense)|0.5823|0.8687|0.5271|0.6259|0.6545|0.5882|0.5614|0.2581|
> |Search-o1 + Recol.|0.6010|0.8990|0.4961|0.6619|0.7091|0.6471|0.5789|0.2796|
> |**Search-o1 + RF-Mem**|**0.6146**|0.9293|0.5349|0.6978|0.7455|0.6471|0.5789|0.2043|

---

> ### Author Response · Authors · 2025-11-19
> **Response to Reviewer QRxk [Part 4]**
>
> ## 4. Privacy Concern
> >**Your Concern (Q2):** Safety/privacy. Since RF-Mem is explicitly better at surfacing detailed personal history, it can also resurface sensitive information the user didn’t mean to bring into the current turn. The authors acknowledge this risk but do not quantify or mitigate it.
>
> We sincerely appreciate you for raising this important privacy concern. While RF-Mem itself operates on static, pre-constructed embedding indices, we agree that real-world deployments should incorporate additional safeguards, **we have added the description in the Ethics Statement section**.
>
> >We acknowledge that real-world deployments of preference-based systems may inadvertently over-amplify user traits or reinforce behavioral biases. Additionally, improper handling of user histories could expose sensitive information or lead to unintended profiling effects, underscoring the need for careful governance.
>
> **And we have added the description in the Limitation section.**
> >Moreover, because RF-Mem improves the surfacing of long-term user history, downstream deployments should include safeguards to prevent unintended resurfacing of sensitive information, a direction we identify as important future work for quantifying and mitigating such risks.
>
>
> ---
>
> ## 5. Failure Case Analysis
> >**Your Concern (Q2):** Can you show qualitative failure cases where the gate picked the wrong mode?
>
> In response to your comment of show qualitative failure case, **we have added a filure case in the Appendix D.7.2.**
>
> > To analysis the failure case, we again compare one-shot Familiarity retrieval with the multi-round Recollection process on the PersonaMem dataset. As shown in Figure 19, we show the retrieval result of dual paths for the query *''I'm considering developing a tool to manage finances while traveling. How could I ensure it helps me prioritize experiences like local cuisine or guided tours?’’*. Familiarity retrieval outperforms Recollection in serving the user’s intent. The one-shot Familiarity retriever returns a broad and query-aligned set of memories, many of which explicitly mention 'local experiences', 'finance management apps', or `last trip'. These results directly address the user’s need to “prioritize experiences while traveling,” illustrating that high-scoring surface cues can sometimes be sufficient for intent coverage.
> >
> > In contrast, the multi-round Recollection process drifts along a different semantic trajectory. Although its iterative clustering-and-refinement mechanism successfully produces deeper and more structured traces, the retrieved branches gradually converge toward 'long-term financial habits' and 'personal finance management'. This indicates that the recollection trajectory can overfit to dominant semantic clusters in the memory corpus, especially when several high-density branches are thematically coherent but misaligned with the user’s immediate intent. As a result, the recollection path becomes increasingly anchored in financial-management narratives, overlooking the travel-related aspects of the query.
> >
> > Despite being a bad case for retrieval accuracy, this example is instructive. It exposes a core trade-off of deliberative recollection: the mechanism excels at reconstructing temporally dispersed, conceptually rich memory chains, yet it may over-emphasize internal coherence at the cost of task-oriented alignment. This case highlights the importance of balancing depth with intent fidelity, and underscores the need for more precise familiarity-uncertainty–guided strategy selection in future work.
> ---
>
>
> We hope these responses address your concerns comprehensively. Feel free to let us know if you have any additional questions, and we are more than happy to provide further clarification on any aspect of our work.
>
> *Respectfully and sincerely,*
>
> **The Authors**
>
> [1] Hsieh, Ya-Ping, et al. "Riemannian stochastic optimization methods avoid strict saddle points." Advances in Neural Information Processing Systems (2023)
>
> [2] Kasai, Hiroyuki, Pratik Jawanpuria, and Bamdev Mishra. "Riemannian adaptive stochastic gradient algorithms on matrix manifolds." International conference on machine learning.
>
> [3] Luyu Gao, Xueguang Ma, Jimmy Lin, and Jamie Callan. 2023. Precise Zero-Shot Dense Retrieval without Relevance Labels. In Proceedings of the 61st Annual Meeting of the Association for Computational Linguistics
>
> [4] Xiaoxi Li, Guanting Dong, Jiajie Jin, Yuyao Zhang, Yujia Zhou, Yutao Zhu, Peitian Zhang, and Zhicheng Dou. 2025. Search-o1: Agentic Search-Enhanced Large Reasoning Models. In Proceedings of the 2025 Conference on Empirical Methods in Natural Language Processing,
>
> [5] Li, Hao, et al. "Hello again! llm-powered personalized agent for long-term dialogue." Proceedings of the 2025 Conference of the Nations of the Americas Chapter of the Association for Computational Linguistics: Human Language Technologies

---

> > ### Comment · Reviewer_QRxk · 2025-11-27
> > **Response to Authors**
> >
> > Dear authors,
> >
> > Thank you very much for your detailed and thorough response. I think that these additional experiments do augment the paper very nicely. Your experiments showing that this your method can be used in a drop-in manner to improve the retrieval mechanism for more complex pipelines is a particularly interesting result. I'm happy to maintain my positive score.

---

> > > ### Author Response · Authors · 2025-11-27
> > > **Appreciation for Your Thoughtful Reassessment**
> > >
> > > Dear Reviewer QRxk,
> > >
> > > Thank you for your positive feedback and for updating us on the score. We genuinely appreciate your insightful and constructive comments. We truly appreciate your thoughtful reconsideration, as your engagement has greatly enhanced the quality of our work.
> > >
> > > Thank you again for your valuable feedback.
> > >
> > > *Respectfully and sincerely,*
> > >
> > > **The Authors**

---

### Official Review · Reviewer_jKrT · 2025-11-01

**Soundness:** 3
**Presentation:** 3
**Contribution:** 3
**Rating:** 6
**Confidence:** 4

**Summary:**

This paper introduces RF-Mem (Recollection-Familiarity Memory Retrieval), a new framework for personalizing Large Language Models (LLMs) by improving how they retrieve user-specific memory. The authors argue that current methods for memory retrieval are inefficient, either by overloading the LLM with the user's entire history (which is unscalable) or by using a simple, one-shot similarity search that only captures surface-level matches. Inspired by cognitive science, RF-Mem is based on the Recollection-Familiarity Dual-Process Theory of human memory. Experiments on three benchmarks show that RF-Mem consistently outperforms both one-shot retrieval and full-context methods, achieving higher accuracy and recall while maintaining low latency.

**Strengths:**

1. The "Recollection" path is a novel iterative mechanism that uses KMeans clustering and $\alpha$-mixing to reconstruct evidence chains purely in the embedding space, without requiring a costly LLM-in-the-loop for query reformulation.
2. The paper includes a full suite of supporting analyses in the appendix. This includes detailed sensitivity studies for all key hyperparameters ($\alpha$, $\tau$, B, F, K), a compelling qualitative case study that visually demonstrates the method's superiority,  and a formal theoretical analysis that mathematically grounds the risk-minimizing properties of the switching mechanism.

**Weaknesses:**

1. The primary experimental baseline ("Dense Retrieval") is a non-iterative, one-shot method. This is a weak baseline, as many advanced RAG systems already use multi-step or iterative retrieval.
2. This adaptive switch mechanism is heuristic and lacks robustness. The optimal thresholds are highly dependent on the specific embedding model used and the score distribution of the dataset.

**Questions:**

1. To improve robustness and reduce tuning overhead, have the authors considered a learned switching policy? A small, lightweight classifier trained on the probe retrieval's features (mean score, entropy, etc.) might provide a more generalizable switch than fixed thresholds.

---

> ### Author Response · Authors · 2025-11-19
> **Response to Reviewer jKrT [Part 1]**
>
> **Dear Reviewer jKrT**,
>
> Thank you very much for your thoughtful and constructive review. We sincerely appreciate your careful reading of our submission and your recognition of the conceptual coherence, theoretical grounding, and empirical depth of RF-Mem. Your summary reflects a deep understanding of our cognitive-science motivation, the embedding-space recollection design, and the comprehensive analyses provided in the appendix. Your feedback on limitations, robustness, and baseline choices is highly insightful and has guided several meaningful improvements in the revised version. Below, we address each of your concerns in detail and explain how we have incorporated your suggestions.
>
> ## 1. RF-Mem under stronger LLM-based query expansion and iterative retrieval baselines
> >**Your Concern (W1):** The primary experimental baseline ("Dense Retrieval") is a non-iterative, one-shot method. This is a weak baseline, as many advanced RAG systems already use multi-step or iterative retrieval.
>
> Thank you very much for raising this important point. We agree that positioning RF-Mem against more advanced LLM-based retrieval frameworks is necessary for a comprehensive evaluation. Conceptually, RF-Mem operates **solely at the retrieval stage** and is orthogonal to query expansion,or iterative retrieval. In principle, RF-Mem can be inserted into these pipelines as a *retriever-side controller*, and we have demonstrated one such adaptation on MemoryBank (Seciton 3.4) in the main paper. For further validation, we have now conducted additional experiments integrating RF-Mem with two representative LLM-based retrieval frameworks: (1) HyDE [1] and (2) Search-o1 [2]. **This experiment have added in the Section 3.4.2 and 3.4.3 in the revised version.**
>
> ### 1.1 Experiment setup
>
> 1. **LLM-based query expansion (HyDE, ACL 2023, CMU)[1]**
>    We adopt HyDE as a representative query expansion baseline.
>    Following the standard protocol for retrieval-oriented methods, we evaluate Recall on PersonaBench using multi-qa-MiniLM-L6-cos-v1.
>
> 2. **LLM-based iterative retrieval (Search-o1, EMNLP 2025, RUC)[2]**
>    We adopt Search-o1 as a representative iterative multi-step retrieval baseline, which implements a comparable multi-step retrieval process. Following standard practice for iterative methods, we evaluate generation accuracy on PersonaMem.
>
>
> ### 1.2 Findings
> 1. **RF-Mem is compatible with LLM-based query expansion.**
>    Adding RF-Mem on top of HyDE consistently improves R@5 and R@10. Notably, HyDE alone underperforms the base query, which we conjecture is due to the personalized nature of this task, where generic query expansions may dilute user-specific signals.
> 2. **RF-Mem is also compatible with iterative LLM-based retrieval.**
>    Integrated into Search-o1, RF-Mem further improves personalized generation accuracy across multiple categories. Interestingly, Search-o1 alone underperforms the base RAG setting, which we conjecture is due to the personalized nature of the task, where the iteratively generated queries may fail to stay aligned with the user’s memory or may amplify retrieval errors over multiple steps.
>
> These results confirm that RF-Mem complements query expansion and LLM-based iterative retrieval rather than competing with them, and that its benefits extend beyond standard dense retrieval.
>
> ### 1.3 Results
>
> **HyDE (LLM-based query expansion) on PersonaBench**
>
> |Method|R@5 Overall|Basic Info|Social Info|Pref Easy|Pref Hard|R@10 Overall|Basic Info|Social Info|Pref Easy|Pref Hard|
> |-|-|-|-|-|-|-|-|-|-|-|
> |Famili.|0.4484|0.4515|0.4852|0.4904|0.3659|0.5964|0.5879|0.6220|0.6442|0.5561|
> |Recol.|0.4491|0.4379|0.4903|0.5128|0.3854|0.6062|0.5924|0.6859|0.5659|0.6267|
> |**RF-Mem**|**0.4701**|0.4788|0.5091|0.4872|0.3854|**0.6071**|0.5924|0.6799|0.5707|0.6267|
> ||||||||||
> |HyDE + Famili.|0.3464|0.3106|0.3909|0.4615|0.3122|0.5120|0.5000|0.4991|0.5737|0.5220|
> |HyDE + Recol.|0.3482|0.3015|0.4135|0.4615|0.3171|0.5046|0.4909|0.5028|0.5929|0.4878|
> |**HyDE + RF-Mem**|**0.3504**|0.3061|0.4135|0.4615|0.3171|**0.5194**|0.5091|0.5028|0.5929|0.5220|
>
>
> **Search-o1 (LLM-based iterative retrieval) on PersonaMem**
>
> |Method|overall|Revisit Reasons|Shared Facts|Track Evolution|Aligned Recs|Latest Prefs|New Scenarios|New Ideas|
> |-|-|-|-|-|-|-|-|-|
> |RAG-Famili. (Dense)|0.5908|0.9091|0.5426|0.6475|0.6364|0.6471|0.5614|0.2151|
> |RAG-Recol.|0.6214|0.9495|0.5194|0.6547|0.7818|0.7059|0.5965|0.2688|
> |**RAG-RF-Mem**|**0.6350**|0.9495|0.5659|0.6619|0.7818|0.7059|0.6140|0.2688|
> |Search-o1 + Famili. (Dense)|0.5823|0.8687|0.5271|0.6259|0.6545|0.5882|0.5614|0.2581|
> |Search-o1 + Recol.|0.6010|0.8990|0.4961|0.6619|0.7091|0.6471|0.5789|0.2796|
> |**Search-o1 + RF-Mem**|**0.6146**|0.9293|0.5349|0.6978|0.7455|0.6471|0.5789|0.2043|

---

> ### Author Response · Authors · 2025-11-19
> **Response to Reviewer jKrT [Part 2]**
>
> ## 2. Robustness of the switching mechanism and a Learned seleciton inspired by your feedback
> >**Your Concern (W1):** This adaptive switch mechanism is heuristic and lacks robustness. The optimal thresholds are highly dependent on the specific embedding model used and the score distribution of the dataset.
> >
> >**Your Concern (Q1):** To improve robustness and reduce tuning overhead, have the authors considered a learned switching policy? A small, lightweight classifier trained on the probe retrieval's features (mean score, entropy, etc.) might provide a more generalizable switch than fixed thresholds.
>
> Thank you very much for this insightful comment. We fully agree that the adaptive switch mechanism in our current method are manually set and heuristic. This design choice is primarily motivated by `privacy considerations`: under data protection regulations such as GDPR, systems are generally prohibited from adaptively learning from a user’s personal memory corpus without explicit consent. In many real-world industry deployments, this means that a non-learned gating mechanism is not only operationally simpler but also the legally compliant option.
>
> However, inspired by your suggestion, we additionally designed and trained a lightweight MLP classifier that takes probe retrieval statistics (mean similarity, entropy) as input and predicts whether to route to familiarity or recollection. This learned gate exhibits robustness, and its generalization benefits grow naturally with increasing training data. **This experiment have added in the Appendix D.6 in the revised version.**
>
> ### 2.1 Experiment setup
> Motivated by your suggestion, we conducted a new experiment to evaluate whether the gate can be learned from data.
>
> **Traning Data.** We used the public PersonaMem-32k (589 samples) and PersonaMem-128k (2727 samples) subsets. For 32k, we used the first 400 samples for training and the remaining 189 for testing; for 128k, the first 2000 samples for training and the remaining 727 for testing. We formed a binary classification dataset using only cases where one path clearly wins (ties removed).
>
> **Training Details.** Each sample is represented by the two gate signals, mean similarity s and entropy H(p), and the label 0/1 indicates whether Familiarity or Recollection is better, respectively. We used an 80%–20% train–validation split and accuracy as the metric. Each experiment was repeated 10 times with different random seeds.
>
> ### 2.2 Findings
> 1. **The effectiveness of the auto-tuning mechanism grows with sample sizes..**
>    With limited supervision (32k), the learned gate is statistically unstable and may even underperform a simple “always recollect’’ baseline. As the training data increases (128k), the gate becomes noticeably more stable and its performance steadily improves, approaching—but not surpassing—the hand-tuned parameters.
>
> 2. **The mean similarity $\bar{s}$ and entropy $H(p)$ are proper signals.**
>    The auto-tuning results validate our choice of using mean similarity $\bar{s}$ and entropy $H(p)$ as the two gating signals: even a simple classifier using only these two features achieves competitive performance across data scales. This indicates that our hand-tuned parameters are not ad hoc but capture a meaningful decision boundary in the data.
>
> ### 2.3 Results
> **PersonaMem-32k**: First 400 samples for training, last 189 for testing
>
> |Training Size|Famili. win|Recol. win|Tie|
> |-|-|-|-|
> |75=(34+41)|34|41|325|
>
> |Method|Overall|Revisit Reasons|Shared Facts|Track Evolution|Aligned Recs|Latest Prefs|New Scenarios|New Ideas|Num of Famili.|Num of Recol.|
> |-|-|-|-|-|-|-|-|-|-|-|
> |Famili.|0.6296|0.9688|0.7297|0.6327|0.8125|0.2857|0.5385|0.2286|189|0|
> |Recol.|$\underline{\text{0.6667}}$|0.9688|0.7297|0.7551|0.8750|0.1429|0.7297|0.2286|0|189|
> |RF-Mem (hand gate)|**0.6720**|1.0000|0.7838|0.6939|0.8750|0.1429|0.6154|0.2571|83|106|
> |RF-Mem (learned gate)|0.6482±0.0067|0.9688±0.0000|0.7514±0.0114|0.6878±0.0255|0.8750±0.0000|0.1429±0.0000|0.5385±0.0000|0.2286±0.0000|97.5±6.7|91.5±6.7|
>
>
> **PersonaMem-128k**: First 2000 samples for training, last 727 for testing
>
> |Training Size|Famili. win|Recol. win|Tie|
> |-|-|-|-|
> |363=(178+185)|178|185|1637|
>
> |Method|overall|Revisit Reasons|Shared Facts|Track Evolution|Aligned Recs|Latest Prefs|New Scenarios|New Ideas|Num of Famili.|Num of Recol.|
> |-|-|-|-|-|-|-|-|-|-|-|
> |Famili.|0.5131|0.7742|0.6410|0.6162|0.5176|0.5122|0.3529|0.3043|727|0|
> |Recol.|0.5158|0.7634|0.5897|0.6162|0.5647|0.5366|0.3824|0.2609|0|727|
> |RF-Mem (hand gate)|**0.5199**|0.7527|0.6410|0.6162|0.5059|0.5463|0.3824|0.2971|402|325|
> |RF-Mem (learned gate)|$\underline{\text{0.5175±0.0039}}$|0.7591±0.0055|0.6180±0.0255|0.6091±0.0048|0.5553±0.0182|0.5400±0.0108|0.3794±0.0152|0.2718±0.0070|334.4±28.3|392.6±28.3|

---

> ### Author Response · Authors · 2025-11-19
> **Response to Reviewer jKrT [Part 3]**
>
> We hope these responses address your concerns comprehensively. Feel free to let us know if you have any additional questions, and we are more than happy to provide further clarification on any aspect of our work.
>
> *Respectfully and sincerely,*
>
> **The Authors**
>
>
>
> [1] Luyu Gao, Xueguang Ma, Jimmy Lin, and Jamie Callan. 2023. Precise Zero-Shot Dense Retrieval without Relevance Labels. In Proceedings of the 61st Annual Meeting of the Association for Computational Linguistics (Volume 1: Long Papers), pages 1762–1777, Toronto, Canada. Association for Computational Linguistics.
>
> [2] Xiaoxi Li, Guanting Dong, Jiajie Jin, Yuyao Zhang, Yujia Zhou, Yutao Zhu, Peitian Zhang, and Zhicheng Dou. 2025. Search-o1: Agentic Search-Enhanced Large Reasoning Models. In Proceedings of the 2025 Conference on Empirical Methods in Natural Language Processing, pages 5420–5438, Suzhou, China. Association for Computational Linguistics.

---

### Official Review · Reviewer_52gJ · 2025-11-09

**Soundness:** 3
**Presentation:** 3
**Contribution:** 2
**Rating:** 4
**Confidence:** 5

**Summary:**

The paper proposes RF-Mem, a dual path that gates between fast “Familiarity” retrieval and slower “Recollection” based on mean similarity and an entropy signal.
When the gate flags uncertainty, it runs an in-embedding recollection loop that retrieves, clusters with KMeans, α-mixes cluster centroids with the query, and iterates under tight beam/fanout/round caps.
Experiments on personalized generation and retrieval tasks show a better accuracy–latency balance than one-shot dense retrieval or full-context prompting.

**Strengths:**

1. The cognitive grounding is crisp and the gate is concrete (mean score + entropy with a sharpness λ), so the controller is easy to reason about.

2. Recollection is lightweight and model-agnostic since it lives entirely in vector space using simple clustering and linear mixing.


3. Compute stays bounded because the loop exposes explicit knobs for beam width, fanout, and depth instead of open-ended expansion.

**Weaknesses:**

1. The many thresholds and weights feel hand-tuned, with little guidance for auto-tuning across users and domains.

2. KMeans centroids can be unstable on anisotropic or overlapping memory clusters, so α-mixing may drift the query off-intention.

3. Entropy over the top-K score simplex depends on λ and retriever scale, which could make the gate brittle when swapping encoders or normalizations.

4. The controller assumes unit-normalized cosine and a sub-Gaussian mean-similarity story, which may not hold in heavy-tailed similarity landscapes.

5. The latency write-up emphasizes averages, while real deployments care about tail spikes when recollection fans out.

**Questions:**

1. Would a density-based or spectral clustering help when memory clusters are non-spherical or imbalanced.



2. Does the gate remain calibrated when swapping embedding models or scaling memory to much larger corpora.



3. Can you add early-stop checks that detect “recollection loops” and cut branches before they waste budget.

---

> ### Author Response · Authors · 2025-11-19
> **Response to  Reviewer 52gJ [Part 1]**
>
> **Dear Reviewer 52gJ**,
>
> Thank you sincerely for the exceptionally thoughtful review. We are grateful for your clear summary of our work and for highlighting the strengths. Your comments show deep engagement with the paper, and we appreciate it. And we have carefully addressed each point in the response below.
>
> ## 1. The robustness of the strategy selection threshold choices
> >**Your Concern (w1):** The many thresholds and weights feel hand-tuned, with little guidance for auto-tuning across users and domains.
>
> Thank you for this insightful comment. We agree that the thresholds in our current gate are manually set. This design choice is primarily motivated by `privacy considerations`: under data protection regulations such as GDPR, systems are generally prohibited from adaptively learning from a user’s personal corpus. In many real-world industry deployments, this means that a non-learned gating mechanism is not only operationally simpler but also the legally compliant option.
>
> **As shown in Appendix D.2 (Sensitivity Study), our hand-crafted thresholds follow clear empirical guidance. We find the threshold $\\tau$ is consistently stable across datasets, and values in the range $0.25-0.30$ yield the strongest overall performance.** This stability suggests that the gate is not overly sensitive to hyperparameters and can be selected using a small validation split without per-user tuning.
>
> However, inspired by your suggestion, we additionally designed and trained a lightweight MLP classifier to predict whether to route to familiarity or recollection. This learned gate exhibits robustness, and its generalization benefits grow naturally with increasing training data. **This experiment have added in the Appendix D.6 in the revised version.**
>
> ### 1.1 Experiment setup
> Motivated by your suggestion, we conducted a new experiment to evaluate whether the gate can be learned from data.
>
> **Traning Data.** We used the public PersonaMem-32k (589 samples) and PersonaMem-128k (2727 samples) subsets. For 32k, we used the first 400 samples for training and the remaining 189 for testing; for 128k, the first 2000 samples for training and the remaining 727 for testing. We formed a binary classification dataset using only cases where one path clearly wins (ties removed).
>
> **Training Details.** Each sample is represented by the two gate signals, mean similarity $\\bar{s}$ and entropy $H(p)$, and the label 0/1 indicates whether Familiarity or Recollection is better, respectively. We used an 80%–20% train–validation split and accuracy as the metric. Each experiment was repeated 10 times with different random seeds.
>
> ### 1.2 Findings
> 1. **The effectiveness of the auto-tuning mechanism grows with sample sizes..**
>    With limited supervision (32k), the learned gate is statistically unstable and may even underperform a simple “always recollect’’ baseline. As the training data increases (128k), the gate becomes noticeably more stable and its performance steadily improves, approaching—but not surpassing—the hand-tuned parameters.
>
> 2. **The mean similarity $\\bar{s}$ and entropy $H(p)$ are proper signals.**
>    The auto-tuning results validate our choice of using mean similarity \(\bar{s}\) and entropy \(H(p)\) as the two gating signals: even a simple classifier using only these two features achieves competitive performance across data scales. This indicates that our hand-tuned parameters are not ad hoc but capture a meaningful decision boundary in the data.
>
> ### 1.3 Results
> **PersonaMem-32k**: First 400 samples for training, last 189 for testing
>
> |Training Size|Famili. win|Recol. win|Tie|
> |-|-|-|-|
> |75=(34+41)|34|41|325|
>
> |Method|Overall|Revisit Reasons|Shared Facts|Track Evolution|Aligned Recs|Latest Prefs|New Scenarios|New Ideas|Num of Famili.|Num of Recol.|
> |-|-|-|-|-|-|-|-|-|-|-|
> |Famili.|0.6296|0.9688|0.7297|0.6327|0.8125|0.2857|0.5385|0.2286|189|0|
> |Recol.|$\underline{\text{0.6667}}$|0.9688|0.7297|0.7551|0.8750|0.1429|0.7297|0.2286|0|189|
> |RF-Mem (hand gate)|**0.6720**|1.0000|0.7838|0.6939|0.8750|0.1429|0.6154|0.2571|83|106|
> |RF-Mem (learned gate)|0.6482±0.0067|0.9688±0.0000|0.7514±0.0114|0.6878±0.0255|0.8750±0.0000|0.1429±0.0000|0.5385±0.0000|0.2286±0.0000|97.5±6.7|91.5±6.7|
>
>
> **PersonaMem-128k**: First 2000 samples for training, last 727 for testing
>
> |Training Size|Famili. win|Recol. win|Tie|
> |-|-|-|-|
> |363=(178+185)|178|185|1637|
>
> |Method|overall|Revisit Reasons|Shared Facts|Track Evolution|Aligned Recs|Latest Prefs|New Scenarios|New Ideas|Num of Famili.|Num of Recol.|
> |-|-|-|-|-|-|-|-|-|-|-|
> |Famili.|0.5131|0.7742|0.6410|0.6162|0.5176|0.5122|0.3529|0.3043|727|0|
> |Recol.|0.5158|0.7634|0.5897|0.6162|0.5647|0.5366|0.3824|0.2609|0|727|
> |RF-Mem (hand gate)|**0.5199**|0.7527|0.6410|0.6162|0.5059|0.5463|0.3824|0.2971|402|325|
> |RF-Mem (learned gate)|$\underline{\text{0.5175±0.0039}}$|0.7591±0.0055|0.6180±0.0255|0.6091±0.0048|0.5553±0.0182|0.5400±0.0108|0.3794±0.0152|0.2718±0.0070|334.4±28.3|392.6±28.3|

---

> ### Author Response · Authors · 2025-11-19
> **Response to Reviewer 52gJ [Part 2]**
>
> ## 2. Stability of KMeans centroids in anisotropic or overlapping memory clusters
> >**Your Concern (w2):** KMeans centroids can be unstable on anisotropic or overlapping memory clusters, so α-mixing may drift the query off-intention.
> >
> >**Your Concern (Q1):** Would a density-based or spectral clustering help when memory clusters are non-spherical or imbalanced.
>
> Thank you very much for this constructive suggestion. We appreciate your concern that KMeans may behave sub-optimally on anisotropic or overlapping clusters, potentially affecting the stability of our α-mixing updates. To address this, we conducted additional experiments using alternative clustering methods (DBSCAN and Spectral). **Across all variants, α-mixing remained effective and consistently improved retrieval accuracy.** These results indicate that our update rule is not tied to the assumptions of KMeans and remains robust under different cluster geometries. **This experiment has been added in Appendix D.8 in the revised version.**
>
> ### 2.1 Experiment setup
>
> To rigorously evaluate whether the potential instability of KMeans on anisotropic or overlapping memory clusters affects our recollection module, we performed an additional set of experiments on the public PersonaBench retrieval benchmark. Our goal was to test whether alternative clustering strategies—especially those specifically designed for non-spherical or imbalanced clusters—would change the effectiveness of the recollection path.
>
> To this end, we replaced KMeans with two widely used and theoretically distinct clustering algorithms:
>
> - **DBSCAN [1]**: a density-based cluster method that handles arbitrary-shaped clusters and noise.
> - **Spectral Clustering [2]**: a spectral cluster method effective for non-spherical or imbalanced cluster structures.
>
> We kept all other components of the recollection pipeline unchanged (vector retrieval, entropy computation, α-mixing, and iteration caps) to isolate the effect of the clustering choice. All experiments were conducted under identical hyperparameter settings and evaluated on the same PersonaBench subsets to ensure comparability.
>
> ### 2.2 Findings
> 1. **Across all variants, α-mixing remained effective and consistently improved retrieval accuracy.**
>   Results showing that our recollection α-mixing updates path is robust even when KMeans is replaced by density-based or spectral methods.
> 2. **These results offer practical insights for deployment**.
>   Industrial systems may flexibly choose the clustering method that best matches their memory structure without changing the overall RF-Mem design.
>
> ### 2.3 Results
>
> **PersonaBench Clustering Experiments (Recall@5)**
>
> |Method|Overall|Basic Info|Social Info|Pref Easy|Pref Hard|
> |--------|---------|--|---|-|-|
> |Famili.|0.4484|0.4515|0.4852|0.4904|0.3659|
> |Recol.(KMeans)|0.4491|0.4379|0.4903|0.5128|0.3854|
> |RF-Mem (KMeans)|**0.4701**|0.4788|0.5091|0.4872|0.3854|
> |||||||
> |Recol.(DBSCAN)|0.4431|0.4424|0.4940|0.4872|0.3512|
> |RF-Mem (DBSCAN)|**0.4669**|0.4879|0.5129|0.4744|0.3463|
> |||||||
> |Recol.(Spectral)|0.4522|0.4591|0.4739|0.5000|0.3756|
> |RF-Mem (Spectral)|**0.4651**|0.4818|0.4928|0.4872|0.3707|
>
> **PersonaBench Clustering Experiments (Recall@10)**
>
> |Method|Overall|Basic Info|Social Info|Pref Easy|Pref Hard|
> |--------|---------|--|---|-|-|
> |fami|0.5964|0.5879|0.6220|0.6442|0.5561|
> |our recol|0.6062|0.5924|0.6859|0.5659|0.6267|
> |RF-Mem (our)|**0.6071**|0.5924|0.6799|0.5707|0.6267|
> |||||||
> |Recol.(DBSCAN)|0.6028|0.6045|0.6079|0.6891|0.5366|
> |RF-Mem (DBSCAN)|**0.6040**|0.5659|0.6079|0.6667|0.6015|
> |||||||
> |Recol.(Spectral)|0.5957|0.6045|0.5978|0.6474|0.5366|
> |RF-Mem (Spectral)|**0.6001**|0.6015|0.6041|0.6474|0.5610|

---

> ### Author Response · Authors · 2025-11-19
> **Response to Reviewer 52gJ [Part 3]**
>
> ## 3. Robustness of the familiarity uncertainty-based selection to encoder changes
> >**Your Concern (w3):** Entropy over the top-\(K\) score simplex depends on λ and retriever scale, which could make the gate brittle when swapping encoders or normalizations.
> >
> >**Your Concern (Q2):** Does the gate remain calibrated when swapping embedding models or scaling memory to much larger corpora.
>
> ### 3.1 Experiment setup
> Thank you very much for raising this concern. We fully agree that an uncertainty signal based on top-K score entropy must remain stable across different embedding models and scoring scales in order to be practical. To directly evaluate this, **we conducted the encoder-swap experiments reported in Table 3 of the main paper**. Specifically, we tested three widely used retrievers—**multi-qa-MiniLM-L6-cos-v1**, **all-mpnet-base-v2**, and **BAAI/bge-base-en-v1.5**—on both versions of the **LongMemEval** benchmark. **LongMemEval-S** contains an average of **50.22** memory sessions per query, while **LongMemEval-M** contains **501.90** sessions; full statistics are provided in Appendix A, Table 10. The gating parameters used in RF-Mem were hand-designed following the patterns illustrated in Figures 10 and 12.
>
> ### 3.2 Findings
> 1. **RF-Mem is robust across encoders.**
>    Across all three embedding models, the entropy-based uncertainty signal remains stable, and RF-Mem consistently outperforms both Familiarity and Recollection alone. This suggests that the entropy term does not become brittle under different normalization scales, cosine ranges, or embedding distributions.
>
> 2. **RF-Mem generalizes across memory lengths.**
>    The same gating design works for both the short-history (LongMemEval-S) and long-history (LongMemEval-M) settings, indicating that the entropy-based uncertainty signal scales well with the amount of available user memory.
>
> ### 3.3 Results
>
> **LongMemEval-S and LongMemEval-M (Encoder Swap Experiments)**
>
>
> |Retriever|Setting|recall_all@5 (S)|recall_all@10 (S)|recall_all@50 (S)|recall_all@5 (M)|recall_all@10 (M)|recall_all@50 (M)|
> |-|---------|----|-----|------|-----|------|------|
> |**multi-qa-MiniLM-L6-cos-v1**|Fami|0.7136|0.8282|0.9761|0.4177|0.5465|0.7518|
> ||Recol|0.7351|0.8425|1.0000|0.4368|0.5585|0.7590|
> ||**RF-Mem**|**0.7375**|**0.8473**|**1.0000**|**0.4391**|**0.5609**|**0.7613**|
> |**all-mpnet-base-v2**|Fami|0.7303|0.8353|0.9832|0.4176|0.5489|0.7637|
> ||Recol|0.7398|0.8305|0.9952|0.4386|0.5871|0.7422|
> ||**RF-Mem**|**0.7398**|**0.8377**|**0.9952**|**0.4391**|**0.5894**|**0.7684**|
> |**BAAI/bge-base-en-v1.5**|Fami|0.7924|0.8926|1.0000|0.4964|0.6611|0.8305|
> ||Recol|0.8162|0.9165|1.0000|0.5131|0.6635|0.8234|
> ||**RF-Mem**|**0.8186**|**0.9189**|**1.0000**|**0.5155**|**0.6635**|**0.8329**|
>
> ---
>
> ## 4. Empirical validation of the sub-Gaussian mean-similarity score assumption
> >**Your Concern (W4):** The controller assumes unit-normalized cosine and a sub-Gaussian mean-similarity story, which may not hold in heavy-tailed similarity landscapes.
>
> ### 4.1 Experiment setup
> Thank you very much for raising this point. We agree that heavy-tailed similarity distributions are common in some embedding models, and it is important to verify whether our sub-Gaussian assumption matches real retrieval landscapes. Motivated by your comment, we conducted an empirical analysis of the mean similarity distribution across three datasets used in our experiments: PersonaMem, PersonaBench, and LongMemEval. **This analysis have added in Figures 7, 9, 11, and 13.**
>
> ### 4.2 Findings
> 1. **Mean similarity distributions are light-tailed.**
>   `As shown in the updated Figures 7, 9, 11, and 13 in the revised paper`, the empirical distribution of the mean top-$K$ similarity score $\\bar{s}$ is consistently light-tailed, without the heavy-tail behavior suggested in the concern. Importantly, the tail characteristics remain stable as the memory corpus grows from small (PersonaBench) to medium (LongMemEval) to large (PersonaMem), indicating that the similarity landscape does not exhibit pathological tail expansion.
>
> To reflect this in the paper, we added the following clarification in Appendix B:
>
> > *“To support the theoretical assumption in Appendix F.3, we show the empirical distribution of the mean similarity score $\\bar{s}$ in Figure 7,9,11,13. All three datasets exhibit light-tailed, bounded distributions without heavy-tail behavior, and the tail shape remains stable as the corpus size grows. This empirically confirms that the similarity landscape does not display the heavy-tailed structure.”*

---

> ### Author Response · Authors · 2025-11-19
> **Response to Reviewer 52gJ [Part 4]**
>
> ## 5. Tail-latency behavior of the RF-Mem
> > **Your Concern (W5):** The latency write-up emphasizes averages, while real deployments care about tail spikes when recollection fans out.
>
> ### 5.1 Experiment setup
> Thank you very much for highlighting this important point. We fully agree that average latency alone is insufficient for real deployments, and that the tail distribution (p90, p95, p99) is the real bottleneck that determines user experience and system-level provisioning. Motivated by your suggestion, we conducted additional tail-latency measurements on the PersonaMem dataset across three memory scales (32k, 128k, and 1M). **Due to the time limit, the whole experiment will be added in the camera-ready version.**
>
> ### 5.2 Findings
> 1. **Tail spikes originate almost entirely from the recollection path.**
>    Across all dataset sizes, the p90–p99 latency of RF-Mem is nearly identical to that of pure Recollection, confirming that long-tail cases correspond to deliberate multi-step reconstruction rather than failures of the gating mechanism.
>
> 2. **RF-Mem significantly reduces mean latency while keeping tail latency no worse than Recollection.**
>    Across all scales, RF-Mem maintains the same tail-latency envelope as pure Recollection, while reducing **mean latency by 25–40%** depending on memory size. These results support the deployability of our controller: RF-Mem preserves bounded tail behavior while delivering substantial speedups for the majority of queries.
>
> ### 5.3 Results
> **Tail-latency statistics**
>
> |32k Retrieval|Mean|p90|p95|p99|
> |-|------|-----|-----|-----|
> |base|3.14ms|3.22ms|3.26ms|3.98ms|
> |recol|7.09ms|7.87ms|7.97ms|9.08ms|
> |RF-Mem|5.09ms|7.78ms|7.96ms|8.85ms|
>
> |128k Retrieval|Mean|p90|p95|p99|
> |--|------|-----|-----|-----|
> |base|3.24ms|3.41ms|3.49ms|4.24ms|
> |recol|7.86ms|8.16ms|8.73ms|9.60ms|
> |RF-Mem|4.27ms|8.15ms|8.31ms|9.44ms|
>
> |1M Retrieval|Mean|p90|p95|p99|
> |----|------|-----|-----|-----|
> |base|4.42ms|4.41ms|4.56ms|5.21ms|
> |recol|8.12ms|8.45ms|9.96ms|10.41ms|
> |RF-Mem|6.28ms|8.71ms|9.67ms|10.31ms|
>
>
> ---
>
> ## 6. Early-stop mechanisms for detecting and preventing recollection loops
>
> > **Your Concern (Q3):** Can you add early-stop checks that detect “recollection loops” and cut branches before they waste budget.
>
>
> ### 6.1 Experiment setup
> Thank you very much for this helpful suggestion. We agree that without early stopping, iterative recollection may continue exploring branches that contribute little to the final result. In fact, our current implementation already includes a global early-stop rule: once the total number of retrieved candidates reaches the expacted retrieval numbner \(K\), the recollection loop terminates. This design was introduced specifically to keep latency bounded.
>
> Your question also motivated an additional analysis: **what would happen if we removed the early-stop mechanism entirely?** To answer this, we conducted a controlled experiment on the PersonaBench dataset using the multi-qa-MiniLM-L6-cos-v1 retriever. **This experiment has been added in Appendix D.9 in the revised version.**
>
> ### 6.2 Findings
> 1. **Removing early-stop can bring accuracy gains**, since the recollection path explores more cluster branches.
> 2. **However, latency increases noticeably**, indicating that the early-stop rule helps maintain a favorable accuracy–efficiency tradeoff in practice.
>
> ### 6.3 Results
> **Results with and without early stopping**
>
> |Method|Time (R@5)|R@5 Overall|Basic Info|Social Info|Pref Easy|Pref Hard|Time (R@10)|R@10 Overall|Basic Info|Social Info|Pref Easy|Pref Hard|
> |--------|--|---|---|---|-|-|---|----|---|---|-|-|
> |Famili.|8.40ms|0.4484|0.4515|0.4852|0.4904|0.3659|13.68ms|0.5964|0.5879|0.6220|0.6442|0.5561|
> |Our Recol.|9.65ms|0.4491|0.4379|0.4903|0.5128|0.3854|17.29ms|0.6062|0.5924|0.6859|0.5659|0.6267|
> |**RF-Mem (our)**|9.16ms|**0.4701**|0.4788|0.5091|0.4872|0.3854|15.22ms|**0.6071**|0.5924|0.6799|0.5707|0.6267|
> |||||
> |Our Recol. (no early stop)|11.27ms|0.4537|0.4530|0.4827|0.5128|0.3805|21.15ms|0.6036|0.6015|0.6204|0.6763|0.5415|
> |**RF-Mem (no early stop)**|9.79ms|**0.4709**|0.4848|0.5016|0.5000|0.3756|17.46ms|**0.6083**|0.6015|0.6267|0.6667|0.5659|
>
> ---
>
> We hope these responses address your concerns comprehensively. Feel free to let us know if you have any additional questions, and we are more than happy to provide further clarification on any aspect of our work.
>
> *Respectfully and sincerely,*
>
> **The Authors**
>
> [1] Ester M, Kriegel H P, Sander J, et al. A density-based algorithm for discovering clusters in large spatial databases with noise[C]//kdd. 1996, 96(34): 226-231.
>
> [2] Ng A, Jordan M, Weiss Y. On spectral clustering: Analysis and an algorithm[J]. Advances in neural information processing systems, 2001, 14.

---

> > ### Comment · Reviewer_52gJ · 2025-11-24
> >
> > Thanks for your detailed rebuttals. Most of my concerns have been addressed, so I update my score to 6. Good luck.

---

> ### Author Response · Authors · 2025-11-24
> **Grateful for Feedback and Open to Further Clarifications**
>
> Dear Reviewer 52gJ,
>
> Thank you very much for your time, for your thoughtful engagement with our work, and for the score update to 6. We are genuinely grateful that your constructive suggestions helped us sharpen the contribution and articulate the core ideas more clearly.
>
> If there are any remaining points that you feel could benefit from further clarification, we would be more than happy to elaborate. Please feel free to let us know.
>
> Thank you again for your valuable feedback.
>
> *Respectfully and sincerely,*
>
> **The Authors**

---

### Official Review · Reviewer_pdxs · 2025-11-10

**Soundness:** 3
**Presentation:** 3
**Contribution:** 3
**Rating:** 6
**Confidence:** 3

**Summary:**

The paper “Evoking User Memory: Personalizing LLM via Recollection–Familiarity Adaptive Retrieval” proposes RF-Mem, a dual-process memory retrieval framework for personalized large language models. Inspired by cognitive science, the model integrates two complementary retrieval modes—familiarity, for fast and confident recognition, and recollection, for slower and more deliberate evidence reconstruction. RF-Mem measures familiarity using the mean similarity score and entropy, dynamically switching between the two paths based on uncertainty. When familiarity is high, it performs efficient one-shot retrieval; when uncertainty increases, it activates a clustering-based recollection process that iteratively refines queries through an α-mixing strategy in embedding space. This design allows the system to balance efficiency with depth, maintaining the low latency of one-shot retrieval for straightforward queries while engaging in more thorough evidence gathering for ambiguous ones. Experiments on several benchmarks demonstrate that RF-Mem outperforms traditional one-shot and full-context methods in both accuracy and efficiency, achieving scalable, adaptive, and human-like personalization for LLMs.

**Strengths:**

This paper presents a highly original and conceptually grounded contribution by introducing RF-Mem, a dual-process memory retrieval framework inspired by the Recollection–Familiarity theory in cognitive science. The idea of aligning LLM personalization with human memory processes is both novel and intellectually appealing, extending beyond conventional retrieval-augmented generation paradigms. Methodologically, the paper is well executed: the formulation of familiarity uncertainty through mean similarity and entropy is simple yet effective, and the recollection mechanism—implemented via clustering and α-mixing—offers a practical and computationally efficient way to approximate deliberate evidence reconstruction in embedding space. The experiments are comprehensive, covering multiple personalized benchmarks and retriever architectures, and the empirical results convincingly support the method’s claimed benefits in both accuracy and efficiency. The paper is also clearly written and well-structured, maintaining strong conceptual coherence between cognitive theory and technical implementation.

**Weaknesses:**

Despite its conceptual elegance, the paper’s empirical validation should be further strengthened.
First, the experimental comparisons are restricted to retrieval-only baselines, omitting stronger LLM-based retrieval or reasoning systems such as query rewriting, iterative retrieval (e.g., Search-Mem), or graph-based methods. Including such more advanced baselines would better position RF-Mem within current state-of-the-art approaches and clarify whether its advantages hold beyond standard dense retrieval.
Second, the paper lacks an ablation study of the gating mechanism. It is unclear how much the mean similarity score and entropy respectively contribute to routing decisions, and how often each retrieval path is activated.
Third, the definition of uncertainty is limited to similarity score distributions, which may fail to capture semantic ambiguity or misleading high-confidence matches; this limitation should be acknowledged and discussed.

**Questions:**

1. Could adaptive or learned gating mechanisms outperform the hand-tuned parameters used here?
2.The recollection process relies on clustering in embedding space. Have the authors tested alternative approaches, such as graph-based retrieval or semantic expansion via LLM-generated paraphrases, to validate the robustness of the proposed method?
3.As the field increasingly uses LLMs for complex retrieval tasks, can you demonstrate RF-Mem's advantage over a baseline that uses a powerful LLM  for a single step of query expansion or decomposition before a dense retrieval step?
4.The entire framework, including the uncertainty estimation, relies heavily on the quality of the underlying embedding model.  Did you observe significant performance variance across the three retrievers (MiniLM, MPNet, BGE) in terms of the routing accuracy?  Is there a risk that a poorly calibrated embedding model could lead to a cascading failure in the gating mechanism?

---

> ### Author Response · Authors · 2025-11-19
> **Response to Reviewer pdxs [Part 1]**
>
> **Dear Reviewer pdxs**,
>
> Thank you for your thoughtful and generous review. We are truly grateful for the time and care you invested in evaluating our work. We are deeply thankful for the constructive weaknesses and questions you raised; they have significantly helped us improve both the empirical validation and the clarity of our presentation.
>
> ## 1. RF-Mem under stronger LLM-based retrieval and reasoning baselines
> >**Your Concern (W1):** First, the experimental comparisons are restricted to retrieval-only baselines, omitting stronger LLM-based retrieval or reasoning systems such as query rewriting, iterative retrieval (e.g., Search-Mem), or graph-based methods. Including such more advanced baselines would better position RF-Mem within current state-of-the-art approaches and clarify whether its advantages hold beyond standard dense retrieval.
> >
> >**Your Concern (Q3):** As the field increasingly uses LLMs for complex retrieval tasks, can you demonstrate RF-Mem's advantage over a baseline that uses a powerful LLM for a single step of query expansion or decomposition before a dense retrieval step?
>
> Thank you very much for raising this important point. We agree that positioning RF-Mem against more advanced LLM-based retrieval frameworks is necessary for a comprehensive evaluation. Conceptually, RF-Mem operates **solely at the retrieval stage** and is orthogonal to query expansion,or iterative retrieval. In principle, RF-Mem can be inserted into these pipelines as a *retriever-side controller*, and we have demonstrated one such adaptation on MemoryBank (Seciton 3.4) in the main paper. For further validation, we have now conducted additional experiments integrating RF-Mem with two representative LLM-based retrieval frameworks: (1) HyDE [1] and (2) Search-o1 [2]. **This experiment have added in the Section 3.4.2 and 3.4.3 in the revised version.**
>
> ### 1.1 Experiment setup
>
> 1. **LLM-based query expansion (HyDE, ACL 2023, CMU)[1]**
>    We adopt HyDE as a representative query expansion baseline.
>    Following the standard protocol for retrieval-oriented methods, we evaluate Recall on PersonaBench using multi-qa-MiniLM-L6-cos-v1.
>
> 2. **LLM-based iterative retrieval (Search-o1, EMNLP 2025, RUC)[2]**
>    We adopt Search-o1 as a representative iterative multi-step retrieval baseline.
>    While we were unable to locate a system explicitly named “Search-Mem,” we believe your comment refers to memory-augmented iterative retrieval frameworks such as Search-o1, which implements a comparable multi-step retrieval process. Following standard practice for iterative methods, we evaluate generation accuracy on PersonaMem.
>
>
> ### 1.2 Findings
> 1. **RF-Mem is compatible with LLM-based query expansion.**
>    Adding RF-Mem on top of HyDE consistently improves R@5 and R@10. Notably, HyDE alone underperforms the base query, which we conjecture is due to the personalized nature of this task, where generic query expansions may dilute user-specific signals.
> 2. **RF-Mem is also compatible with iterative LLM-based retrieval.**
>    Integrated into Search-o1, RF-Mem further improves personalized generation accuracy across multiple categories. Interestingly, Search-o1 alone underperforms the base RAG setting, which we conjecture is due to the personalized nature of the task, where the iteratively generated queries may fail to stay aligned with the user’s memory or may amplify retrieval errors over multiple steps.
>
> These results confirm that RF-Mem complements rather than competing with them, and that its benefits extend beyond standard dense retrieval.
>
> ### 1.3 Results
>
> **HyDE (LLM-based query expansion) on PersonaBench**
>
> |Method|R@5 Overall|Basic Info|Social Info|Pref Easy|Pref Hard|R@10 Overall|Basic Info|Social Info|Pref Easy|Pref Hard|
> |-|-|-|-|-|-|-|-|-|-|-|
> |Famili.|0.4484|0.4515|0.4852|0.4904|0.3659|0.5964|0.5879|0.6220|0.6442|0.5561|
> |Recol.|0.4491|0.4379|0.4903|0.5128|0.3854|0.6062|0.5924|0.6859|0.5659|0.6267|
> |**RF-Mem**|**0.4701**|0.4788|0.5091|0.4872|0.3854|**0.6071**|0.5924|0.6799|0.5707|0.6267|
> ||||||||||
> |HyDE + Famili.|0.3464|0.3106|0.3909|0.4615|0.3122|0.5120|0.5000|0.4991|0.5737|0.5220|
> |HyDE + Recol.|0.3482|0.3015|0.4135|0.4615|0.3171|0.5046|0.4909|0.5028|0.5929|0.4878|
> |**HyDE + RF-Mem**|**0.3504**|0.3061|0.4135|0.4615|0.3171|**0.5194**|0.5091|0.5028|0.5929|0.5220|
>
>
> **Search-o1 (LLM-based iterative retrieval) on PersonaMem**
>
> |Method|overall|Revisit Reasons|Shared Facts|Track Evolution|Aligned Recs|Latest Prefs|New Scenarios|New Ideas|
> |-|-|-|-|-|-|-|-|-|
> |RAG-Famili. (Dense)|0.5908|0.9091|0.5426|0.6475|0.6364|0.6471|0.5614|0.2151|
> |RAG-Recol.|0.6214|0.9495|0.5194|0.6547|0.7818|0.7059|0.5965|0.2688|
> |**RAG-RF-Mem**|**0.6350**|0.9495|0.5659|0.6619|0.7818|0.7059|0.6140|0.2688|
> |Search-o1 + Famili. (Dense)|0.5823|0.8687|0.5271|0.6259|0.6545|0.5882|0.5614|0.2581|
> |Search-o1 + Recol.|0.6010|0.8990|0.4961|0.6619|0.7091|0.6471|0.5789|0.2796|
> |**Search-o1 + RF-Mem**|**0.6146**|0.9293|0.5349|0.6978|0.7455|0.6471|0.5789|0.2043|

---

> ### Author Response · Authors · 2025-11-19
> **Response to Reviewer pdxs [Part 2]**
>
> ## 2. Ablation of the gating mechanism
>
> > **Your Concern (W2):** Second, the paper lacks an ablation study of the gating mechanism. It is unclear how much the mean similarity score and entropy respectively contribute to routing decisions, and how often each retrieval path is activated.
>
> Thank you very much for highlighting this important point. We agree that understanding the respective contributions of the mean similarity score and entropy is essential for interpreting the behavior of the controller. Following your suggestion, we conducted an ablation study on PersonaBench (using the implement setup described in Appendix B) to show the effect of each signal. We report results using two representative retrievers: multi-qa-MiniLM-L6-cos-v1 and BAAI/bge-base-en-v1.5.
>
> ### 2.1 Experiment setup
> For each retriever, we evaluate three variants:  (1) the full RF-Mem gate,  (2) RF-Mem without the mean similarity score, and
> (3) RF-Mem without entropy.  We measure routing frequency (number of queries sent to Familiarity vs. Recollection), latency, and retrieval accuracy (R@5).
>
> ### 2.2 Findings
> 1. **Both mean similarity and entropy are important for model performance.** Removing either signal degrades accuracy or destabilizes routing.
> 2. **On MiniLM**, the mean similarity score primarily governs **latency** (controlling when faster Familiarity is used), while entropy mainly affects **recall accuracy**.
> 3. **On BGE**, both signals significantly influence **recall accuracy**, indicating that this retriever benefits from using the two jointly.
>
> ### 2.3 Results
>
> **multi-qa-MiniLM-L6-cos-v1 (PersonaBench, R@5)**
>
> |Method|Time|Route to Famili.|Route to Recol.|Overall|Basic Info|Social Info|Pref Easy|Pref Hard|
> |--------|------|----------------|--------------|---------|------------|-------------|-----------|-----------|
> |**RF-Mem**|9.16ms|60|203|**0.4701**|0.4788|0.5091|0.4872|0.3854|
> |w/o mean score|9.67ms|51|212|**0.4701**|0.4788|0.5091|0.4872|0.3854|
> |w/o entropy|8.95ms|107|156|0.4419|0.4379|0.4852|0.4904|0.3659|
>
> **BAAI/bge-base-en-v1.5 (PersonaBench, R@5)**
>
> |Method|Time|Route to Famili.|Route to Recol.|Overall|Basic Info|Social Info|Pref Easy|Pref Hard|
> |--------|------|----------------|--------------|---------|------------|-------------|-----------|-----------|
> |**RF-Mem**|10.14ms|140|123|**0.3836**|0.4015|0.3619|0.4487|0.3220|
> |w/o mean score|9.59ms|180|83|0.3815|0.3924|0.3619|0.4615|0.3268|
> |w/o entropy|10.28ms|145|118|0.3790|0.3970|0.3431|0.4583|0.3268|
>
>
> ---
>
> ## 3. On the scope of the uncertainty signal
> >**Your Concern (W3):** Third, the definition of uncertainty is limited to similarity score distributions, which may fail to capture semantic ambiguity or misleading high-confidence matches; this limitation should be acknowledged and discussed.
>
>
> Thank you very much for raising this important point. We fully agree that our current uncertainty signal, which relies solely on the distribution of similarity scores, does not explicitly model deeper forms of semantic ambiguity or cases where misleading high-confidence matches occur. This is an inherent limitation of similarity-based gating, and we will clarify this point in the revised manuscript.
>
> In response to your comment, we have added a discussion in the **Limitation and Future Work** section.
> > In **Limitaion** we add, *“Third, our uncertainty signal is based on similarity scores and entropy, which may not fully capture semantic ambiguity.”*
> >
> > In **Future Work** we add, *“In addition, incorporating semantic or contextual uncertainty cues beyond similarity distributions may further improve the reliability of the strategy selection mechanism.”*

---

> ### Author Response · Authors · 2025-11-19
> **Response to Reviewer pdxs [Part 3]**
>
> ## 4. The learning mehtod of the strategy selection mechanism
> >**Your Concern (Q1):** Could adaptive or learned gating mechanisms outperform the hand-tuned parameters used here?
>
> Thank you very much for this insightful comment. We fully agree that the thresholds in our current gate are manually set. This design choice is primarily motivated by `privacy considerations`: under data protection regulations such as GDPR, systems are generally prohibited from adaptively learning from a user’s personal memory corpus without explicit consent. In many real-world industry deployments, this means that a non-learned gating mechanism is not only operationally simpler but also the legally compliant option.
>
> However, inspired by your suggestion, we additionally designed and trained a lightweight MLP classifier that takes probe retrieval statistics (mean similarity, entropy) as input and predicts whether to route to familiarity or recollection. This learned gate exhibits robustness, and its generalization benefits grow naturally with increasing training data. **This experiment have added in the Appendix D.6 in the revised version.**
>
> ### 4.1 Experiment setup
> Motivated by your suggestion, we conducted a new experiment to evaluate whether the gate can be learned from data.
>
> **Traning Data.** We used the public PersonaMem-32k (589 samples) and PersonaMem-128k (2727 samples) subsets. For 32k, we used the first 400 samples for training and the remaining 189 for testing; for 128k, the first 2000 samples for training and the remaining 727 for testing. We formed a binary classification dataset using only cases where one path clearly wins (ties removed).
>
> **Training Details.** Each sample is represented by the two gate signals, mean similarity s and entropy H(p), and the label 0/1 indicates whether Familiarity or Recollection is better, respectively. We used an 80%–20% train–validation split and accuracy as the metric. Each experiment was repeated 10 times with different random seeds.
>
> ### 4.2 Findings
> 1. **The effectiveness of the auto-tuning mechanism grows with sample sizes.**
>    With limited supervision (32k), the learned gate is statistically unstable and may even underperform a simple “always recollect’’ baseline. As the training data increases (128k), the gate becomes noticeably more stable and its performance steadily improves, approaching—but not surpassing—the hand-tuned parameters.
>
> 2. **The mean similarity $\bar{s}$ and entropy $H(p)$ are proper signals.**
>    The auto-tuning results validate our choice of using mean similarity $\bar{s}$ and entropy $H(p)$ as the two gating signals: even a simple classifier using only these two features achieves competitive performance across data scales. This indicates that our hand-tuned parameters are not ad hoc but capture a meaningful decision boundary in the data.
>
> ### 4.3 Results
> **PersonaMem-32k**: First 400 samples for training, last 189 for testing
>
> |Training Size|Famili. win|Recol. win|Tie|
> |-|-|-|-|
> |75=(34+41)|34|41|325|
>
> |Method|Overall|Revisit Reasons|Shared Facts|Track Evolution|Aligned Recs|Latest Prefs|New Scenarios|New Ideas|Num of Famili.|Num of Recol.|
> |-|-|-|-|-|-|-|-|-|-|-|
> |Famili.|0.6296|0.9688|0.7297|0.6327|0.8125|0.2857|0.5385|0.2286|189|0|
> |Recol.|$\underline{\text{0.6667}}$|0.9688|0.7297|0.7551|0.8750|0.1429|0.7297|0.2286|0|189|
> |RF-Mem (hand gate)|**0.6720**|1.0000|0.7838|0.6939|0.8750|0.1429|0.6154|0.2571|83|106|
> |RF-Mem (learned gate)|0.6482±0.0067|0.9688±0.0000|0.7514±0.0114|0.6878±0.0255|0.8750±0.0000|0.1429±0.0000|0.5385±0.0000|0.2286±0.0000|97.5±6.7|91.5±6.7|
>
>
> **PersonaMem-128k**: First 2000 samples for training, last 727 for testing
>
> |Training Size|Famili. win|Recol. win|Tie|
> |-|-|-|-|
> |363=(178+185)|178|185|1637|
>
> |Method|overall|Revisit Reasons|Shared Facts|Track Evolution|Aligned Recs|Latest Prefs|New Scenarios|New Ideas|Num of Famili.|Num of Recol.|
> |-|-|-|-|-|-|-|-|-|-|-|
> |Famili.|0.5131|0.7742|0.6410|0.6162|0.5176|0.5122|0.3529|0.3043|727|0|
> |Recol.|0.5158|0.7634|0.5897|0.6162|0.5647|0.5366|0.3824|0.2609|0|727|
> |RF-Mem (hand gate)|**0.5199**|0.7527|0.6410|0.6162|0.5059|0.5463|0.3824|0.2971|402|325|
> |RF-Mem (learned gate)|$\underline{\text{0.5175±0.0039}}$|0.7591±0.0055|0.6180±0.0255|0.6091±0.0048|0.5553±0.0182|0.5400±0.0108|0.3794±0.0152|0.2718±0.0070|334.4±28.3|392.6±28.3|

---

> ### Author Response · Authors · 2025-11-19
> **Response to Reviewer pdxs [Part 4]**
>
> ## 5. Alternative recollection mechanisms beyond clustering
> >**Your Concern (Q2):** The recollection process relies on clustering in embedding space. Have the authors tested alternative approaches, such as graph-based retrieval or semantic expansion via LLM-generated paraphrases, to validate the robustness of the proposed method?
>
> Thanks for raising this important question. We agree that testing the robustness of the recollection path under alternative mechanisms is essential. Conceptually, we avoid LLM-generated expansions because RF-Mem aims to perform reconstruction **in embedding space**, preserving efficiency and scalability; LLM paraphrasing introduces substantial latency and cost. To directly address your concern, **we conducted experiments in Appenix D.8 to evaluate whether the recollection path remains effective.**
>
> ### 5.1 Experiment setup
> We replaced the KMeans-based recollection with three alternative mechanisms:
>
> 1. **Graph recollection**
>    Following [3], we build a KNN graph over the memory corpus and apply a Breadth-First Search(BFS)-style expansion to gather supporting evidence.
> 2. **Density clustering**
>    Using DBSCAN [4] as a non-centroid density-based alternative.
> 3. **Spectral clustering**
>    Using the classic algorithm of [5] to evaluate performance under manifold-structured clusters.
>
> ### 5.2 Findings
>
> 1. **Clustering-based recollection generally outperforms graph-based recollection.**
>    We conjecture this is because clustering better aggregates semantically coherent neighborhoods, whereas BFS tends to over-expand in dense memory regions.
>
> 2. **The proposed strategy slection mechanism is effective.**
>    RF-Mem consistently improves over its corresponding recollection variant across all alternatives.
>
> 3. **These results offer practical insights for deployment**.
>   Industrial systems may flexibly choose the clustering method that best matches their memory structure without changing the overall RF-Mem design.
>
> ### 5.3 Results
>
> |Method|R@5 Overall|Basic Info|Social Info|Pref Easy|Pref Hard|R@10 Overall|Basic Info|Social Info|Pref Easy|Pref Hard|
> |-|-|-|-|-|-|-|-|-|-|-|
> |Famili.|0.4484|0.4515|0.4852|0.4904|0.3659|0.5964|0.5879|0.6220|0.6442|0.5561|
> |Recol.(KMeans)|0.4491|0.4379|0.4903|0.5128|0.3854|0.6062|0.5924|0.6859|0.5659|0.6267|
> |**RF-Mem (our)**|**0.4701**|0.4788|0.5091|0.4872|0.3854|**0.6071**|0.5924|0.6799|0.5707|0.6267|
> |||
> |Recol.(Graph+BFS)|0.4314|0.4167|0.4739|0.4872|0.3805|0.5722|0.5591|0.5887|0.5887|0.5366|
> |**RF-Mem (Graph+BFS)**|**0.4349**|0.4258|0.4739|0.4872|0.3756|**0.5899**|0.5742|0.6220|0.6442|0.5561|
> |||
> |Recol.(DBSCAN)|0.4431|0.4424|0.4940|0.4872|0.3512|0.6028|0.6045|0.6079|0.6891|0.5366|
> |**RF-Mem(DBSCAN)**|**0.4669**|0.4879|0.5129|0.4744|0.3463|**0.6040**|0.5659|0.6079|0.6667|0.6015|
> |||
> |Recol.(Spectral)|0.4522|0.4591|0.4739|0.5000|0.3756|0.5957|0.6045|0.5978|0.6474|0.5366|
> |**RF-Mem(Spectral)**|**0.4651**|0.4818|0.4928|0.4872|0.3707|**0.6001**|0.6015|0.6041|0.6474|0.5610|
>
> ### 5.4 Additional Analysis: Full-exploration re-ranking
>
> To further address your concern, we additionally evaluate a variant where the **recollection path removes the early-stop rule(i.e., it continues traversal rather than stopping once the top-K quota is met).**.
> In this setting, the method **full explor** (up to a predefined maximum depth), collects the full reachable candidate set, and then **re-ranks all collected memory using similarity to the query**.
>
> ### 5.5 Findings
>
> 1. **Allowing full exploration with global re-ranking further improves RF-Mem**, compared to its early-stopped variant, showing that our recollection design remains effective when given a larger search budget.
> 2. **Under the full-exploration–and–rerank setting, the graph-based variant performs better then Famili.(dense) path**, which further supports our claim that deliberate, structure-aware retrieval over the memory space is a valuable and robust wihtin alternative formulation.
>
> ### 5.6 Results
> |Method|R@5 Overall|Basic Info|Social Info|Pref Easy|Pref Hard|R@10 Overall|Basic Info|Social Info|Pref Easy|Pref Hard|
> |-|-|-|-|-|-|-|-|-|-|-|
> |Famili.|0.4484|0.4515|0.4852|0.4904|0.3659|0.5964|0.5879|0.6220|0.6442|0.5561|
> |Recol.(ours)|0.4491|0.4379|0.4903|0.5128|0.3854|0.6062|0.5924|0.6859|0.5659|0.6267|
> |**RF-Mem(ours)**|**0.4701**|0.4788|0.5091|0.4872|0.3854|**0.6071**|0.5924|0.6799|0.5707|0.6267|
> |||
> |Recol.(Graph+BFS)|0.4314|0.4167|0.4739|0.4872|0.3805|0.5722|0.5591|0.5887|0.5887|0.5366|
> |**RF-Mem(Graph+BFS)**|**0.4349**|0.4258|0.4739|0.4872|0.3756|**0.5899**|0.5742|0.6220|0.6442|0.5561|
> |||
> |Recol.(ours,full explo)|0.4537|0.4530|0.4827|0.5128|0.3805|0.6036|0.6015|0.6204|0.6763|0.5415|
> |**RF-Mem(ours,full explo)**|**0.4709**|0.4848|0.5016|0.5000|0.3756|**0.6083**|0.6015|0.6267|0.6667|0.5659|
> |||
> |Recol.(Graph+BFS, full)|0.4486|0.4530|0.4701|0.4776|0.3902|0.5841|0.5818|0.5642|0.6635|0.5659|
> |**RF-Mem(Graph+BFS, full explo)**|**0.4629**|0.4758|0.4890|0.4776|0.3854|**0.5986**|0.5924|0.6220|0.6442|0.5561|

---

> ### Author Response · Authors · 2025-11-19
> **Response to Reviewer pdxs [Part 5]**
>
> ## 6. On embedding quality and routing robustness
> >**Your Concern (Q4):** The entire framework, including the uncertainty estimation, relies heavily on the quality of the underlying embedding model. Did you observe significant performance variance across the three retrievers (MiniLM, MPNet, BGE) in terms of the routing accuracy? Is there a risk that a poorly calibrated embedding model could lead to a cascading failure in the gating mechanism?
>
> Thank you very much for raising this important question. We address your concern in two parts. First, we examine whether routing behavior varies substantially across different embedding models. Second, we discuss the failures case when the route select a wrong mode. **This experiment has been added in Appendix D.10 in the revised version.**
>
> ### 6.1 Experiment Setup
> To answer *“Did you observe significant performance variance across the three retrievers (MiniLM, MPNet, BGE) in terms of the routing accuracy?”*,  we measure routing statistics for all three embedding models on PersonaBench, following the same setup as in the main experiments.
>
> ### 6.2 Findings
>
> 1. **Stronger retrievers route more often to the recollection path while eaker retrievers route more often to fast familiarity.**
>    MiniLM and MPNet exhibit a much higher number of transitions into the recollection branch, suggesting that stronger embedding models provide more reliable cluster structure for deliberate reconstruction.
>    BGE (base English v1.5), whose similarity distributions are known to be less sharply calibrated, routes more queries to familiarity, reflecting its weaker ability to support iterative cluster-based refinement.
>
> 2. **These differences align with our expectations.**
>    The gating mechanism adapts naturally to the geometry of the embedding space:  when recollection is likely to help, it chooses it; when the embedding structure is noisy, it choose to the safer one-shot familiarity mode.
>
> ### 6.3 Results
>
> |Retriever|Time|Route→Famili.|Route→Recol.|Overall R@5|Basic Info|Social Info|Pref Easy|Pref Hard|
> |-|-|-|-|-|-|-|-|-|
> |multi-qa-MiniLM-L6-cos-v1|9.16ms|60|203|0.4701|0.4788|0.5091|0.4872|0.3854|
> |all-mpnet-base-v2|8.33ms|15|248|0.4009|0.4515|0.4487|0.4000|0.2730|
> |BAAI/bge-base-en-v1.5|10.14ms|140|123|0.3836|0.4015|0.3619|0.4487|0.3220|
>
>
>
> ### 6.4 Filure Case Analysis
> In response to your comment of *Is there a risk that a poorly calibrated embedding model could lead to a cascading failure in the gating mechanism?*, **we have added a filure case in the Appendix D.7.2.**
>
> > To analysis the failure case, we again compare one-shot Familiarity retrieval with the multi-round Recollection process on the PersonaMem dataset. As shown in Figure 19, we show the retrieval result of dual paths for the query *''I'm considering developing a tool to manage finances while traveling. How could I ensure it helps me prioritize experiences like local cuisine or guided tours?’’*. Familiarity retrieval outperforms Recollection in serving the user’s intent. The one-shot Familiarity retriever returns a broad and query-aligned set of memories, many of which explicitly mention 'local experiences', 'finance management apps', or `last trip'. These results directly address the user’s need to “prioritize experiences while traveling,” illustrating that high-scoring surface cues can sometimes be sufficient for intent coverage.
> >
> > In contrast, the multi-round Recollection process drifts along a different semantic trajectory. Although its iterative clustering-and-refinement mechanism successfully produces deeper and more structured traces, the retrieved branches gradually converge toward 'long-term financial habits' and 'personal finance management'. This indicates that the recollection trajectory can overfit to dominant semantic clusters in the memory corpus, especially when several high-density branches are thematically coherent but misaligned with the user’s immediate intent. As a result, the recollection path becomes increasingly anchored in financial-management narratives, overlooking the travel-related aspects of the query.
> >
> > Despite being a bad case for retrieval accuracy, this example is instructive. It exposes a core trade-off of deliberative recollection: the mechanism excels at reconstructing temporally dispersed, conceptually rich memory chains, yet it may over-emphasize internal coherence at the cost of task-oriented alignment. This case highlights the importance of balancing depth with intent fidelity, and underscores the need for more precise familiarity-uncertainty–guided strategy selection in future work.
>
> ---
>
> We hope these responses address your concerns comprehensively. Feel free to let us know if you have any additional questions, and we are more than happy to provide further clarification on any aspect of our work.
>
> *Respectfully and sincerely,*
>
> **The Authors**

---

> ### Author Response · Authors · 2025-11-19
> **Response to Reviewer pdxs [Part 6]**
>
> [1] Luyu Gao, Xueguang Ma, Jimmy Lin, and Jamie Callan. 2023. Precise Zero-Shot Dense Retrieval without Relevance Labels. In Proceedings of the 61st Annual Meeting of the Association for Computational Linguistics (Volume 1: Long Papers), pages 1762–1777, Toronto, Canada. Association for Computational Linguistics.
>
> [2] Xiaoxi Li, Guanting Dong, Jiajie Jin, Yuyao Zhang, Yujia Zhou, Yutao Zhu, Peitian Zhang, and Zhicheng Dou. 2025. Search-o1: Agentic Search-Enhanced Large Reasoning Models. In Proceedings of the 2025 Conference on Empirical Methods in Natural Language Processing, pages 5420–5438, Suzhou, China. Association for Computational Linguistics.
>
> [3] Zhang, Qinggang, et al. "A survey of graph retrieval-augmented generation for customized large language models." arXiv preprint arXiv:2501.13958 (2025).
>
> [4] Ester M, Kriegel H P, Sander J, et al. A density-based algorithm for discovering clusters in large spatial databases with noise[C]//kdd. 1996, 96(34): 226-231.
>
> [5] Ng A, Jordan M, Weiss Y. On spectral clustering: Analysis and an algorithm[J]. Advances in neural information processing systems, 2001, 14.

---

### Official Review · Reviewer_eDaP · 2025-11-13

**Soundness:** 3
**Presentation:** 4
**Contribution:** 3
**Rating:** 6
**Confidence:** 3

**Summary:**

This paper introduces RF-Mem, a dual-process memory retrieval framework inspired by cognitive science, combining a fast Familiarity path and a deliberate Recollection path for personalized LLM memory retrieval. By using a familiarity-uncertainty gating mechanism and a retrieve–cluster–mix procedure, RF-Mem adaptively retrieves evidence under varied user-memory relevance conditions.

**Strengths:**

1. The paper effectively motivates its design using the Recollection–Familiarity Dual-Process Theory (Lines 61–83) and successfully maps the cognitive analogy into a computational retriever.

2. The familiarity-uncertainty gating (Lines 162–201) provides principled decision logic for choosing between retrieval modes, reducing unnecessary expansion while preserving robustness.

3. The adaptive study (Lines 432–454) shows that RF-Mem complements rather than replaces indexing approaches like MemoryBank, strengthening its practicality.

**Weaknesses:**

1. The α-mix formula (Lines 246–257) is introduced but lacks theoretical clarity on why linear mixing is optimal for query expansion.

2. $\tau$ (Lines 174–201) is fixed globally but may vary significantly across users, domains, or embedding models.

3. Although the paper compares fairly against retrieval-only baselines (Lines 289–303), retrieval today heavily relies on query rewriting, which is absent from the comparisons.

4. The ethical considerations are brief and do not address challenges like preference over-amplification, exposure to sensitive data, or user-profiling bias.

**Questions:**

1. Could the authors consider ablation comparing linear vs. nonlinear mixing (e.g., gating networks, learned interpolation), or reference theoretical grounding such as manifold-preserving mixing used in retrieval augmentation?

2. Could the authors consider learned gating or meta-calibrated thresholds conditioned on query distribution or similarity statistics?

3. Include comparisons to LLM-based query reformulation modules such as LD-Agent or MemoCue, since they also address deeper reasoning retrieval.

---

> ### Author Response · Authors · 2025-11-19
> **Response to Reviewer eDaP [Part 1]**
>
> **Dear Reviewer eDaP**,
>
> Thank you very much for your thoughtful, detailed, and constructive review. We deeply appreciate your generous recognition of the paper’s conceptual grounding, the clarity of its presentation, and the strength of its dual-process motivation. Your comments precisely identify both the theoretical and practical aspects of RF-Mem, and we are grateful for how carefully you engaged with the cognitive analogy, the gating formulation, and the adaptive studies.
> Your suggestions are highly valuable to us, and we have revised the paper to address each concern with new analyses, additional experiments, and clearer discussion. We sincerely thank you for your insightful assessment and constructive suggestions, all of which helped us significantly strengthen the work.
>
>
> ## 1. Theoretical grounding of α-mixing and Gate empirical study
> >**Your Concern (W1):** The α-mix formula (Lines 246–257) is introduced but lacks theoretical clarity on why linear mixing is optimal for query expansion.
> >**Your Concern (Q1):** Could the authors consider ablation comparing linear vs. nonlinear mixing (e.g., gating networks, learned interpolation), or reference theoretical grounding such as manifold-preserving mixing used in retrieval augmentation?
>
> Thank you for your insightful comments. We have conducted both **theoretical analysis** and **additional alternative study** comparing linear α-mixing with nonlinear gated interpolation, following the reviewer’s suggestion. Our findings consistently support the use of our α-mix formulation.
>
>
> ### 1.1 Theoretical grounding of α-mix
>
> Our query update
> $$
> \mathbf{x}_b^{(r+1)} =\mathrm{norm}\big(\alpha\mathbf{x}^{(r)} + (1-\alpha)\mathbf{g}_b^{(r)} + \mathbf{x}_t \big)
> $$
> is equal to
>
> $$
> \mathbf{x}_b^{(r+1)} =\mathrm{norm}\big(\alpha(\mathbf{x}^{(r)}+\mathbf{x}_t) + (1-\alpha)(\mathbf{g}_b^{(r)} + \mathbf{x}_t) \big)
> $$
> which can simply define as
> $$
>    x _ {\text{new}} = \mathrm{norm}(\alpha x + (1-\alpha) g)
> $$
> is not merely a linear interpolation. Since all embeddings are unit-normalized, this operation is a **Riemannian retraction**[1] on the **unit sphere manifold**.
>
> For two points \(x, g \in \mathbb{S}^{d-1}\), a Euclidean update
> $$
> x + v,\quad v=(1-\alpha)(g-x)
> $$
> moves off the manifold[2]. Retraction maps it back, and substituting \(v\) gives:
>
> $$
>   R_x(v) = \frac{x + v}{\|x + v\|} = \frac{\alpha x + (1-\alpha) g}{\|\alpha x + (1-\alpha) g\|} = \mathrm{norm}(\alpha x + (1-\alpha) g)
> $$
> which is exactly our α-mix.
>
> Thus, α-mix is a **first-order approximation of spherical interpolation** and ensures:
>
> 1. manifold-preserving mixing aligned with cosine similarity geometry
> 2. stable and bounded direction updates
> 3. theoretical reliability under sub-Gaussian similarity (Appendix F.3)
>
> ---
>
> ### 1.2 Nonlinear gated mixing alternative study
>
> Following the reviewer’s suggestion, we implemented **nonlinear gated mixing**:
>
> $$
> g(\mathbf{x}^{(r)},\mathbf{g}_b^{(r)})=\sigma(\mathbf{x}^{(r)\top}\mathbf{g}_b^{(r)}),
> \quad
> \mathbf{x}_b^{(r+1)} =\mathrm{norm}\big(g\mathbf{x}^{(r)} + (1-g)\mathbf{g}_b^{(r)} + \mathbf{x}_t \big)
> $$
>
> where $\mathbf{x}^{(r)}$ is the current query, $\mathbf{g}_b^{(r)}$ the centroid, and $\sigma$ is the sigmoid function.
>
> Experiments were performed on **PersonaBench** using **multi-qa-MiniLM-L6-cos-v1**. **This experiment have added in the Appendix D.8 in the revised version.**
>
> ### 1.3 Findings
> These results support α-mix as the more robust and theoretically grounded mixing rule for non-parametric retrieval.
> 1. **Gated mixing is a reasonable alternative** and improves over simple Familiarity retrieval.
> 2. **However, it does not outperform α-mix**, especially in Social and Pref-Hard categories, where recollection is most critical. The α-mix yields **more stable, geometry-respecting updates**, consistent with its manifold interpretation.
>
> ### 1.4 Experimental results
>
> | Method | R@5 Overall | Basic | Social | Pref Easy | Pref Hard | R@10 Overall | Basic | Social | Pref Easy | Pref Hard |
> |-------|-------------|-------|--------|------------|-----------|--------------|--------|--------|-------------|------------|
> | Famili. | 0.4484 | 0.4515 | 0.4852 | 0.4904 | 0.3659 | 0.5964 | 0.5879 | 0.6220 | 0.6442 | 0.5561 |
> | Recol. (ours) | 0.4491 | 0.4379 | 0.4903 | 0.5128 | 0.3854 | 0.6062 | 0.5924 | 0.6859 | 0.5659 | 0.6267 |
> | **RF-Mem (ours)** | **0.4701** | 0.4788 | 0.5091 | 0.4872 | 0.3854 | **0.6071** | 0.5924 | 0.6799 | 0.5707 | 0.6267 |
> |||
> | Recol. + gate | 0.4491 | 0.4439 | 0.4739 | 0.5128 | 0.3902 | 0.5936 | 0.5924 | 0.6016 | 0.6795 | 0.5317 |
> | **RF-Mem + gate** | **0.4602** | 0.4667 | 0.4928 | 0.5000 | 0.3756 | **0.5988** | 0.5924 | 0.6079 | 0.6667 | 0.5610 |

---

> ### Author Response · Authors · 2025-11-19
> **Response to Reviewer eDaP [Part 2]**
>
> ## 2. The learning mehtod of the strategy selection mechanism
> >**Your Concern (W2):** $\tau$ (Lines 174–201) is fixed globally but may vary significantly across users, domains, or embedding models.
> >
> >**Your Concern (Q2):** Could the authors consider learned gating or meta-calibrated thresholds conditioned on query distribution or similarity statistics?
>
> Thank you very much for this insightful comment. We fully agree that the thresholds in our current gate are manually set. This design choice is primarily motivated by `privacy considerations`: under data protection regulations such as GDPR, systems are generally prohibited from adaptively learning from a user’s personal memory corpus without explicit consent. In many real-world industry deployments, this means that a non-learned gating mechanism is not only operationally simpler but also the legally compliant option.
>
> However, inspired by your suggestion, we additionally designed and trained a lightweight MLP classifier that takes probe retrieval statistics (mean similarity, entropy) as input and predicts whether to route to familiarity or recollection. This learned gate exhibits robustness, and its generalization benefits grow naturally with increasing training data. **This experiment have added in the Appendix D.6 in the revised version.**
>
> ### 2.1 Experiment setup
> Motivated by your suggestion, we conducted a new experiment to evaluate whether the gate can be learned from data.
>
> **Traning Data.** We used the public PersonaMem-32k (589 samples) and PersonaMem-128k (2727 samples) subsets. For 32k, we used the first 400 samples for training and the remaining 189 for testing; for 128k, the first 2000 samples for training and the remaining 727 for testing. We formed a binary classification dataset using only cases where one path clearly wins (ties removed).
>
> **Training Details.** Each sample is represented by the two gate signals, mean similarity $\bar{s}$ and entropy $H(p)$, and the label 0/1 indicates whether Familiarity or Recollection is better, respectively. We used an 80%–20% train–validation split and accuracy as the metric. Each experiment was repeated 10 times with different random seeds.
>
> ### 2.2 Findings
> 1. **The effectiveness of the auto-tuning mechanism grows with sample sizes.**
>    With limited supervision (32k), the learned gate is statistically unstable and may even underperform a simple “always recollect’’ baseline. As the training data increases (128k), the gate becomes noticeably more stable and its performance steadily improves, approaching—but not surpassing—the hand-tuned parameters.
>
> 2. **The mean similarity $\bar{s}$ and entropy $H(p)$ are proper signals.**
>    The auto-tuning results validate our choice of using mean similarity $\bar{s}$ and entropy $H(p)$ as the two gating signals: even a simple classifier using only these two features achieves competitive performance across data scales. This indicates that our hand-tuned parameters are not ad hoc but capture a meaningful decision boundary in the data.
>
> ### 2.3 Results
> **PersonaMem-32k**: First 400 samples for training, last 189 for testing
>
> |Training Size|Famili. win|Recol. win|Tie|
> |-|-|-|-|
> |75=(34+41)|34|41|325|
>
> |Method|Overall|Revisit Reasons|Shared Facts|Track Evolution|Aligned Recs|Latest Prefs|New Scenarios|New Ideas|Num of Famili.|Num of Recol.|
> |-|-|-|-|-|-|-|-|-|-|-|
> |Famili.|0.6296|0.9688|0.7297|0.6327|0.8125|0.2857|0.5385|0.2286|189|0|
> |Recol.|$\underline{\text{0.6667}}$|0.9688|0.7297|0.7551|0.8750|0.1429|0.7297|0.2286|0|189|
> |RF-Mem (hand gate)|**0.6720**|1.0000|0.7838|0.6939|0.8750|0.1429|0.6154|0.2571|83|106|
> |RF-Mem (learned gate)|0.6482±0.0067|0.9688±0.0000|0.7514±0.0114|0.6878±0.0255|0.8750±0.0000|0.1429±0.0000|0.5385±0.0000|0.2286±0.0000|97.5±6.7|91.5±6.7|
>
>
> **PersonaMem-128k**: First 2000 samples for training, last 727 for testing
>
> |Training Size|Famili. win|Recol. win|Tie|
> |-|-|-|-|
> |363=(178+185)|178|185|1637|
>
> |Method|overall|Revisit Reasons|Shared Facts|Track Evolution|Aligned Recs|Latest Prefs|New Scenarios|New Ideas|Num of Famili.|Num of Recol.|
> |-|-|-|-|-|-|-|-|-|-|-|
> |Famili.|0.5131|0.7742|0.6410|0.6162|0.5176|0.5122|0.3529|0.3043|727|0|
> |Recol.|0.5158|0.7634|0.5897|0.6162|0.5647|0.5366|0.3824|0.2609|0|727|
> |RF-Mem (hand gate)|**0.5199**|0.7527|0.6410|0.6162|0.5059|0.5463|0.3824|0.2971|402|325|
> |RF-Mem (learned gate)|$\underline{\text{0.5175±0.0039}}$|0.7591±0.0055|0.6180±0.0255|0.6091±0.0048|0.5553±0.0182|0.5400±0.0108|0.3794±0.0152|0.2718±0.0070|334.4±28.3|392.6±28.3|

---

> ### Author Response · Authors · 2025-11-19
> **Response to Reviewer eDaP [Part 3]**
>
> ## 3. RF-Mem under stronger LLM-based query expansion and iterative retrieval baselines
> >**Your Concern (W3):** Although the paper compares fairly against retrieval-only baselines (Lines 289–303), retrieval today heavily relies on query rewriting, which is absent from the comparisons.
>
> Thank you very much for raising this important point. We agree that positioning RF-Mem against more advanced LLM-based retrieval frameworks is necessary for a comprehensive evaluation. Conceptually, RF-Mem operates **solely at the retrieval stage** and is orthogonal to query expansion,or iterative retrieval. In principle, RF-Mem can be inserted into these pipelines as a *retriever-side controller*, and we have demonstrated one such adaptation on MemoryBank (Seciton 3.4) in the main paper. For further validation, we have now conducted additional experiments integrating RF-Mem with two representative LLM-based retrieval frameworks: (1) HyDE [3] and (2) Search-o1 [4]. **This experiment have added in the Section 3.4.2 and 3.4.3 in the revised version.**
>
> ### 3.1 Experiment setup
>
> 1. **LLM-based query expansion (HyDE, ACL 2023, CMU)[3]**
>    We adopt HyDE as a representative query expansion baseline.
>    Following the standard protocol for retrieval-oriented methods, we evaluate Recall on PersonaBench using multi-qa-MiniLM-L6-cos-v1.
>
> 2. **LLM-based iterative retrieval (Search-o1, EMNLP 2025, RUC)[4]**
>    We adopt Search-o1 as a representative iterative multi-step retrieval baseline, which implements a comparable multi-step retrieval process. Following standard practice for iterative methods, we evaluate generation accuracy on PersonaMem.
>
>
> ### 3.2 Findings
> 1. **RF-Mem is compatible with LLM-based query expansion.**
>    Adding RF-Mem on top of HyDE consistently improves R@5 and R@10. Notably, HyDE alone underperforms the base query, which we conjecture is due to the personalized nature of this task, where generic query expansions may dilute user-specific signals.
> 2. **RF-Mem is also compatible with iterative LLM-based retrieval.**
>    Integrated into Search-o1, RF-Mem further improves personalized generation accuracy across multiple categories. Interestingly, Search-o1 alone underperforms the base RAG setting, which we conjecture is due to the personalized nature of the task, where the iteratively generated queries may fail to stay aligned with the user’s memory or may amplify retrieval errors over multiple steps.
>
> These results confirm that RF-Mem complements query expansion and LLM-based iterative retrieval rather than competing with them, and that its benefits extend beyond standard dense retrieval.
>
> ### 3.3 Results
>
> **HyDE (LLM-based query expansion) on PersonaBench**
>
> | Method | R@5 Overall | Basic Info | Social Info | Pref Easy | Pref Hard | R@10 Overall | Basic Info | Social Info | Pref Easy | Pref Hard |
> |--------|-------------|-------------|-------------|-----------|-----------|--------------|-------------|-------------|-----------|-----------|
> | Famili. | 0.4484 | 0.4515 | 0.4852 | 0.4904 | 0.3659 | 0.5964 | 0.5879 | 0.6220 | 0.6442 | 0.5561 |
> | Recol. | 0.4491 | 0.4379 | 0.4903 | 0.5128 | 0.3854 | 0.6062 | 0.5924 | 0.6859 | 0.5659 | 0.6267 |
> | **RF-Mem** | **0.4701** | 0.4788 | 0.5091 | 0.4872 | 0.3854 | **0.6071** | 0.5924 | 0.6799 | 0.5707 | 0.6267 |
> ||||||||||
> | HyDE + Famili. | 0.3464 | 0.3106 | 0.3909 | 0.4615 | 0.3122 | 0.5120 | 0.5000 | 0.4991 | 0.5737 | 0.5220 |
> | HyDE + Recol. | 0.3482 | 0.3015 | 0.4135 | 0.4615 | 0.3171 | 0.5046 | 0.4909 | 0.5028 | 0.5929 | 0.4878 |
> | **HyDE + RF-Mem** | **0.3504** | 0.3061 | 0.4135 | 0.4615 | 0.3171 | **0.5194** | 0.5091 | 0.5028 | 0.5929 | 0.5220 |
>
>
> **Search-o1 (LLM-based iterative retrieval) on PersonaMem**
>
> | Method | overall | Revisit Reasons | Shared Facts | Track Evolution | Aligned Recs | Latest Prefs | New Scenarios | New Ideas |
> |--------|---------|-----------------|--------------|-----------------|--------------|--------------|----------------|-----------|
> | RAG-Famili. (Dense) | 0.5908 | 0.9091 | 0.5426 | 0.6475 | 0.6364 | 0.6471 | 0.5614 | 0.2151 |
> | RAG-Recol. | 0.6214 | 0.9495 | 0.5194 | 0.6547 | 0.7818 | 0.7059 | 0.5965 | 0.2688 |
> | **RAG-RF-Mem** | **0.6350** | 0.9495 | 0.5659 | 0.6619 | 0.7818 | 0.7059 | 0.6140 | 0.2688 |
> | Search-o1 + Famili. (Dense) | 0.5823 | 0.8687 | 0.5271 | 0.6259 | 0.6545 | 0.5882 | 0.5614 | 0.2581 |
> | Search-o1 + Recol. | 0.6010 | 0.8990 | 0.4961 | 0.6619 | 0.7091 | 0.6471 | 0.5789 | 0.2796 |
> | **Search-o1 + RF-Mem** | **0.6146** | 0.9293 | 0.5349 | 0.6978 | 0.7455 | 0.6471 | 0.5789 | 0.2043 |

---

> ### Author Response · Authors · 2025-11-19
> **Response to Reviewer eDaP [Part 4]**
>
> ## 4. RF-Mem in LD-Agent
> >**Your Concern (Q3):** Include comparisons to LLM-based query reformulation modules such as LD-Agent or MemoCue, since they also address deeper reasoning retrieval.
>
> We sincerely thank you for this valuable suggestion. LD-Agent[5] was not included in our original scope because our experiments primarily focused on retrieval-stage methods, but following your recommendation, we have now conducted additional evaluations integrating RF-Mem into LD-Agent to assess its plug-in compatibility.  Apologies for the time constraint; we will include MemoCue in the camera-ready version.
>
> ### 4.1 Experiment Setup
> Following LD-Agent [5], which primary focus on generation task, we reproduced its agent pipeline on the PeronsaMem benchmark and integrated RF-Mem in the longmemory retrieval stage.
>
> ### 4.2 Findings
> RF-Mem consistently improves LD-Agent’s performance across multiple reasoning categories. This further confirms that RF-Mem is *plug-in in retrieve stage mtehod* and can be seamlessly integrated into LLM-based query reformulation systems.
>
> ### 4.3 Results
>
> | Method | overall | Revisit Reasons | Shared Facts | Track Evolution | Aligned Recs | Latest Prefs | New Scenarios | New Ideas |
> |--------|---------|------------------|---------------|------------------|---------------|----------------|----------------|-------------|
> | Famili. | 0.5908 | 0.9091 | 0.5426 | 0.6475 | 0.6364 | 0.6471 | 0.5614 | 0.2151 |
> | Recol. | 0.6214 | 0.9495 | 0.5194 | 0.6547 | 0.7818 | 0.7059 | 0.5965 | 0.2688 |
> | **RF-MEM** | **0.6350** | 0.9495 | 0.5659 | 0.6619 | 0.7818 | 0.7059 | 0.6140 | 0.2688 |
> |||
> | LD-Agent (Famili.) | 0.4873 | 0.8081 | 0.4031 | 0.6043 | 0.4364 | 0.5294 | 0.4386 | 0.1398 |
> | LD-Agent (recol.) | 0.5008 | 0.8283 | 0.4118 | 0.5899 | 0.4545 | 0.4496 | 0.4737 | 0.1505 |
> | **LD-Agent + RF-MEM** | **0.5093** | 0.8182 | 0.4341 | 0.6259 | 0.4727 | 0.4706 | 0.4912 | 0.1505 |
>
>
> ## 5. Ethical consideration
> >**Your Concern (W4):** The ethical considerations are brief and do not address challenges like preference over-amplification, exposure to sensitive data, or user-profiling bias.
>
> We sincerely appreciate you for highlighting this important ethical concern. In response to your comment, **we have added the description in the Ethics Statement section.**
>
> >*We acknowledge that real-world deployments of preference-based systems may inadvertently over-amplify user traits or reinforce behavioral biases. Additionally, improper handling of user histories could expose sensitive information or lead to unintended profiling effects, underscoring the need for careful governance.*
>
> ---
>
>
> We hope these responses address your concerns comprehensively. Feel free to let us know if you have any additional questions, and we are more than happy to provide further clarification on any aspect of our work.
>
> *Respectfully and sincerely,*
>
> **The Authors**
>
> [1] Hsieh, Ya-Ping, et al. "Riemannian stochastic optimization methods avoid strict saddle points." Advances in Neural Information Processing Systems 36 (2023): 29580-29601.
>
> [2] Kasai, Hiroyuki, Pratik Jawanpuria, and Bamdev Mishra. "Riemannian adaptive stochastic gradient algorithms on matrix manifolds." International conference on machine learning. PMLR, 2019.
>
> [3] Luyu Gao, Xueguang Ma, Jimmy Lin, and Jamie Callan. 2023. Precise Zero-Shot Dense Retrieval without Relevance Labels. In Proceedings of the 61st Annual Meeting of the Association for Computational Linguistics (Volume 1: Long Papers), pages 1762–1777, Toronto, Canada. Association for Computational Linguistics.
>
> [4] Xiaoxi Li, Guanting Dong, Jiajie Jin, Yuyao Zhang, Yujia Zhou, Yutao Zhu, Peitian Zhang, and Zhicheng Dou. 2025. Search-o1: Agentic Search-Enhanced Large Reasoning Models. In Proceedings of the 2025 Conference on Empirical Methods in Natural Language Processing, pages 5420–5438, Suzhou, China. Association for Computational Linguistics.
>
> [5] Li, Hao, et al. "Hello again! llm-powered personalized agent for long-term dialogue." Proceedings of the 2025 Conference of the Nations of the Americas Chapter of the Association for Computational Linguistics: Human Language Technologies (Volume 1: Long Papers). 2025.

---

### Author Response · Authors · 2025-11-21
**Rebuttal Summary**

**Dear Reviewers, AC, SAC, and PC,**

We sincerely thank you for the time, expertise, and thoughtful feedback you have provided in reviewing our manuscript. We greatly appreciate the reviewers’ recognition of the conceptual novelty of **RF-Mem**, the principled grounding in dual-process cognitive theory, the clarity and reproducibility of our framework design, and the practical value. Your constructive comments have been invaluable in helping us refine both the technical presentation and the experimental evaluation. For convenience, we summarize below the major strengths, followed by the key revisions we have made in response to the concerns.

---
**Key strengths noted by the reviewers:**

- S1: **Strong conceptual novelty grounded in cognitive science.**
  Reviewers highlighted the clear and principled mapping from Recollection–Familiarity theory to a computational retrieval controller (`R1, R2, R3, R5`).
- S2: **Clear and well-specified framework design.**
  The gating rule, retrieval controller, clustering-based recollection, and α-mixing were praised for their clean formalism and reimplementability (`R1, R2, R5`).
- S3: **Practical, modular, and efficient system behavior.**
  RF-Mem maintains low overhead, uses bounded compute, avoids LLM-in-the-loop expansion, and scales to very large memory stores (`R3, R5`).
- S4: **Strong empirical performance and robustness.**
  Reviewers noted RF-Mem’s superior accuracy–latency trade-off and complementary benefits beyond standard dense retrieval and indexing approaches (`R1, R2, R5`).
- S5: **Clear writing and solid supporting analyses.**
  The paper’s clarity, organization, and extensive sensitivity and qualitative analyses were highlighted as strengths (`R2, R4, R5`).

---

**Key revisions in response to the comments:**

We have revised the paper carefully, with major changes summarized below:

- [R1, R2, R3, R4, R5] To address the request for a **learnable or generalized switching mechanism**, we added a new experiment in `Appendix D.6` where we train a lightweight classifier-based gate on top of the same features (mean similarity and entropy). We also clarify that our original choice of fixed thresholds was motivated by **privacy concern**, and now position learned gating as a complementary option.
- [R2, R4, R5] To better situate RF-Mem with respect to **query expansion and iterative RAG systems**, we added adaptation experiments where RF-Mem is plugged into query expansion and iterative retrieval pipelines in `Sections 3.4.2 and 3.4.3`. We emphasize that RF-Mem can be used as a plug-in controller inside existing RAG workflows.
- [R2, R5] In response to requests for **failure-case analysis**, we added a failure case in `Appendix D.7.2` illustrating cases where the gate selects the wrong mode together with a discussion of the underlying causes.
- [R1, R2, R3] To test **the robustness of the recollection mechanism**, we conducted alternative experiments with alternative recollection strategies *(density clustering, spectral clustering, gate-based expansion, and graph-based retrieval)*. The new results in `Appendix D.8` show that our proposed recollection procedure remains consistently beneficial over the familiarity-only baseline across these variants.
- [R3] Following the insightful suggestion on **early stopping**, we clarify that RF-Mem already includes an implicit early-stop criterion once a satisfactory top-$K$ set is obtained, and we further add an experiment in `Appendix D.9` analyzing depth-based stopping rules and post-hoc reranking by query similarity.
- [R2] Building on the reviewer’s observation about **routing robustness across retrievers**, we performed an additional analysis in `Appendix D.10`. The results indicate that stronger encoders trigger recollection more, whereas weaker ones favor familiarity, offering further insight into the gating behavior.
- [R3] To support **the assumptions on unit-normalized cosine similarity and sub-Gaussian mean-similarity**, we added empirical distribution plots of similarity scores across datasets and retrievers in `Figures 7, 9, 11, and 13`.
- [R1, R5] We appreciate the reviewers’ comments on **privacy and ethics**. We have updated the `Ethics Statement` and `Limitations` sections with additional discussion on potential risks and deployment considerations.
- [R2] We thank the reviewer for highlighting that **similarity-based uncertainty may not fully capture semantic ambiguity**. We have included an explicit note on this point in the `Limitations and Future Work` section together with potential directions for improvement.

All the revised content has been marked in ${\color{blue}\text{blue}}$.

(* We refer to Reviewer eDaP as R1, Reviewer pdxs as R2, Reviewer 52gJ as R3, Reviewer jKrT as R4, and Reviewer QRxk as R5.)

---

Thank you again for your time, careful evaluation, and for managing the review process. We appreciate your effort, and we hope this summary is helpful.

*Sincerely,*

**The Authors**

---

### Author Response · Authors · 2025-12-04
**Author Final Remarks by Authors**

**Dear Reviewers, AC, SAC, and PC,**

We sincerely thank the Reviewers, AC, SAC, and PC for their careful evaluation and constructive feedback throughout the review process. We have addressed all major concerns and incorporated the suggested clarifications and additional experiments where appropriate with marked in ${\color{blue}\text{blue}}$. We are grateful for the reviewers’ positive reassessments and believe that the revisions have further strengthened the clarity, robustness, and practical relevance of the paper.

We appreciate your time and effort, and we hope that the current version is now suitable for acceptance.

---

### **1. Reviewer Score and Feedback Summary**

| Reviewer ID | Score (Initial → Updated) | Feedback Summary |
|-------------|---------------------------|------------------|
| eDaP (R1)   | 6                         | No additional feedback was available at this stage. |
| pdxs (R2)   | 6                         | No additional feedback was available at this stage. |
| 52gJ (R3)   | **4 → 6**                     | *“Most of my concerns have been addressed, so I update my score to 6.”* |
| jKrT (R4)   | 6                         | No additional feedback was available at this stage. |
| QRxk (R5)   | **8 → 8**                     | *“These additional experiments do augment the paper very nicely … I’m happy to maintain my positive score.”* |

---


### **2. Summary of Strengths and Revisions**
For clarity, we briefly restate the key strengths recognized by the reviewers and summarize the main revisions made in response to their comments.

`Strengths.` Reviewers consistently recognized the **conceptual novelty of RF-Mem grounded in dual-process cognitive theory**, together with its **clean, modular framework design**, **computational efficiency**, and **strong empirical performance** across large-scale personalization settings.

`Key revisions.` In response to the reviews, we **expanded the experimental evaluation to validate the robustness of the recollection and routing mechanisms**, demonstrated that **RF-Mem can be integrated as a drop-in controller within more complex retrieval pipelines**, and **clarified key design assumptions, and potential privacy considerations** through additional analyses and discussion.

---

Thank you again for your time, careful evaluation, and for managing the review process. We appreciate your effort, and we hope this remark is helpful.

*Sincerely,*

**The Authors**

---

### Meta-Review · Area_Chair_LDVM · 2025-12-06

**Summary:**

The paper proposes RF-Mem, a dual-process memory retrieval framework for personalized LLMs inspired by cognitive science. It dynamically switches between a fast "Familiarity" path (dense retrieval) and a slower, iterative "Recollection" path (clustering and $\alpha$-mixing) based on uncertainty signals. The reviewers unanimously praised the strong cognitive grounding, the clean and modular design, and the method's ability to scale to large memory stores while maintaining efficiency. Initial concerns focused on the heuristic nature of the gating mechanism, the lack of comparison against stronger LLM-based retrieval baselines (like iterative RAG or query expansion), and the absence of privacy/failure analysis. The authors provided a comprehensive rebuttal, including new experiments with learned gating, integration with advanced pipelines (HyDE, Search-o1), and detailed failure/sensitivity analyses.

**Reviewer Concerns:**

The rebuttal successfully addressed the vast majority of the reviewers' concerns.

Addressed Concerns:
1. Heuristic Gating (R1, R2, R3, R5): Reviewers questioned the robustness of hand-tuned thresholds. The authors added a new experiment training a lightweight classifier (learned gate) on probe features. While the hand-tuned version remains slightly superior due to data scarcity, the learned gate showed competitive performance and stability, validating the chosen signals (mean similarity and entropy).
2. Weak Baselines (R1, R2, R4, R5): Reviewers noted that comparing only against dense retrieval was insufficient given the existence of advanced RAG methods. The authors demonstrated that RF-Mem acts as a plug-and-play controller by integrating it into HyDE (query expansion), Search-o1 (iterative retrieval), and LD-Agent. In all cases, RF-Mem improved performance, proving it complements rather than competes with these architectures.
3. Recollection Robustness (R2, R3): Concerns about the stability of KMeans clustering were addressed by testing alternative mechanisms (DBSCAN, Spectral Clustering, Graph+BFS). The results showed that the proposed $\alpha$-mixing strategy remains effective across different clustering geometries.
4. Privacy and Failure Analysis (R1, R2, R5): The authors added a qualitative failure case analysis (showing where recollection overfits to semantic clusters vs. user intent) and expanded the Ethics Statement regarding privacy risks in surfacing long-term history.

Outstanding Concerns: There are no significant outstanding technical concerns. Reviewer R2 noted that uncertainty based on similarity scores might not capture semantic ambiguity perfectly, which the authors acknowledged as a limitation for future work.

**Reviewer Scores:**

Reviewer 52gJ (R3): Current Score: 6. (Explicitly updated from 4 to 6 after the rebuttal).

Reviewer QRxk (R5): Current Score: 8. (Explicitly maintained the score of 8, citing the "drop-in" capability for complex pipelines as a particularly strong result).

---

### Decision · Program_Chairs · 2026-01-26

Accept (Poster)